# Enhancing neuronal chloride extrusion rescues α2/α3 GABA$_A$-mediated analgesia in neuropathic pain

Louis-Etienne Lorenzo [1,2,13], Antoine G. Godin [1,3,4,13], Francesco Ferrini [1,3,4,5], Karine Bachand[1], Isabel Plasencia-Fernandez[1,4], Simon Labrecque[1], Alexandre A. Girard[1,6], Dominic Boudreau[1,4], Irenej Kianicka[7,12], Martin Gagnon [1,8], Nicolas Doyon[1,9], Alfredo Ribeiro-da-Silva[2,10,11] & Yves De Koninck [1,2,3,4,11]*

Spinal disinhibition has been hypothesized to underlie pain hypersensitivity in neuropathic pain. Apparently contradictory mechanisms have been reported, raising questions on the best target to produce analgesia. Here, we show that nerve injury is associated with a reduction in the number of inhibitory synapses in the spinal dorsal horn. Paradoxically, this is accompanied by a BDNF-TrkB-mediated upregulation of synaptic GABA$_A$Rs and by an α1-to-α2GABA$_A$R subunit switch, providing a mechanistic rationale for the analgesic action of the α2,3GABA$_A$R benzodiazepine-site ligand L838,417 after nerve injury. Yet, we demonstrate that impaired Cl$^-$ extrusion underlies the failure of L838,417 to induce analgesia at high doses due to a resulting collapse in Cl$^-$ gradient, dramatically limiting the benzodiazepine therapeutic window. In turn, enhancing KCC2 activity not only potentiated L838,417-induced analgesia, it rescued its analgesic potential at high doses, revealing a novel strategy for analgesia in pathological pain, by combined targeting of the appropriate GABA$_A$R-subtypes and restoring Cl$^-$ homeostasis.

[1] CERVO Brain Research Centre, Quebec Mental Health Institute, Québec, QC, Canada. [2] Department of Pharmacology & Therapeutics, McGill University, Montreal, QC, Canada. [3] Department of Psychiatry & Neuroscience, Université Laval, Québec, QC, Canada. [4] Graduate program in Neuroscience, Université Laval, Québec, QC, Canada. [5] Department of Veterinary Sciences, University of Turin, Turin, Italy. [6] Ecole Polytechnique, IP Paris, Palaiseau, France. [7] Chlorion Pharma, Laval, Québec, QC, Canada. [8] Centre for Innovation, University of Otago, Dunedin, New Zealand. [9] Finite Element Interdisciplinary Research Group (GIREF), Université Laval, Québec, QC, Canada. [10] Department of Anatomy & Cell Biology, McGill University, Montreal, QC, Canada. [11] Alan Edwards Centre for Research on Pain, McGill University, Montreal, QC, Canada. [12] Present address: Laurent Pharmaceuticals Inc., Montreal, QC, Canada. [13] These authors contributed equally: Louis-Etienne Lorenzo, Antoine G. Godin. *email: yves.dekoninck@neuro.ulaval.ca

Noxious and innocuous somatosensory inputs are segregated and filtered at the level of the superficial dorsal horn (SDH) of the spinal cord by different inhibitory mechanisms[1]. GABA$_A$ and glycine receptor-mediated transmission mechanisms have been shown to be largely involved in this regulation. Thus, a deficit in GABA$_A$ and glycine signaling has been identified as a primary mechanism underlying pathological pain. Blocking inhibition, for example, replicates symptoms resembling those observed in neuropathic pain patients[2]. Yet, conflicting reports indicate that different determinants of the strength of GABA$_A$/glycine-mediated transmission may change in opposing directions. Down-regulation of the K$^+$-Cl$^-$ co-transporter KCC2 has been reported in multiple studies[3–7]. In contrast, what happens to GABA$_A$ and glycine signaling remains controversial: global measurements of the number of inhibitory interneurons, GABA levels, its synthesis enzyme glutamate acid decarboxylase (GAD) or its vesicular transporter (VGAT) as well as GABA$_A$ receptors (GABA$_A$Rs) yield contradictory results[8–18]. In addition, none of these measurements distinguishes changes in pre- vs. postsynaptic inhibition—i.e., on nociceptive primary afferent terminals vs. SDH neurons. Finally, they do not address whether there is a change in the number of synaptic contacts specifically onto SDH neurons and, in particular, if there is a change in the total number of receptors per synapse. In a recent study, we found a decrease in GABAergic terminals in the SDH after peripheral nerve injury (PNI)[15], but no systematic quantitative studies on the nature and degree of receptor expression at SDH synapses are available. This information is important to assess the net functional impact of a deficit in Cl$^-$ homeostasis resulting from KCC2 hypofunction[19]. To fill this gap, we performed a detailed quantitative analysis of the GABA$_A$Rs and glycine receptors (GlyRs) on SDH neurons at inhibitory postsynaptic sites. We used gephyrin labeling as a tool to define these postsynaptic sites because we previously found that gephyrin clusters are exclusively postsynaptic in the SDH, as they are present on dorsal neurons but absent from primary afferent terminals[20,21].

We found a decrease in the number of inhibitory synapses on SDH neurons after PNI. This decrease in number of synapses was however associated with a synaptic scaling, as there was after PNI an increase in GABA$_A$Rs per remaining synapse without any change in synaptic GlyRs. In addition, we found a switch in GABA$_A$R subunit composition towards enriched α2,3 GABA$_A$Rs, which generate longer GABAergic mIPSCs[22,23]. These seemingly opposing KCC2/GABA$_A$R changes raised the question of the best therapeutic strategy to adopt in such conditions. We first tested the impact of targeting the α2,3 GABA$_A$Rs with the positive modulator L838,417[24]. We found that a single injection of this benzodiazepine site ligand produced an analgesic effect in PNI animals, but not in shams, consistent with the switch towards α2,3 subunits after injury. This switch in subunit composition thus provides a mechanistic explanation for the analgesic action of α1-sparing benzodiazepines[25]. Interestingly, at high doses of L838,417, we observed a significant collapse of analgesia. We hypothesized that this was due to KCC2 hypofunction resulting from nerve injury. Indeed, the ensuing impaired Cl$^-$ extrusion capacity of the cells renders them prone to a collapse of the transmembrane Cl$^-$ gradient upon strong Cl$^-$ load induced by enhanced GABA$_A$ receptor activity[26,27]. To challenge this hypothesis, we tested the effect of rescuing Cl$^-$ homeostasis with the KCC2-enhancer CLP257[28] on the analgesic action of the benzodiazepine site ligand L838,417. We found that combining CLP257 and L838,417 produced a synergistic analgesia and reversed the collapse of analgesia observed with higher doses of the benzodiazepine. These findings point to a novel therapeutic strategy for

treatment of neuropathic pain, by combined targeting of the appropriate GABA$_A$R subtypes and restoring Cl$^-$ homeostasis.

## Results

To perform an unbiased quantitative analysis of the changes in markers of GABA$_A$R- and GlyR-containing synapses following PNI, it was important to objectively define the territory in the SDH that corresponded to the central projections of the lesioned afferents. We used rats in which we performed a constriction injury (cuff model) of the sciatic nerve, resulting in a progressive decrease in paw withdrawal threshold down to <2 g between 14 and 28 days post-surgery (Fig. 1a). By this time point, a distinct patch of loss in IB4 labeling was clearly identifiable in lamina II of the SDH in the L4–L5 spinal segments ipsilateral to the nerve lesion (Fig. 1b). We used the territory defined by this IB4 labeling loss to delineate the core of the central projection of injured afferents[15,29] (Fig. 1b). In this territory, IB4 labeling was decreased to 35.9 ± 1.9% of sham animal values (Fig. 1c). A total of 82 rats were used for subsequent quantitative immunocytochemical analyses of markers of GABA$_A$R and GlyR synapses. The loss in IB4-labeled non-peptidergic terminals was not associated with a loss in calcitonin gene related peptide (CGRP)-immunoreactive peptidergic terminals, making the non-peptidergic afferents a better marker to define a lesion after PNI (Fig. 1d).

**Decrease in number of inhibitory synapses after nerve injury.** To study the number of inhibitory synapses, we used a cluster recognition algorithm[20]. Typical examples of this cluster detection procedure are presented in Fig. 1e. Using this approach, we defined binary masks delineating clusters of staining representing either pre- or postsynaptic protein aggregates (Fig. 1e). The global immunostaining intensity levels of post- and presynaptic inhibitory markers did not change (gephyrin, VGAT, GAD65 and GAD67; Fig. 1f). However, even if the intensities of VGAT and GAD65,67 were higher within their own mask (Fig. 1e, g), we found a significant decrease in the number of varicosities for these markers, indicating a reduction in the number of both inhibitory synaptic terminals and postsynaptic densities (Fig. 1e, h). To test whether these changes correspond to alterations in GABA$_A$ or glycine synapses, we also quantified the number of clusters of the α1, α2, α3 and β3 subunits of the GABA$_A$Rs and the α1 GlyRs. We found a significant decrease in the number of clusters of α2, α3 and β3 GABA$_A$Rs and of α1 GlyRs, but not of the α1 GABA$_A$Rs (Fig. 1i). Taken together, these findings indicate a decrease in the number of both synapses with GABA$_A$R and GlyR in the SDH after nerve injury.

**GABA$_A$R synaptic scaling and subunit switch.** Given the loss in afferent terminals in the SDH, the changes in GABA$_A$R clusters we observed are difficult to interpret because GABA$_A$Rs are located both on SDH neurons and on terminals of primary afferents[20,30,31]. In contrast, we have previously shown that gephyrin and GlyR clusters are not located on primary afferent terminals[20]. Thus, we quantified the intensity of GABA$_A$R and GlyR immunostaining within gephyrin clusters (Fig. 2a, c). We found that, globally, the levels of β2,3 GABA$_A$R staining per gephyrin cluster were significantly increased ($P < 0.05$ $t$-test with Welch correction; Fig. 2b, c) whereas the intensity of the β3 GABA$_A$R subunit staining within gephyrin clusters did not change in PNI animals (Fig. 2c) suggesting a synaptic increase of the β2 but not of the β3 GABA$_A$R (Figs. 1i, 2c). The greatest variation was measured for the GABA$_A$R α2 subtype (241 ± 35%, $P < 0.01$ $t$-test with Welch correction; Fig. 2c). In contrast, we did not observe any significant changes in the α1 and α5 subunits of GABA$_A$Rs nor in α1 GlyR ($P > 0.05$ $t$-test with Welch correction;

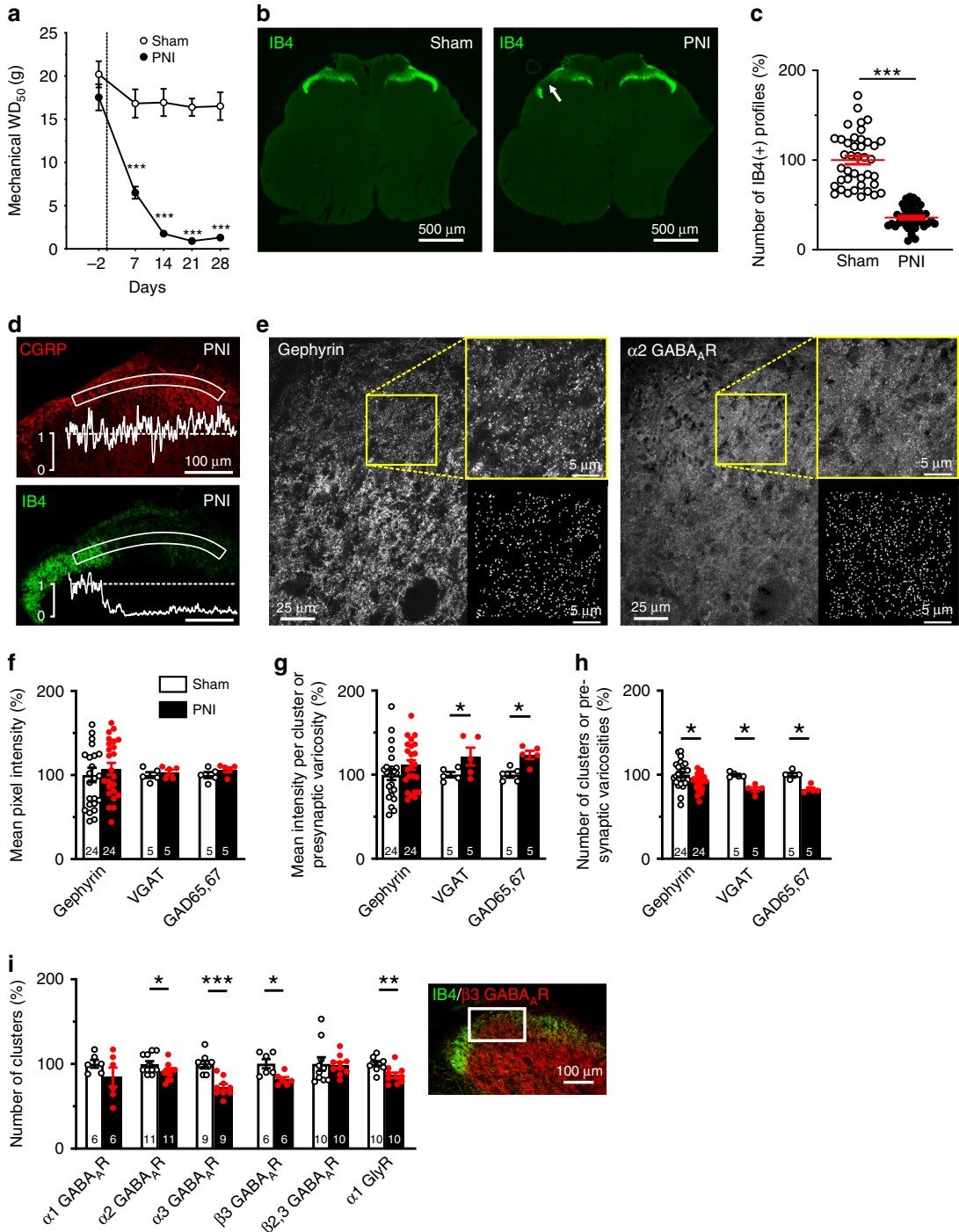

**Fig. 1 Peripheral nerve injury (PNI) causes a loss of inhibitory synapses. a** Time course of the change in paw withdrawal threshold (WD$_{50}$) to mechanical stimulation following PNI in contrast to sham operated rats ($n = 36$ rats, 18 shams and 18 PNIs). In this study, all sham or PNI rats were used between day 14 and 28 post-surgery. **b** Loss of IB4(+) terminals in the ipsilateral SDH (arrow). **c** Quantification of the number of IB4(+) terminals in sham and PNI rats ($n = 82$ rats, 40 shams and 42 PNIs). **d** Intensity profiles of IB4 and CGRP immunolabeling in the superficial ipsilateral SDH from PNI rats. **e** Example confocal images of gephyrin and α2 GABA$_A$R immunolabeling in the SDH to illustrate the binary mask segmentation of clusters (right insets). **f–h** Quantification of: the overall mean pixel intensity **f**, the mean intensity per cluster **g**, and the number of clusters **h** for selected inhibitory synaptic markers in the area defined by the loss of IB4 labeling in PNI rats and corresponding area in sham-operated rats (Fig. 1b, d). **i** Quantification of the number of clusters for a subset of GABA$_A$ and glycine receptor subunits in the area defined for Fig. 1f-h. Inset: example immunolabeling in the region of interest defined by the loss of IB4 labeling. IB4, isolectin B4; CGRP, calcitonin gene related protein; VGAT, Vesicular GABA Transporter. GAD65,67, Glutamatic Acid Decarboxylases 65 and 67; WD$_{50}$, 50% paw withdrawal threshold. Number in each bar represents the number of sham or PNI rats used. Error bars in all panels represent S.E.M. (*$P < 0.05$; **$P < 0.01$; ***$P < 0.001$). Source data is available as a Source Data file.

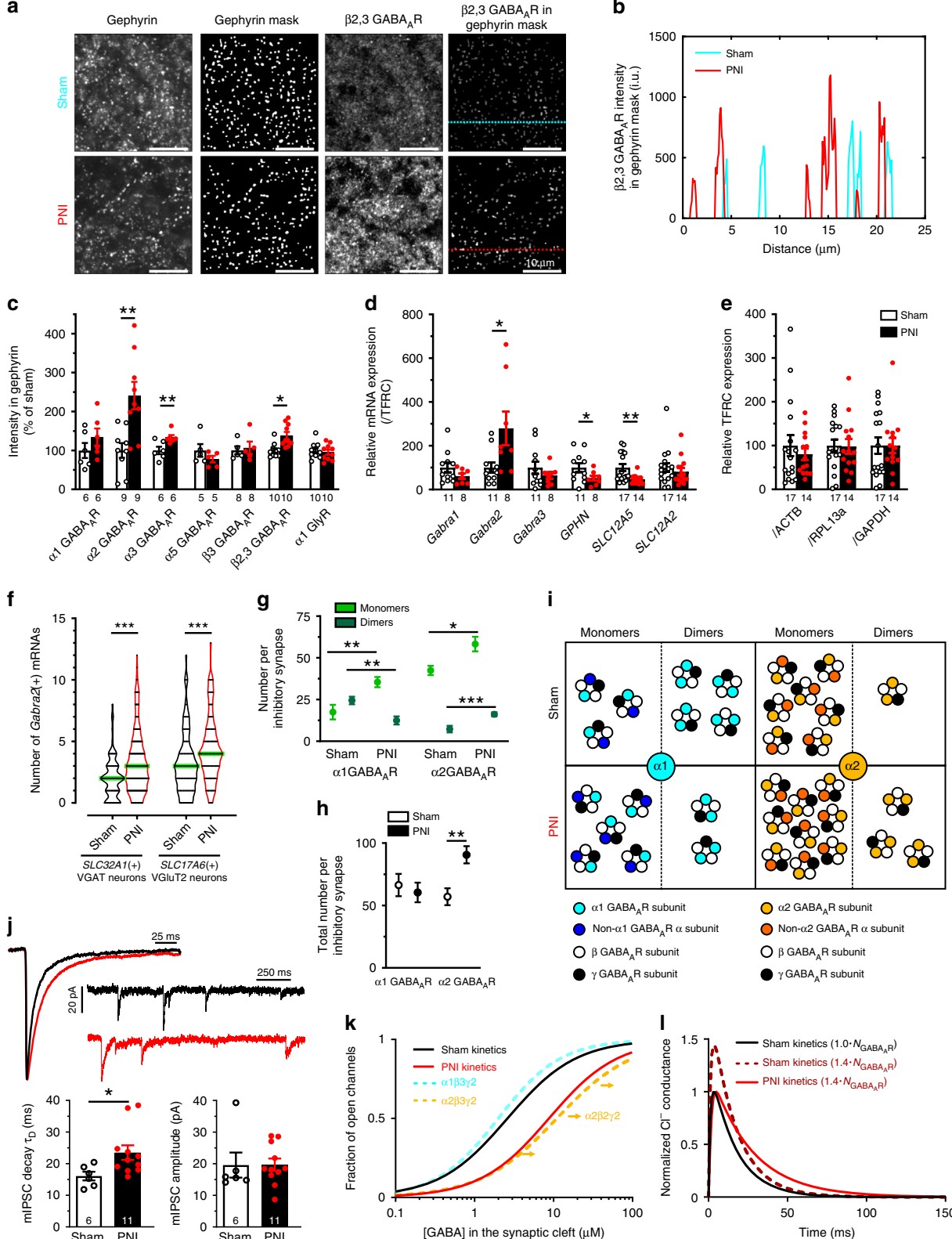

Fig. 2c). The same observations were made with other anti-GABA$_A$R antibodies raised in guinea pig[32] (Supplementary Fig. 1). The increase in β2 subunits in inhibitory postsynaptic sites suggests a synaptic scaling[33,34], whereby the number of receptors per synapse was increased. The increase in α2 and α3 subunit staining, together with no change in α1, points to a switch in GABA$_A$R composition at inhibitory postsynaptic sites.

To compare the immunocytochemical changes with alterations in mRNA expression, we used RT-qPCR of SDHs (Fig. 2d) and found that the expression of mRNA coding for the GABA$_A$R α2 subunit (*Gabra2* mRNA) in PNI animals was significantly greater than that in shams ($276 \pm 108\%$, $P < 0.05$ *t*-test with Welch correction). Interestingly, with the same rats analyzed, no significant change occurs for *Gabra1* or *Gabra3* mRNA, which

**Fig. 2 Postsynaptic scaling and GABA$_A$R subunit switch after peripheral nerve injury (PNI). a** Illustration of the approach used to analyze immunostaining at gephyrin(+) postsynaptic sites, from left to right: gephyrin immunostaining; its binarization (gephyrin mask); β2,3 GABA$_A$R immunostaining in gray scale; the product of the gephyrin mask with the image of the β2,3 GABA$_A$R immunostaining allowing the quantification of intensity of immunolabelled GABA$_A$Rs specifically at inhibitory postsynaptic sites. **b** Profile plot examples of the β2,3 GABA$_A$R immunolabelling intensity in sham vs. PNI animals. **c** Bar graph summarizing the results obtained for GABA$_A$R and α1 GlyR subunits at synapses defined by gephyrin. **d** qRT-PCR in the ipsilateral SDH of *Gabra1, Gabra2, Gabra3, GPHN* (gephyrin), *SLC12A5* (KCC2) and *SLC12A2* (NKCC1) mRNAs in shams vs. PNIs. **e** The house-keeping gene *TFRC* compared to three other house-keeping genes known to be stable after PNI (*ACTB, RPL13a* and *GAPDH*). **f** Violin plot of *Gabra2* mRNA copy index detected by RNAscope in 5 μm-thick z-stack spinal cord sections, in inhibitory *SLC32A1*(+) and excitatory *SLC17A6*(+) neurons of sham and PNI rats. **g** Spatial Intensity distribution analysis (SpIDA) of the α1 and α2 GABA$_A$R monomers or dimers within gephyrin clusters in sham vs. PNI rats. **h** Total number of the α1 and α2 GABA$_A$Rs (-mers) in shams vs. PNIs. **i** Schematic representation of the distribution of oligomerization states for these two subunits in sham vs. PNI rats. **j** Quantitative analysis of the decay time-constant ($\tau_D$) and amplitude of GABA$_A$R mIPSCs in sham vs. PNI rats. Insets: representative electrophysiological traces of GABA$_A$ mIPSC recordings along with superimposed averaged traces (of the 100 first detected mIPSCs). **k** Computer simulations showing the fraction of open GABA$_A$ channels as a function of GABA concentration in the synaptic cleft in sham and PNI rats. For comparison, these relations for pure α1β3γ2 (blue) and α2β3γ2 (orange) GABA$_A$R stoichiometries[39, 41] are also shown. As illustrated by the orange arrows, α2β2γ2 channels are supposed to have slower kinetics than α2β3γ2 channel[41]. **l** Computer simulations of Cl$^-$ conductance during a synaptic event in sham (black) and PNI (red) rats as well as with the assumption of a conductance increase without a change in affinity (i.e., no synaptic switch in GABA$_A$R subunit stoichiometry modifying the affinity for GABA, dashed purple line). i.u., intensity unit. (*$P < 0.05$; **$P < 0.01$; ***$P < 0.001$). Source data is available as a Source Data file.

are associated with the GABA$_A$R α1 and α3 subunits, respectively. Both *Gphn* mRNA transcripts, encoding the inhibitory post-synaptic scaffolding protein, gephyrin, and gene *SLC12A5* encoding KCC2 were significantly decreased to $52 \pm 11\%$ ($P < 0.05$ $t$-test with Welch correction) and $47 \pm 6\%$ ($P < 0.01$ $t$-test with Welch correction), respectively; whereas in the same rat samples, the expression of the NKCC1 gene (*SLC12A2*) did not change significantly ($82 \pm 17\%$, $P > 0.05$ $t$-test with Welch correction). The rate of *TFRC* mRNA, as a house-keeping gene product, was also constant compared to other house-keeping genes that are known to stay stable in the dorsal spinal cord after PNI[35] like *ACTB, RPL13a* and *GAPDH* (Fig. 2e). These results indicate a loss of inhibitory synapse scaffolding protein expression together with changes in opposite directions for *Gabra2* vs. *SLC12A5* gene expressions. The use of RT-qPCR to quantify the number of *Gabra2* mRNA copies did not reveal in which kinds of neurons this up-regulation occurred. For this reason, we used the RNAscope technique in combination with Neurotrace Nissl staining to reveal inhibitory (*SLC32A1*) or excitatory (*SLC17A6*) neurons (Fig. 2f and Supplementary Fig. 2, 3). The up-regulation of *Gabra2* expression occurred in both types of neurons: from $2.0 \pm 0.2$ to $3.3 \pm 0.2$ copy index in *SLC32A1*(+) neurons ($P < 0.001$ $t$-test with Welch correction) and from $3.1 \pm 0.2$ to $4.4 \pm 0.2$ copy index in *SLC17A6* (+) neurons for sham vs. PNI, respectively ($P < 0.001$ $t$-test with Welch correction; horizontal green bars indicate median values for each violin plot in Fig. 2f).

To test for a change in GABA$_A$R subunit stoichiometry after PNI, we applied Spatial Intensity Distribution Analysis (SpIDA)[36] to α1 and α2 GABA$_A$R subunit staining within gephyrin clusters. SpIDA was developed to detect oligomerization states from conventional immunostaining in tissue samples. One of the strengths of SpIDA is that it can be applied to subsets of pixels within single images, such as, in this study, the binary masks defined by the gephyrin staining[36,37], allowing us to analyze subunit stoichiometry at synapses. The analysis revealed an increase in GABA$_A$R α1 monomers but a decrease in α1 dimers whereas both monomeric and dimeric α2 GABA$_A$R increased significantly (Fig. 2g). The total number of subunit count (1 per monomer + 2 per dimer) revealed by SpIDA (Fig. 2h) was consistent with the immunostaining intensity analysis within gephyrin clusters (Fig. 2c). The results are consistent with both a scaling of GABA$_A$Rs at synapses and an α1 to α2 GABA$_A$R subunit switch (Fig. 2h, i).

To determine whether the enrichment in α2,3 GABA$_A$R subunits had an effect on the kinetics of postsynaptic GABA$_A$R events, we recorded miniature (TTX-resistant) GABA$_A$ IPSCs in

SDH neurons. We found that the decay kinetics of GABA$_A$ mIPSCs were significantly slower while their amplitudes were unchanged ($P < 0.05$ Mann-Whitney U-test; Fig. 2j). These modifications in mIPSC decay kinetics are consistent with reported kinetics for α1 vs. α2,3 GABA$_A$R subtypes[38]. Yet, the known lower affinity for GABA of the α2,3 GABA$_A$R subunits compared to that of the α1 GABA$_A$R[39] (Fig. 2k) can explain why the prolongation in GABA$_A$ mIPSCs decay time constant did not yield higher peak amplitudes for these synaptic events, consistent with previous reports for GABA$_A$R[38] or GlyR-mediated mIPSCs[40]. Indeed, the lower affinity subunits yield slower on-rate kinetics, resulting in lower peak channel open probability[39,41] (see simulations in Fig. 2l).

**KCC2 hypofunction does not explain changes in GABA$_A$R expression.** We previously reported a down-regulation of KCC2 in SDH neurons following PNI using an immunoblotting approach[3]. Here, we compared KCC2 immunostaining intensities in sham vs. PNI rats (Fig. 3a–c). Following PNI, KCC2 was downregulated by >75% compared to sham (Fig. 3c). We also measured the distribution of KCC2 immunostaining around the area defined by loss in IB4 labeling. Given that KCC2 is not expressed in primary afferents[3,20], the loss of IB4 does not bias KCC2 quantification. Focusing on this region, however, provides a common reference across sections allowing us to establish more precisely the spatial relationship between KCC2 loss and the region of the SDH where alterations in inhibitory synaptic markers were quantified (Fig. 3b–d and Supplementary Fig. 4). This raised the question of whether the loss in KCC2 may lead to the altered GABA$_A$R expression we observed, as a relationship between the two has been suggested to occur through Cl$^-$ itself[42,43]. To test this hypothesis, we blocked KCC2 in spinal cord slices of control rats with a selective KCC2 antagonist (VU0240551)[28,44–46] and we assessed Cl$^-$ extrusion capacity by measuring $E_{GABA}$ under an imposed intracellular Cl$^-$ load (29 mM), as described in Ferrini et al.[47]. Under these conditions, the measured experimental value for $E_{GABA}$ was more hyperpolarized than the expected value, as calculated by Goldman-Hodgkin-Katz equation ($-44.1 \pm 1.7$ mV vs. $-37$ mV, respectively). Acute administration of VU0240551 induced a significant depolarization of $E_{GABA}$ near to the expected value ($-36.1 \pm 1.0$ mV; $P < 0.001$ paired $t$-test; Fig. 3e, f), indicating a complete block of Cl$^-$ extrusion capacity. Despite the depolarizing shift in $E_{GABA}$ caused by blocking KCC2, no changes in inhibitory postsynaptic markers

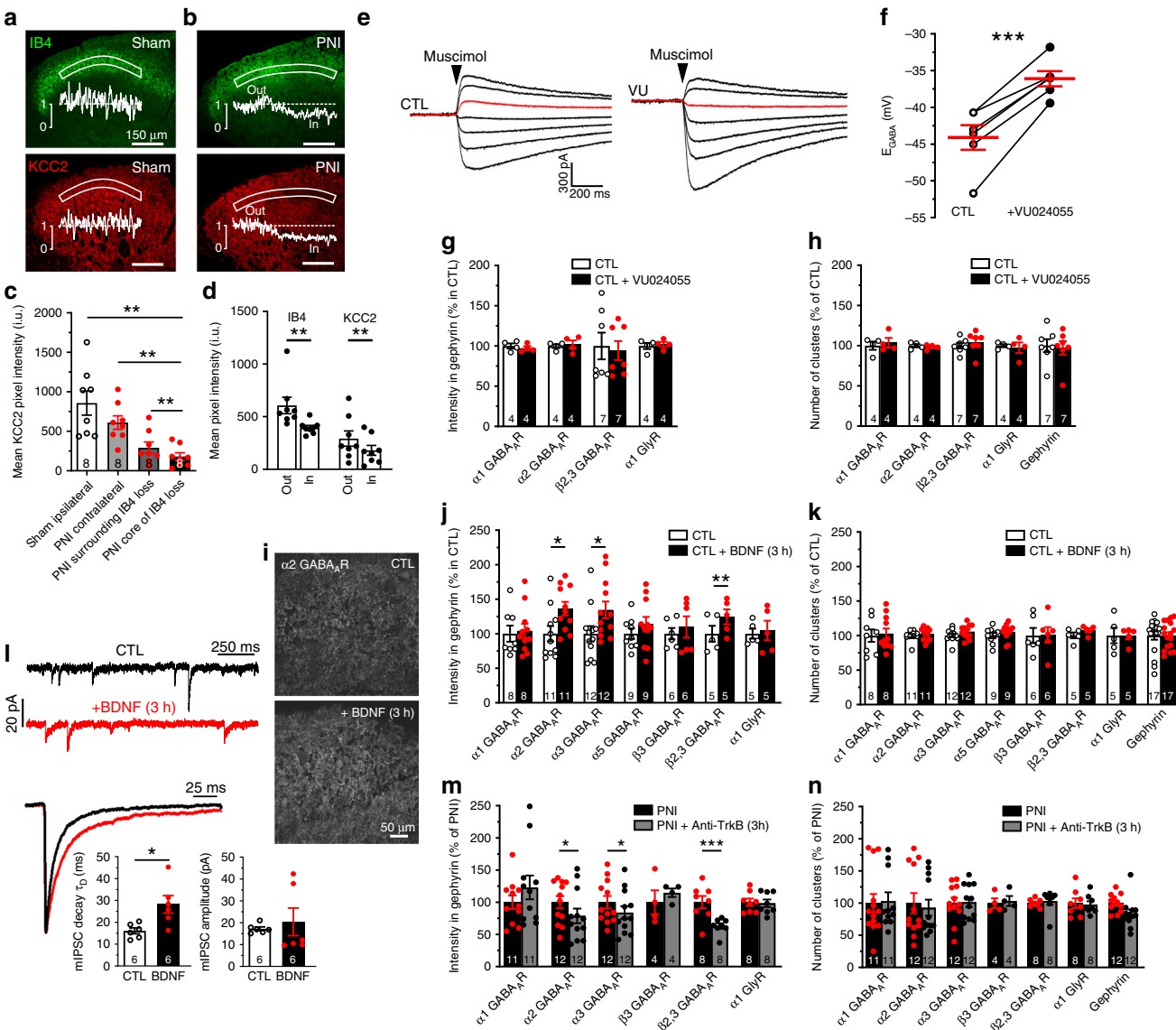

**Fig. 3 Replication of the GABAAR subunit switch through BDNF-TrkB signaling, but not by blocking KCC2. a** Low magnification confocal acquisitions of IB4 and KCC2 immunostainings in sham SDH and their intensity profiles in the SDH **b** id. in PNI rats. **c** From left to right, averaged pixel KCC2 immunostaining intensities in the SDH of the L4-L5 lumbar segments of: ipsilateral SDH of shams, contralateral SDH of PNIs, in the surrounding (´out´) and exactly in the region defined by the loss in IB4 labeling that delineates the core of the central projection of lesioned afferents in PNI rats (´in´). **d** Parallel intensity quantifications of the IB4 and KCC2 immunostaining in the SDH of PNI rats in the same two last regions defined by the loss of IB4 terminals (outside vs. inside). **e** Currents evoked by muscimol puffs in dorsal horn neurons from control rats or in presence of the KCC2 antagonist (VU0240551 at 15 μM) applied at increasing 12.5 mV voltage steps ranging from −93 to −18 mV. In red the muscimol response recorded at –43 mV. **f** $E_{GABA}$ recorded from superficial dorsal horn neurons of naive rats in control and in the presence of VU0240551 (-44.1 ± 1.7 mV, CTL; -36.1 ± 1 mV, VU; $n = 6$ cells). **g** Antagonizing KCC2 did not modify the synaptic location at inhibitory sites of GABAARs and GlyRs. **h** Blocking KCC2 did not affect the number of clusters of the α1, α2 or β2,3 GABAAR nor of the α1 GlyR or gephyrin. **i** Example of the synaptic α2 GABAAR expression increase in control spinal cord slices incubated in 50 ng·ml⁻¹ BDNF in artificial cerebrospinal fluid (ACSF) for 3 h. **j** The quantification of the synaptic GABAAR subunits showed an increase in synaptic α2, α3 and β2,3 GABAAR subunits in BDNF-treated slices. The synaptic expressions in α1, α5 GABAAR and α1 GlyR remained unchanged with BDNF treatment. **k** BDNF however did not modify the number of inhibitory synapses by itself. **l** Examples of GABAA mIPSC traces recorded from control lumbar spinal cord slices incubated in ACSF with or without BDNF and averaged mIPSC aligned by rise time and y-scaled by amplitude. The incubation in a BDNF-containing ACSF significantly prolonged the decay time of the GABAA mIPSC currents in dorsal horn neurons but did not significantly modify their mean amplitude. **m** Blocking BDNF-TrkB signaling with an anti-TrkB antibody in PNI spinal cord paired-sections (2 μg ml⁻¹ in ACSF for 3 h) was able to reverse the GABAAR synaptic over-expression and phenotypic switch in PNI rats, **n** but did not modify the number in GABAAR, GlyR or gephyrin clusters. Numbers of rats are indicated at the bottom of each bar; CTL, control rat. (*$P < 0.05$; **$P < 0.01$; ***$P < 0.001$). Source data is available as a Source Data file.

were observed. The rate of GABA$_A$Rs and GlyR synaptic location (Fig. 3g) and their number of clusters (Fig. 3h) in the SDH remained stable in VU0240551 treated slices. This observation leads to the conclusion that the change in inhibitory markers after nerve injury is not secondary to KCC2 down-regulation.

**The switch in GABA$_A$R subunit composition is BDNF-dependent.** After PNI, activation of spinal microglial P2X4Rs evokes the release of BDNF[48,49] which is known to down-regulate neuronal KCC2 expression[50–54]. To test whether BDNF was also involved in the GABA$_A$R subunit switch, we incubated slices from control animals for 3 h with BDNF (Fig. 3i). We found an increase in the intensity of the α2,3 ($P < 0.05$ $t$-test with Welch correction) and β2 GABA$_A$Rs ($P < 0.01$ t-test with Welch correction), but not in α1,5 GABA$_A$Rs nor in α1 GlyRs at synapses (Fig. 3j). These changes reproduce those found after nerve injury (Fig. 2c). In contrast to PNI animals, however, we observed no significant changes in number of clusters for any of the subunits ($P > 0.05$ $t$-test with Welch correction; Fig. 3k). These results indicate that BDNF is sufficient to replicate the subunit switch, but not the change in number of synapses. These observations indicate that the postsynaptic GABA$_A$R subunit configuration (e.g., scaling and subunit switch) and the number of inhibitory synapses are regulated by distinct mechanisms.

Consistent with our electrophysiological results in PNI animals (Fig. 2j), incubating spinal slices with BDNF caused a significant prolongation of GABA$_A$R mIPSCs without affecting their amplitude (controls $\tau_D = 16.1 \pm 1.3$ ms, BDNF $\tau_D = 28.2 \pm 4.0$ ms, $P < 0.05$ Wilcoxon matched-pairs signed rank test; Fig. 3l). As previously noted, this finding is consistent with the slower kinetics associated with α2,3 and β2 GABA$_A$R subunits[23,38,55,56]. Taken together, our findings indicate that, in addition to a reduced KCC2 expression[3] due to TrkB activation in neuropathic animals[49], GABA$_A$Rs themselves undergo important modifications with a subunit switch and an enrichment in α2,3 and β2 GABA$_A$Rs at inhibitory postsynaptic sites with BDNF (Fig. 3i–k). As shown above, blocking KCC2 alone is not sufficient to generate this synaptic up-regulation in specific GABA$_A$R subtypes, indicating that the change in GABA$_A$R expression is not secondary to TrkB-induced KCC2 downregulation. Our results thus suggest that GABA$_A$Rs and KCC2 may be both regulated by TrkB signaling. We therefore tested whether blocking TrkB in PNI animals could restore the original GABA$_A$R subunit composition. Incubating spinal slices from PNI animals with a function-blocking anti-TrkB antibody[49,57] caused an inverse change in GABA$_A$R subunit expression to that of BDNF applied to control slices (Fig. 3m), without significantly affecting the number of clusters for each of the synaptic markers (Fig. 3n). In conclusion, BDNF-TrkB signaling appears necessary and sufficient to explain the signature of change in postsynaptic GABA$_A$R composition after nerve injury.

**BDNF-induced GABA$_A$R modulation is Ca$^{2+}$-dependent.** As Ca$^{2+}$ has been shown to modulate GABA$_A$Rs[58–61], we investigated the Ca$^{2+}$-dependence of the effects of BDNF and PNI. For this, we used the faster and slower Ca$^{2+}$ chelators BAPTA and EGTA, respectively[62,63] (Supplementary Fig. 6a). In control animals, no significant difference was observed in decay times between BAPTA and EGTA (BAPTA $\tau_D = 21 \pm 2.6$ ms, EGTA $\tau_D = 16.1 \pm 1.3$ ms, $P > 0.05$ Mann-Whitney $U$-test; Supplementary Fig. 6b) but in presence of BDNF, GABA$_A$ mIPSC decays were significantly faster with BAPTA ($\tau_D = 16.7 \pm 1.8$ ms) vs. EGTA incubations ($\tau_D = 28.2 \pm 4$ ms, $P < 0.05$ Mann-Whitney $U$-test; Supplementary Fig. 6a). BAPTA abolished the differences in mIPSC decay time constant between controls, BDNF-

incubated and PNI rat spinal slices (controls, $\tau_D = 21 \pm 2.66$ ms, BDNF, $\tau_D = 16.7 \pm 1.8$ ms, PNI, $\tau_D = 17 \pm 1.4$ ms, $P > 0.05$ Mann-Whitney U-test; Supplementary Fig. 6b). In contrast to BAPTA, EGTA is incapable of clamping certain fast Ca$^{2+}$ fluctuations[64]. Fast intracellular Ca$^{2+}$ transients thus appear necessary for the GABA$_A$R subunit switch.

**L838,417 is analgesic in PNI rats but not in shams.** Antihyperalgesia by α2 GABA$_A$R has been shown to preferentially occur *via* a spinal site of action and not through supraspinal sites[65]. We have shown here that the α2,3 GABA$_A$Rs were up-regulated in the SDH of PNI animals. To test whether the GABA$_A$R α2,3 subunit-preferring benzodiazepine L838,417 was more effective in PNI than in sham animals, we injected 3.3 mg kg$^{-1}$, intraperitoneal (IP) of L838,417[25,66]. Rats were tested for mechanical sensitivity every hour for four hours after IP injection of L838,417 (Fig. 4a). In shams, the mechanical withdrawal threshold (WD$_{50}$) was not significantly affected by L838,417 ($P > 0.05$; paired $t$-test Fig. 4b) whereas in PNIs, it was significantly increased ($P < 0.001$ paired $t$-test; Fig. 4b). The relative improvement in WD$_{50}$ between sham and PNI rats was significantly different ($P < 0.001$ one sample $t$-test; Fig. 4c). We conclude that L838,417 had a significant effect on mechanical WD$_{50}$ in PNI animals only, consistent with the switch to α2,3 GABA$_A$Rs (Fig. 2c,g).

**Dose-response profile of L838,417 and CLP257 analgesia.** The strength of GABA$_A$R-mediated transmission can be significantly impaired by a loss of KCC2 and ensuing degradation in transmembrane Cl$^-$ gradient[67,68]. Furthermore, the efficacy of enhancing GABA$_A$-mediated inhibition with a benzodiazepine may be limited in conditions of impaired Cl$^-$ extrusion, because of the inability of the cells to cope with the Cl$^-$ load[27]. This prompted the hypothesis that enhancing Cl$^-$ extrusion by increasing KCC2 activity may potentiate benzodiazepine analgesia. To test for a synergistic effect of tapping into these two effector mechanisms, we first built dose response curves for analgesia by L838,417 and the KCC2 enhancer CLP257[28] (Fig. 4d–g). For L838,417, the EC50 and the maximum possible analgesia (MPA) derived from fitting the Hill equation (Eq. 2) to the data were $0.43 \pm 0.23$ mg kg$^{-1}$ and $31.2 \pm 4.4\%$, respectively (Fig. 4e). The response at 20 mg kg$^{-1}$ (in gray) was not included in the fit because this point was significantly lower from MPA achieved at 3.3 mg kg$^{-1}$ ($P < 0.05$ paired $t$-test; Fig. 4e). This collapse in analgesia at 20 mg kg$^{-1}$ of L838,417, as reported for other benzodiazepines[69], may result from a collapse in Cl$^-$ gradient. Computer simulations predict that such a collapse arises from high Cl$^-$ load (from high GABA$_A$R conductance; $g_{inh}$) combined with impaired Cl$^-$ extrusion capacity (Fig. 4e, inset and methods). Dose response analysis for CLP257 revealed an EC50 of $29.4 \pm 4.2$ mg kg$^{-1}$ and MPA of $22.9 \pm 1.3\%$ without any collapse in analgesia at high doses (>3× the dose required to reach saturation) in contrast to L838,417 (Fig. 4f, g). To estimate the tissue concentration of CLP257, we analyzed by LC/MS (liquid chromatography/mass spectrometry) the concentration of CLP257 in the CNS at different time points after 100 mg kg$^{-1}$ i.p. injections in vivo. We found that the concentration of CLP257 in brain homogenates was less than 0.2 μM when measured 30 min after i.p. injection and dropped to less than 0.06 μM after 1 h. Thus, the effective concentration at the target site in the spinal cord was in the sub-micromolar range. This indicates a submaximal dose for KCC2 activation at the target site, consistent with the poor pharmacokinetic profile of CLP257[28] (Supplementary Fig. 7).

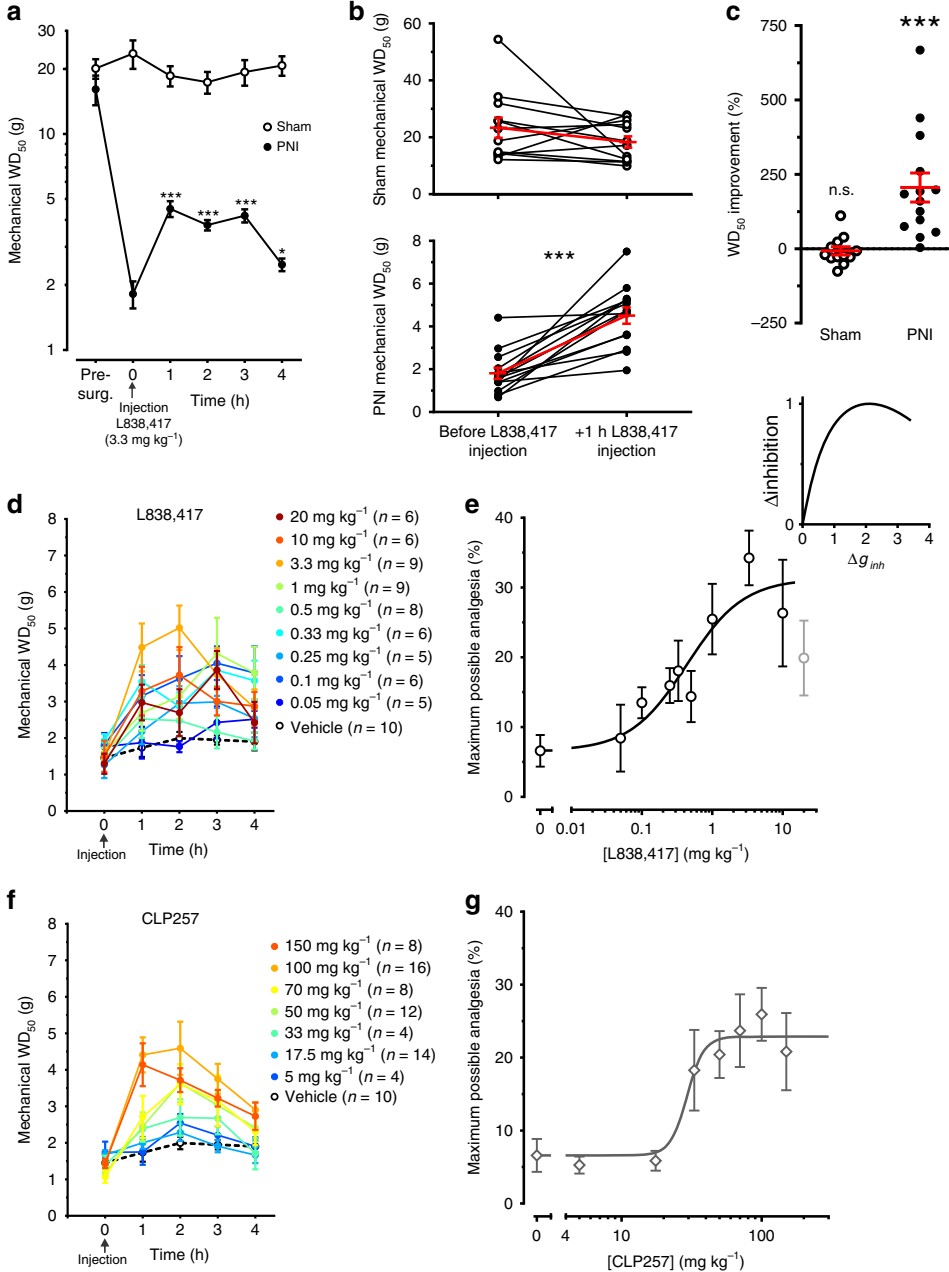

**Fig. 4 Pharmacological characterization of L838,417- and CLP257-induced analgesia. a** Variations in mechanical $WD_{50}$ thresholds over 4 h after IP injection of L838,417 (3.3 mg kg$^{-1}$) in sham and PNI rats, 14–28 day post-surgery. **b** Mechanical $WD_{50}$ thresholds were measured −1 h/+1 h of L838,417 injection in shams and in PNIs. **c** Percentages of change between the two time points in shams vs. PNIs. **d** Partial reversal of mechanical allodynia by different doses of IP injected L838,417 in PNI rats. Animals were tested every hour and followed for 4 h. **e** Dose-response curve of L838,417 in Fig. 4d and fit curve by the Hill equation with variable slope (Eq. 2 in methods; note that the 20 mg kg$^{-1}$ data point was ignored in the fit as the value was significantly lower than the one at 3.3 mg kg$^{-1}$; $P < 0.05$ paired $t$-test). Computer simulations (*Inset*) showing a collapse of net hyperpolarizing anionic current when the inhibitory conductance ($g_{inh}$) is increased beyond an optimal point. **f** Partial reversal of mechanical allodynia by increasing doses of IP injected KCC2 enhancer, CLP257 in PNI rats. **g** Dose-response curve of CLP257 in Fig. 4f and fit curve by the Hill equation with variable slope (Eq. 2 in methods). (*$P < 0.05$; **$P < 0.01$; ***$P < 0.001$). Source data is available as a Source Data file.

**CLP257 enhances membrane expression of KCC2 in PNI animals**. Previous reports have shown that CLP257 acts by enhancing membrane expression of KCC2 in different paradigms[28,70,71]. To confirm this effect in our paradigm, we performed two types of quantitative immunocytochemical analyses. First, we analyzed intensity profiles from hundreds of individual neuronal cell bodies (definable by continuous membrane across the entire circumference) in spinal sections taken

from sham and PNI rats, and immunostained for KCC2 (Fig. 5a). We found that KCC2 profiles displayed an important decrease specifically at the cell membrane in PNIs compared to shams (Fig. 5a, b). In PNI CLP257-treated rats (40 mg kg$^{-1}$), we observed an increase in KCC2 levels at the membrane compared to PNI vehicle injected animals (Fig. 5c, d). Second, to obtain a global index of KCC2 expression at the membrane in both neuronal cell bodies and dendrites, we subtracted the mean pixel

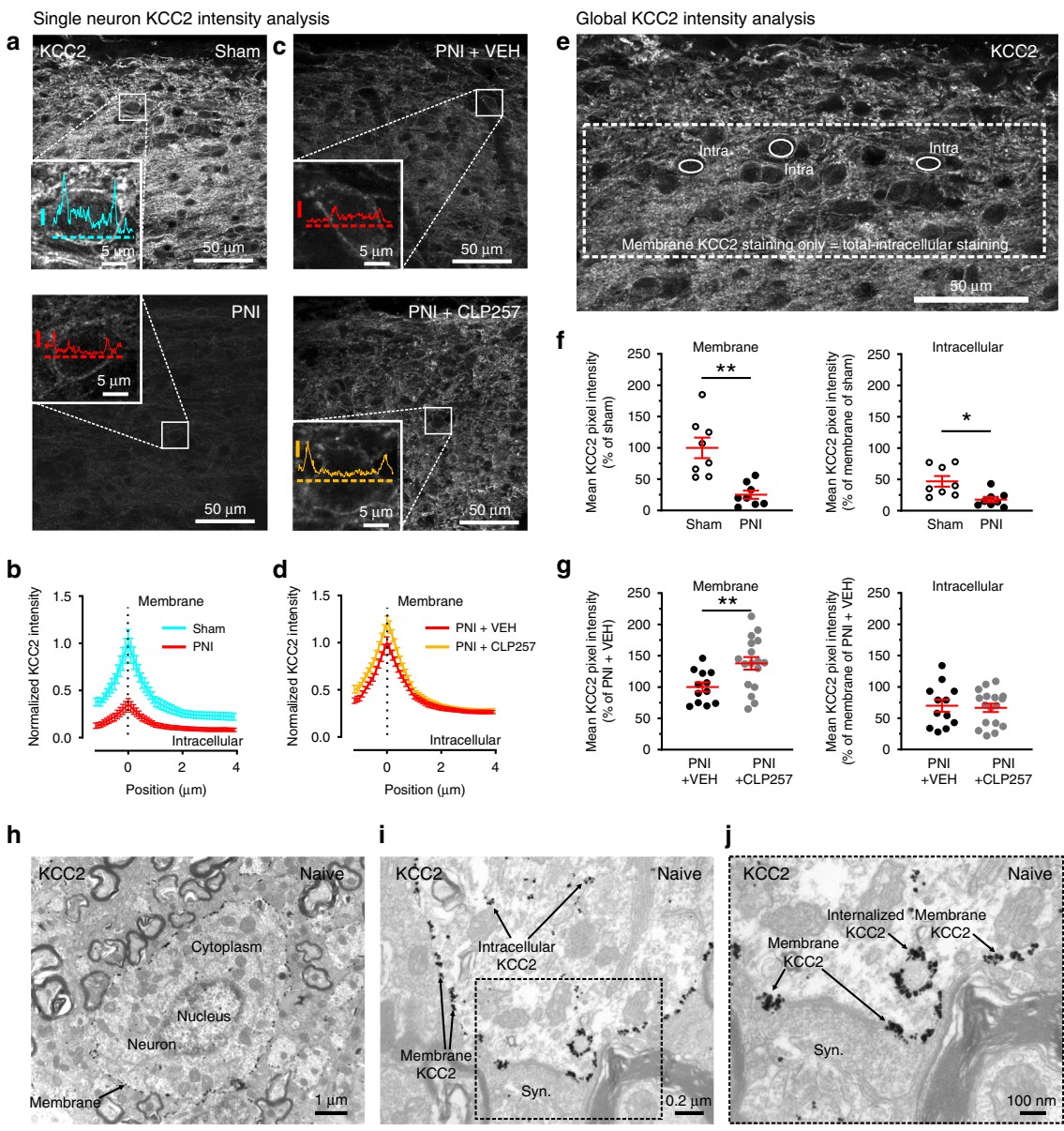

**Fig. 5 CLP257 enhances KCC2 membrane expression in rats with peripheral nerve injury (PNI). a** Confocal laser scanning microscopy (CLSM) images showing KCC2 immunostaining in PNI vs. sham animals. Insets show examples of KCC2 profiles of magnified neurons within these two confocal acquisitions (Y-scale bar = 500 i.u.; horizontal dashed lines are 3 pixel-thick and correspond to a selected KCC2 profile and the intensity axis origin). **b** Averaged intensity KCC2 profiles of visually identified sham and PNI neurons across their plasma membrane. Sham (Cyan) $n = 77$ neurons in 8 rats; PNI (red) $n = 190$ neurons in 8 rats. **c** CLSM images of KCC2 immunostaining in PNI vehicle vs. PNI CLP257 i.t. injected rats. Insets show examples of KCC2 profiles of magnified neurons within these two confocal acquisitions. **d** Averaged intensity KCC2 profiles in PNI + vehicle (red) $n = 268$ neurons in 12 rats and in PNI + CLP257 (orange) $n = 310$ neurons in 17 rats. **e** Method of global index intensity analysis to determine KCC2 staining at the membrane. **f** Global index KCC2 analysis of the averaged KCC2 immunostaining pixel intensity at the membrane and in the intracellular space in PNIs vs. shams. **g** Pixel intensity of KCC2 immunostaining in PNIs where vehicle or 40 mg kg$^{-1}$ of CLP257 was i.t. injected 2 h prior to tissue fixation. Dots in Fig. 5f, g represent single rats. **h** Ultrastructural immunostaining of KCC2 of a dorsal horn spinal cord neuron in a naive rat; subcellular compartments are also displayed. **i** Magnified image showing the subcellular distribution of KCC2. **j** Higher magnification of the area delimited in **i**. Note the membrane KCC2 enriched on both sides of an inhibitory synaptic connection and KCC2 in endosomes. Syn.: synapse; (*$P < 0.05$; **$P < 0.01$). Source data is available as a Source Data file.

intensity in the intracellular compartment (computed from a number of well-defined intracellular areas) from the mean pixel intensity across the entire image area as previously described[71] (Fig. 5e). This global analytic method revealed a decrease of KCC2 immunostaining to $25.1 \pm 6.5\%$ of sham values in PNI animals ($P < 0.01$, $t$-test with Welch correction; Fig. 5f). The average intensity per pixel in the intracellular space (normalized to membrane intensity in sham rats) did also significantly

decrease in PNI animals ($46.9 \pm 8.4\%$ in shams rats vs. $17.8 \pm 4.3\%$ in PNI rats; $P < 0.05$ $t$-test with Welch correction; Fig. 5f). We found that two hours after 40 mg kg$^{-1}$ CLP257 treatment in PNI rats, the membrane KCC2 levels increased to $137.9 \pm 10.1\%$ of sham values ($P < 0.01$ $t$-test with Welch correction; Fig. 5g). In contrast, the average pixel intensity in the intracellular compartment remained stable in both vehicle and CLP257-treated PNI animals (vehicle $70 \pm 9.8\%$ vs. $66.6 \pm 6.8\%$ with CLP257,

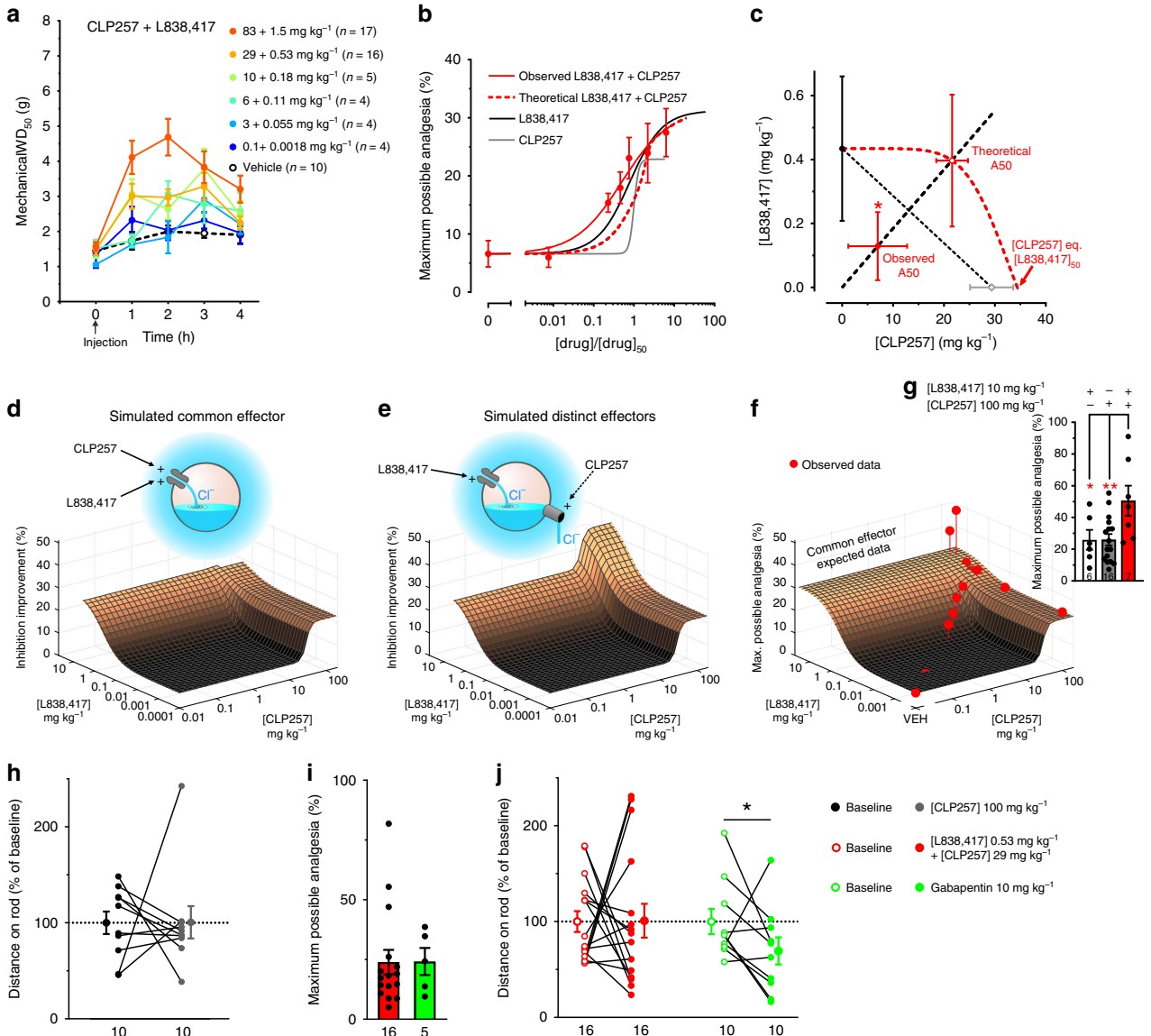

**Fig. 6 Synergistic analgesic action of L838,417 and CLP257. a** Partial reversal of mechanical allodynia produced by L838,417 and CLP257 co-injected at a fixed dose ratio [1:55] in 14-28 day post-surgery PNI rats. **b** Normalized dose-response curve of L838,417 (black line), CLP257 (gray line), expected L838,417 + CLP257 (dashed red line) and observed L838,417 + CLP257 (red line) at different proportions of their EC50s (see methods and Fig. 4). **c** Isobologram of the fits obtained in Fig. 6b. **d, e** Surface plots of computer simulated analgesic effect of co-administration of L838,417 and CLP257 when they act on a common effector (Fig. 6d and Eq. 10) or when they target two distinct mechanisms (inhibitory conductance and KCC2 activity) (Fig. 6e and Eq. 11). **f** 3D plot of the pharmacological synergism between L838,417 and CLP257. Copper-colored surface represents the theoretical analgesia calculated from the respective L838,417 and CLP257 dose-response curves using equivalent dose under the common effector assumption. The experimentally obtained maximum possible analgesia (MPA) for each dose combination tested in PNI animals is indicated by red dots. **g** Over-additive effect of the combined doses of 10 mg kg$^{-1}$ of L838,417 with 100 mg kg$^{-1}$ of CLP257 compared to a single dose of L838,417 or CLP257 or to the expected analgesic effect (MPA ≈ 30%) of the two-drug combination presented in Fig. 6f. **h** Maximum distance on rod of CLP257 at 100 mg kg$^{-1}$ 2 h after i.p. injection. **i** Identical MPA over 4 h for a combination of L838,417 (0.53 mg kg$^{-1}$) with CLP257 (29 mg kg$^{-1}$) vs. Gabapentin (10 mg kg$^{-1}$). **j** Rotarod testing Maximum distance run with the same drugs as in **i**, 2 h after i.p. injection. (*$P < 0.05$; **$P < 0.01$). Source data is available as a Source Data file.

$P > 0.05$ $t$-test with Welch correction; Fig. 5g). These results indicated that CLP257 restores a significant proportion of KCC2 at the neuronal membrane after PNI. Ultrastructural immunostaining for KCC2 using the same antibody, illustrates the distribution of KCC2 in different subcellular compartments in neurons (Fig. 5h–j).

**L838,417/CLP257 act synergistically to produce analgesia.** Our working hypothesis for the paradoxical decrease in analgesia at

high dose with the L838,417 in PNI rats is that enhancement of GABA$_A$R function caused a collapse of the Cl$^-$ gradient due to KCC2 hypofunction (Fig. 4e). Given that CLP257 enhanced KCC2 membrane expression (Fig. 5), we sought to test whether combination of the two drugs would improve analgesia. We administered, in a single bolus, a mixture of L838,417 and CLP257 at a fixed ratio that takes into account the relative affinities of the two drugs (Fig. 4e, g). The experimental mixture of the two compound doses (A50) that produces half of the maximal MPA (Fig. 6b) was significantly lower than the expected concentration calculated

using the isobologram ($P < 0.05$ $t$-test with Welch correction; Fig. 6a–c). This implies that combined doses of L838,417 and CLP257 act synergistically for an increased effect when given simultaneously. Assuming that the analgesic effect of the two drugs is due to their impact on anion current (i.e., the sum of the hyperpolarising $Cl^-$ current and the depolarizing $HCO_3^-$ current through $GABA_A$ channels), we performed two computer simulations to test whether the observed synergistic effect can be explained by the interactions between $Cl^-$ extrusion through KCC2, $Cl^-$ influx through $GABA_A$Rs and net anion current. We first simulated the effects of L838,417 and CLP257 separately and convoluted them to obtain an additive effect as if the they were acting on a common effector (Fig. 6d). But, if we consider that the two drugs act on different mechanisms, L838,417 on the anionic inhibitory conductance ($g_{inh}$) and CLP257 on the $Cl^-$ extrusion capacity ($U_{KCC2}$), they can theoretically act synergistically to increase the $Cl^-$ current through anion permeable channels. This would allow overcoming the upper-limit obtained by the use of a single drug or by their additive effect on a common effector (Fig. 6e). Figure 6f overlays experimental data of MPA measurements of drug combinations as in Fig. 6a, b, and of L838,417 with a saturating dose of CLP257 ($100\,\text{mg}\,\text{kg}^{-1}$) with the expected additive effect based on data obtained from individual dose-response curves (Fig. 4e, g)[72–74]. It shows that drug combinations yielded higher analgesia than expected from mere additive action via a common effector (Fig. 6f). Saturating doses of both drugs produced greater analgesia than that obtained by each compound individually (Fig. 6g). Furthermore, the addition of CLP257 prevented the collapse of analgesia observed at high doses of L838,417, consistent with improved $Cl^-$ homeostasis by KCC2 enhancement (Fig. 6f, g).

To complement our results using the nociceptive reflex response to von Frey stimulation, we used an assay involving a more complex behavioral response[75]. The application of liquid nitrogen with a cotton tip to the hindpaw was used to evoke a combination of prolonged lifting, licking, and shaking behaviors. We similarly found a synergistic effect of combining L838,417 and CLP257 in this assay (Supplementary Fig. 8).

It has been proposed that a significant advantage of using the α2,3-preferring L838,417 is that it can produce analgesia without sedation[25]. We found that CLP257 also does not produce motor side effects (Fig. 6h)[28]. Interestingly, potentiating the action of L838,417 with CLP257 did not induce significant motor impairment at doses that yielded equivalent analgesia to that of gabapentin but which was in contrast affecting time spent on rotarod (Fig. 6i, j)[28].

Finally, we tested the capacity of SDH neurons to maintain their $Cl^-$ gradient in PNI animals in condition of $Cl^-$ load imposed by repetitive $GABA_A$R-mediated synaptic input (Fig. 7). To achieve this, we used repetitive electrical stimulations (Fig. 7a) to evoke monosynaptic inhibitory postsynaptic currents (eIPSCs) in the presence of glutamate receptors blockers. When holding the cells at 0 mV, and delivering a train of 25 pulses at a frequency of 20 Hz, we observed a collapse in amplitude of the eIPSCs as previously described[47,76,77]. We observed a stronger collapse of eIPSCs in the presence of the benzodiazepine site ligand L838,417 as expected from the enhanced $Cl^-$ load produced by potentiation of $GABA_A$R function (enhanced amplitude and decay kinetics; thus, enhanced charge transfer; Fig. 7b). Since CLP257 increases the $Cl^-$ extrusion capacity of the cells, the collapse in eIPSC was significantly reduced when the benzodiazepine was applied in conjunction with either steady-state of 5 μM or within 15 min of 100 μM CLP257 ($P < 0.05$ Wilcoxon matched-pairs signed rank test; Fig. 7c, d). Importantly, adding CLP257 did not affect the properties (peak and decay) of low frequency eIPSCs (not shown) nor of the first eIPSCs at the beginning of each train (Fig. 7b),

indicating that the reversal of the collapse was not due to an effect on $GABA_A$R function, but was rather due to an effect on $Cl^-$ extrusion capacity (Fig. 7a, b). Consistent with this, such a reversal of eIPSC collapse in amplitude was not observed at a holding potential of −90 mV, when the $GABA_A$ currents are dominated by $HCO_3^-$ anions (Fig. 7e, f). Because maintenance of $HCO_3^-$ gradient in the cells is virtually instantaneous compared to the rate of $Cl^-$ regulation by the membrane transporter KCC2, the decline in eIPSC at −90 mV mostly reflects changes in GABA release or $GABA_A$ receptor desensitization rather than ionic plasticity[78,79]. The lack of effect of CLP257 on the decline of inward $HCO_3^-$-mediated eIPSC amplitude therefore indicates that it had no effect on $GABA_A$ receptors nor transmitter release, consistent with selective action of the compound on $Cl^-$ extrusion capacity. These results thus confirmed the postulate made above that CLP257 prevents the $Cl^-$ overload and ensuing collapse of inhibition caused by L838,417.

**The dual advantage of enhancing KCC2 for improved analgesia.** To further validate the postulate that the narrow therapeutic window of L838,417 is due to a collapse in the transmembrane $Cl^-$ gradient, we first conducted computer simulations using a one compartment neuron model which takes into account $Cl^-$ accumulation and extrusion through KCC2[26,27]. We found that increasing only the inhibitory anionic conductance ($g_{inh}$) yielded a collapse in net inhibition when $g_{inh}$ is so large that $Cl^-$ influx overwhelms the $Cl^-$ transmembrane gradient. Increasing in the rate of $Cl^-$ extrusion by 40% was sufficient to prevent the collapse (Fig. 8a). Forty percent of $Cl^-$ extrusion improvement is consistent with the value of increase in KCC2 membrane expression produced by $40\,\text{mg}\,\text{kg}^{-1}$ of CLP257 treatment (Fig. 5, c, d and g). These results indicate that a deficit in $Cl^-$ extrusion can explain the collapse in analgesia at high doses of L838,417 (Fig. 4e). This suggests that the L838,417 dose-response curve is the result of two independent processes. To test this, we modeled a L838,417 dose-response curve as the product of two functions: a normal dose-response curve without collapse (with maximum obtained from Fig. 6g) and an inverse normalized dose-response curve representing the collapse (Fig. 8b). This product of two functions (Eq. 4) dramatically improved the fit quality of the original data set over the fit assuming a single monotonic process (Fig. 8b). These results indicate that combining CLP257 with L838,417 has the dual benefit of pharmacological synergy and extending the therapeutic window of efficacy of the benzodiazepine. Figure 8c summarizes schematically the beneficial effect of the synergistic interaction between a $Cl^-$ extrusion enhancer and a $GABA_A$R positive modulator.

**Discussion**

Loss of inhibition, or disinhibition, at the spinal level has long been hypothesized as underlying hypersensitivity in neuropathic pain. Yet, how $GABA_A$ and glycine receptor-mediated inhibition is modified has remained elusive. Previous studies[13–15] have reported a selective loss in inhibitory GAD-positive terminals in the SDH after nerve injury. It remained uncertain, however, whether this reflected mainly a loss of postsynaptic inhibitory connections. This is because the terminals of sensory afferents, which are important targets of GABAergic inhibition, undergo significant attrition at the spinal level following PNI[29,80]. Here, taking advantage of our previous demonstration that gephyrin clusters are exclusively found on SDH neurons and not on primary sensory terminals[20], we specifically interrogated the postsynaptic component. We found a selective loss in inhibitory postsynaptic sites (attested by the reduction in gephyrin clusters)[40], concurrent to the decrease in number of GAD-positive terminals. Unexpectedly, however,

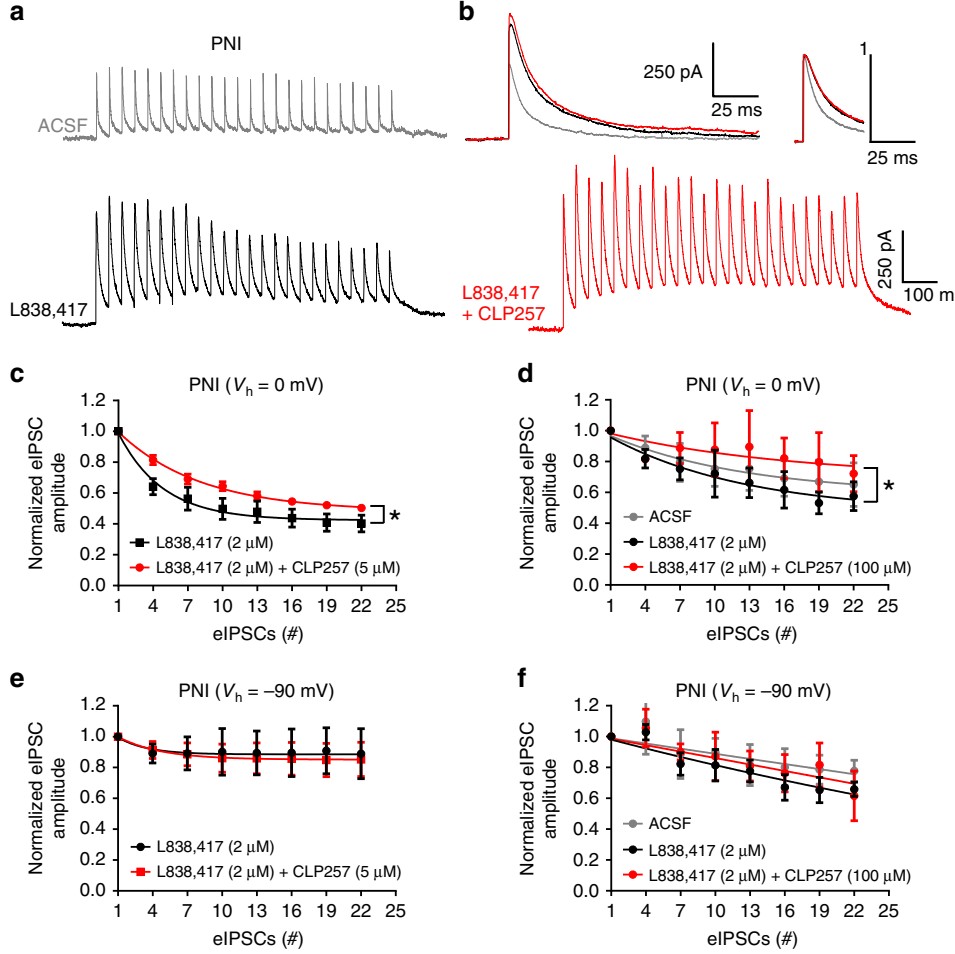

**Fig. 7 CLP257 protects from dynamic collapse in Cl⁻ gradient under a barrage of inhibitory inputs. a** Three representative traces of repetitive inhibitory stimulation (20 Hz, 25 stimulations) at 0 mV in PNI, with ACSF (gray trace), with L838,417 (2 μM, black trace) applied in the bath or with L838,417 and CLP257 (100 μM, red trace) of transversal spinal cord sections. The depression of eIPSCs at 0 mV was amplified by L838,417 in PNI due to Cl⁻ accumulation and can be reversed by CLP257. Stimulus artefacts have been canceled. **b** Representative traces showing the increase in amplitudes and decay times with L838,417 with or without CLP257 vs. ACSF on averaged eIPSCs evoked at 1 Hz. Inset shows the sames traces scaled to peak for kinetics comparison. **c** Averaged eIPSC amplitude depression at 0 mV during the repetitive inhibitory stimulation protocol in two pharmacological conditions: L838,417 (2 μm) and L838,417 (2 μm) + steady-state of CLP257 (5 μM) incubations (one phase decay fits; n = 5 neurons per condition). **d** same stimulations and recording conditions with at least 15 min of pre-incubation of the following conditions: ACSF, L838,417 (2 μm), L838,417 (2 μm) + CLP257 (100 μM) (one phase decay fits; n = 3 to 4 neurons per condition). **e** same as **c** with a holding potential of −90 mV instead of 0 mV. **f** same as **d** with a holding potential of -90 mV instead of 0 mV. eIPSC amplitude were normalized to the average of the first three eIPSCs generated by the 20 Hz stimulation. (*$P < 0.05$). Source data is available as a Source Data file.

the loss of connection was associated with an overall increase in GABA_ARs at the remaining synapses (synaptic scaling) and a phenotypic switch towards the α2,3 subunits on the SDH neurons. In contrast, synaptic α1,5 subunits of the GABA_ARs and α1 GlyRs were statistically unchanged. This observation is consistent with previous findings[81] of a greater plasticity at GABA_AR than GlyR at inhibitory synapses[82]. The stoichiometry SpIDA analysis in α1 and α2 GABA_AR subunit oligomerization[83] displayed a synaptic switch towards α2 monomer and α2 dimer enrichment, reinforcing the importance of the α2 GABA_AR subtype in PNI animals and participating to its ability to reverse pathological pain[25].

It is interesting to note that TrkB signaling appears common to both the loss of KCC2[48,50,51,53,54] as well as to the synaptic scaling at GABA_A synapses and subunit switch, yet the changes in GABA_AR were not secondary to loss of KCC2. We also found that the GABA_AR plasticity was Ca²⁺ dependent. This may explain the fact that previous reports failed to see certain changes in GABA_AR mIPSC kinetics after PNI when recording with low

intracellular pipette Ca²⁺ concentration or strong Ca²⁺ buffering conditions[13,84].

The switch in subunit composition toward α2,3 and β2 is consistent with slower decay kinetics of GABA_AR mIPSCs[39,41]. Switching from α2,3 to α1 GABA_AR composition appears to be a typical change that occurs in several systems during development[85,86]. During the same time period, the β subunits switch from β2 to β3 GABA_AR in the spinal cord[86]. These changes in α and β subunits underlie differences in decay kinetics of synaptic events[38]. The net result is a difference in charge carried, and hence Cl⁻ load. Here, the loss of SDH KCC2 after nerve injury, concurrent to an increase Cl⁻ charge carried by GABA_ARs, raises the intriguing question of the net functional impact of these two opposed phenomena. For example, the resulting apparent increase in GABA_AR tone after nerve injury may be defeated by the loss of KCC2 after nerve injury. The consequent greater Cl⁻ load may in fact be counterproductive and lead to a more dramatic collapse of inhibition[27]. Yet, the subunit switch we observed provides a substrate for the

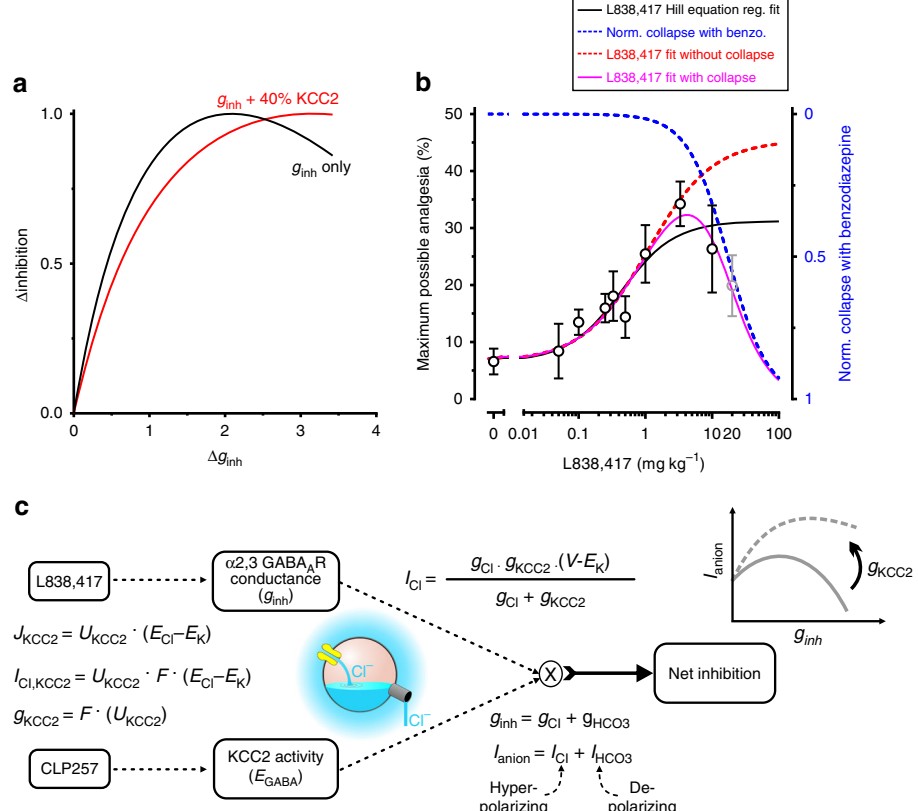

**Fig. 8 Dual advantage of enhancing KCC2: L838,417 potency and efficacy improvement. a** Computer simulations predicting a collapse in anionic current when inhibitory conductance ($g_{inh}$) is increased beyond an optimal value (black curve, see methods). When an increase of 40% in the strength of Cl⁻ extrusion through KCC2 was simulated, the collapse in anionic current for large inhibitory conductance was prevented (red curve as in Fig. 4e). **b** Experimental MPA (maximum possible analgesia) with L838,417 (black empty dots) was fitted with a Hill equation (black curve) where the last data point was ignored (which correspond to a decline in analgesia). To better describe the collapse in inhibition occurring at high L838,417 doses, two independent processes were combined to generate the final fitting function. The fitting function (purple curve, Eq. 4) that takes into account the collapse consists of the product of a normal dose-response curve (Eq. 2) without collapse (dashed red curve with amplitude set to the value obtained in Fig. 6g) and an inverse normalized dose-response curve representing the collapse (dashed blue curve). Combining the two independent processes in a single fitting function (purple curve) yielded a better global fit of the experimental MPA with L838,417 than a simple Hill fit (black curve). **c** A schematic representation of how CLP257 and L838,417 synergistically act on net inhibition. L838,417 increases anionic conductance while CLP257 favors Cl⁻ extrusion via KCC2. This, in turns interacts to determine the value of Cl⁻ reversal potential as well as the net anionic current which promotes analgesia[155]. $F$ is the Faraday constant, $J$ denotes a flux and $U$ is a proportionality constant capturing the strength of KCC2 activity (see Methods). Source data is available as a Source Data file.

analgesic efficacy of α2,3 preferring benzodiazepines after nerve injury[25,66], indicating that, in fact, sufficient residual effective GABA$_A$ inhibition remains after nerve injury to achieve analgesia, despite the loss of KCC2. It also explains the negligible contribution of α1 GABA$_A$R to analgesia after nerve injury[25].

The MPA achieved in our study was at most 50% when combining both L838,417 and CLP257. This contrasts with the MPA reported by Knabl et al., reaching virtually 100% using L838,417 alone[25]. We hypothesized that the difference between our two studies lies in the fact that the level of hyperalgesia achieved in our model was much greater than in Knabl et al. We used a model with higher starting hyperalgesia to better resolve synergism. To test our hypothesis, we conducted an additional set of experiments using larger polyethylene tubing, yielding a WD$_{50}$ of ~5 g (in contrast to the WD$_{50}$ of <2 g normally reached in our study). Under these conditions, a dose of 3.3 mg kg⁻¹ of L838,417 was sufficient to produce ~100% MPA, confirming our assertion (Supplementary Fig. 9).

The hypothesis of synergism between enhancing GABA$_A$R function together with Cl⁻ extrusion holds for postsynaptic inhibition because KCC2 is CNS-specific, i.e., on dorsal horn

neurons[3,19], not primary afferents. Yet, a good part of GABA$_A$R-mediated inhibition in the spinal cord is presynaptic and occurs onto terminals of primary afferents themselves[20,27,30,87]. Interestingly, while this presynaptic inhibition appears to participate in GABA$_A$-mediated analgesia in inflammatory conditions[31], only postsynaptic GABA$_A$Rs appear to be involved in analgesia following nerve injury[31]. The latter is consistent with previous observations of a loss of IB4 + terminals in the SDH after nerve injury, indicating that presynaptic contribution is likely minimal[29,88]. It is also consistent with our finding of an increase in α2,3 GABA$_A$R subunits in postsynaptic sites of dorsal horn neurons after PNI.

It remains that the loss of KCC2 significantly mitigates the efficacy of GABA$_A$ enhancing drugs. A prediction from this is that drugs counteracting the depolarization of $E_{GABA}$ subsequent to increased Cl⁻ influx can enhance the analgesic efficacy of benzodiazepines. Consistent with this was the demonstration that the carbonic anhydrase inhibitor acetazolamide, which causes a collapse of the depolarizing HCO$_3$⁻, enhanced the analgesic efficacy of midazolam[69]. The strategy remains limited however because it does not counteract the deleterious effect of the

collapse in Cl⁻ gradient resulting from KCC2 hypofunction[27]. In contrast, our results with CLP257 indicate that enhancing KCC2 can improve benzodiazepine-induced analgesia on a dual front: enhancing their potency and extending their window of efficacy by preventing their paradoxical loss of efficacy at high doses (Fig. 8), consistent with earlier predictions[28,45]. Inhibiting NKCC1, which normally imports Cl⁻, could appear as a potential alternative to enhancing KCC2. However, the latter has the advantage of offering i) a selective CNS target, avoiding side effect liability of targeting NKCC1 and ii) enhancing Cl⁻ extrusion, thus directly counteracting Cl⁻ load resulting from enhancing GABA$_A$ function[28]. As KCC2 is absent from primary afferents and their terminals[3,20], the synergistic action between L838,417 and CLP257 selectively targets SDH neurons. Such KCC2-enhancer in combination with the proper benzodiazepine can be considered as a promising therapeutic combination. Despite the poor pharmacokinetic profile of CLP257, it was possible to significantly improve benzodiazepine-induced maximal analgesia. This is promising for future, improved versions, of KCC2 enhancers. Our recent finding that microglia-BDNF maintains KCC2 downregulation even at 3 months post-injury in rats indicates that this therapeutic strategy may be relevant to long-term chronic conditions[89].

## Methods

**Animals**. Adult male CD Sprague–Dawley rats, 2 months old at the time of the beginning of the experiments, were used. Rats were housed under a 12 h:12 h light/dark inverted cycle. All experimental procedures were performed in accordance with guidelines from the Canadian Council on Animal Care and the International Association for the Study of Pain.

**PNI pain model and behavioral testings**. PNI was induced by surgically placing a polyethylene cuff (2 mm in length, inner diameter 0.76 mm, Intramedic PE-60, VWR, Canada) around the sciatic nerve of adult rats as previously described[3,49,90]. A second set of PNI rats for complementary experiments was done with PE-90 cuff (inner diameter 0.86 mm, see Supplementary Fig. 9). Another group of rats received sham surgery, in which the sciatic nerve was exposed but no cuff was placed. The 50% paw withdrawal threshold ($WD_{50}$) to mechanical stimulation was assessed, as described[3,28,45,49,91]. Twelve von Frey filaments were used ranging from size 2.44 (0.04 g) to size 5.88 (60 g) in order to adequately test sham as well as lesioned rats (Bioseb Von Frey filament kit). When drugs were used, behavioral baselines were determined just prior to drug single injection ($t = 0$ h ≡ predrug) and then rats were blind-tested at 1 h, 2 h, 3 h and 4 h post-drug injection. The experimenter was blind to drug and dose. Animals were sacrificed 3 weeks post-PNI for tissue processing. Behavioral testing was done in a noise isolated and air conditioned (22 °C) behavior room. The behavioral female[92] experimenters were wearing a 3 M Versaflo TR-300-ECK suit and respirator kit.

As a supplement to simple nociceptive withdrawal reflexes, a more complex behavioral test[75] was added to the up-down Chaplan method[91] described above. Sterile cotton tipped applicators (Puritan; catalog #25-806-1PC) were dipped into liquid nitrogen and then applied under the hindpaw of neuropathic nerve-injured rats (tested between 14 and 28 days post-surgery). Note that one q-tip was used for two liquid nitrogen applications only. For each rat, the test was repeated 5 times spaced by 5 min intervals between each q-tip application and complex behavioral response (E-score) was measured on a semi-quantitative scale of 10 as the sum the 5 successive responses rated as follow: 0 = a rapid transient lifting, licking, or shaking of the hindpaw, which subsides immediately; 1 = lifting, licking, and/or shaking of the hindpaw, which continues beyond the initial application, but subsides within 5 s; 2 = protracted, repeated lifting, licking, and/or shaking of the hindpaw[75]. A baseline response evoked by liquid nitrogen application was obtained one hour before drug i.p. injection. Evoked responses were then measured 2 h post-drug injection. The experimenter was blind to drug and dose.

**Tissue processing**. Before tissue processing in PNI and sham rats, mechanical $WD_{50}$ thresholds were measured to confirm that PNIs were hypersentitive and that shams were not. Rats were anesthetized with a mixture of ketamine/xylazine (0.875 and 0.125 mg kg⁻¹) and quickly sacrificed by decapitation. A skin incision was made along the back of the animal. A 10 ml syringe equipped with an 18 Gauge needle was filled with cold PBS beforehand. The extraction consisted in inserting the needle in the spinal canal at the L6-S1 level and then in ejecting the whole spinal cord by a strong flow of cold PBS through the spinal canal. The spinal cord extracted in this way was immediately frozen in powdered dry ice for one minute. The L4–L5 spinal cord segments were separated from the whole spinal cord with a cold razor blade. In order to keep the tissue frozen and to avoid any protein

denaturation, the spinal cord segments were manipulated with cold forceps on a dish placed upside-down on dry ice. Then the L4–L5 spinal cord segments were placed in pre-cooled 2 ml plastic tubes, which were stored in a −80 °C freezer until further tissue processing. Transverse sections of 14 μm in thickness were cut from the frozen spinal cord with a cryostat (Leica CM 3050S) and mounted onto gelatin-subbed slides (Fisherbrand).

**Immunocytochemistry**. Sections were then immersed for 10 min in freshly depolymerized 4% paraformaldehyde in 0.1 M sodium phosphate-buffered (PB 0.1 M; pH 7.4) and rinsed in phosphate-buffered saline (PBS, 0.01 M; 3 × 10 min). Sections were incubated for 12 h at 4 °C in primary antibody mixtures diluted in PBS containing 4% normal donkey serum (Jackson Immunoresearch; catalog # 017-000-121). Triton X-100 was never added to the solutions. After washing (4 × 5 min), the sections were immersed in a solution containing a mixture of the appropriate fluorochrome-conjugated secondary antibodies diluted (1:500) in PBS (pH 7.4) containing 4% normal donkey serum for 1.5 h, at room temperature. Sections were rinsed in PBS (0.01 M; 4 × 5 min) and incubated in an anti-gephyrin-Oyster antibody and Isolectin-B4 (IB4) solution for 1.5 h (see below for antibody and lectin details). Finally, sections were rinsed (4 × 5 min) and quickly immersed in distilled water and cover-slipped using Aquapolymount (Polysciences). Note that for KCC2 immunostaining, rats were perfused transcardially with 4% paraformaldehyde in 0.1 M phosphate buffer (pH 7.4) for 30 min. Spinal cord segments L4–L5 were collected, postfixed for 60 min in the same fixative, and cryoprotected in 30% sucrose overnight at 4 °C.

**Antibodies**. The antibodies against the α-GABA$_A$R subtypes used in this study were bought from Synaptic Systems (Sysy) and raised in rabbit. A set of complementary experiments using guinea pig anti-α1,2,3 GABA$_A$R antibodies is illustrated in Supplementary Fig. 1, the antibodies were generously provided by Dr. Jean-Marc Fritschy. All these primary polyclonal antibodies were raised against synthetic peptide sequences derived from the GABA$_A$R α1, α2, α3, α5 subunit cDNAs and coupled to KLH[93]. Regarding the Sysy antibodies, the following rat peptide sequences were used: α1 subunit residues 28–43, α2 subunit residues 29–37, α3 subunit residues 29–43, and α5 subunit residues 26–43. All these antibodies were raised in rabbits. The dilutions of the antibodies were: Anti-GABA$_A$R α1 subunit 1:1000 (Sysy Catalog #224 203), α2 subunit 1:1000 (Sysy, Catalog #224 103), α3 subunit 1:1000 (Sysy, Catalog #224 303) and α5 subunit 1:1000 (Sysy, Catalog #224 503). All these antibodies were affinity purified and their affinity has been tested in rats (Sysy). The specificity of the Sysy anti-GABA$_A$R antibodies has been determined by Sysy by using mutant mice lacking the α1 and α2 subunit genes. In our lab, we have tested the anti-α5 GABA$_A$R subunit 1:1000 (Sysy, Catalog #224 503) in Gabra5⁻/⁻ transgenic mice and there was a complete absence of GABA$_A$R immunostaining in both spinal cord[94] and hippocampus. As for the α1, α2, α5 GABA$_A$R subunits, the Sysy anti-α3 GABA$_A$R specificity has been checked by Western-blotting and the correct peptide size was detected (Sysy communication). The signal could also be blocked by pre-adsorption with the specific residues of the corresponding α-subunit. Sysy has also checked cross-reactivity on cells over-expressing the different GABA$_A$R α-isoforms. The Sysy anti-GABA$_A$R antibodies have been sent to a collaborating academic group of Sysy who confirmed the expected tissue distribution by immunohistochemistry. Regarding the Fritschy laboratory antibodies, different amino-acid sequences than Sysy were used: α1 subunit residues 1–16, α2 subunit residues 1–9, α3 subunit residues 1–15, and α5 subunit residues 1–29. All the Fritschy antibodies used here were raised in guinea pig. The dilutions of the antibodies were 1:1000 for every subunit. Immunocytochemical characterization of these polyclonal antibodies has been previously described[20,32,95,96]. Their specificity has been evidenced by using mutant mice lacking the α1, α2, α3, α5 subunit genes. Immunocytochemistry showed a complete absence of immunolabeling of the GABA$_A$ α1, α2, α3, α5 subunits in α1, α2, α3, α5 knockout mice[97,98]. These antibodies have also been well characterized and used immunohistochemically in human tissues, including spinal cord[30,99–101]. The anti-β3 GABA$_A$R subunit antibody was graciously provided by Dr. Werner Sieghart. This antibody was raised in rabbit against the maltose-binding protein and amino-acids 345-408 of the rat β3 subunit, part of the large intracellular loop between putative transmembrane domains M3 and M4[102–105]. The monoclonal anti-β2,3 GABA$_A$R antibody (Millipore Catalog #MAB341, IgG1, clone Bd17) was raised in mouse against the purified GABA$_A$R β-chain from bovine cortex and has been tested in rat[103,104].

The GlyR α1 subunit was detected with a monoclonal antibody (mAb2b, mouse, 1:1000; Sysy; catalog #146111)[106–109]. This monoclonal antibody was raised against a peptide coupled to KLH corresponding to amino-terminal residues 1–10 of the human α1 subunit[110] and showed a single band of 48 kDa on Western blot (data supplied by SySy).

A mouse monoclonal antibody[108,111], raised against purified rat gephyrin, was used in this study (1:1000; Sysy; catalog#147011 mAb7a antibody). The anti-gephyrin mAb7a antibody is highly specific for immunohistochemical investigations of gephyrin as documented in many studies attesting the complete absence of staining in gephyrin knockout mice[112,113]. This antibody does not interfere with isolated domains of gephyrin involved in its own gephyrin clustering and do not interfere neither with the GlyR nor GABA$_A$R clustering at inhibitory postsynaptic sites[114]. As gephyrin immunolabelling identifies a protein associated

with the postsynaptic membrane at inhibitory synapses, its colocalization with GlyR or GABA$_A$R subunits was used as a marker of the postsynaptic location of these receptors[20,115].

The usual strategy of co-immunostaining (i.e., primary antibodies followed by secondary fluorescent antibodies) of gephyrin (mAb7a) together with α1 GlyR (mAb2b) or β2,3 GABA$_A$R (Bd17) could not be applied because the primary antibodies are both mouse monoclonals. To avoid cross-reactions of the secondary antibodies detecting two types of mouse primary antibodies, we modified the protocol. We eliminated the secondary labeling step of gephyrin by using the fluorophore pre-coupled mAb7a anti-gephyrin antibody (1:1000; Sysy #147011C3; anti-gephyrin Oyster-550 nm). Co-immunostainings of anti-gephyrin-Oyster with anti-α1 GlyR (or anti-β2,3 GABA$_A$R) antibodies were processed sequentially as follow: (i) in a first step, the α1 GlyR (or β2,3 GABA$_A$R) were detected using anti-α1 GlyR (or anti-β2,3 GABA$_A$R) primary antibodies (12 h at 4 °C diluted in PBS containing 4% normal goat serum and then rinsed 4 × 5 min). (ii) Then, the anti-α1 GlyR (or anti-β2,3 GABA$_A$R) primary antibodies were revealed by a goat anti-mouse Alexa 647 cross-adsorbed secondary antibody (ThermoFisher Scientific catalog# A-21235; 1:500; 1.5 h at room temperature, rinsed in PBS 4 × 5 min). (iii) Finally, gephyrin was directly revealed by the fluorescent anti-gephyrin-Oyster primary antibody (mAb7a, 1:1000 for 1.5 h at room temperature; rinsed 4 × 5 min in PBS). To check the cross-reaction-free labeling using two primary antibodies raised in the same species, we made sequential co-immunostainings of the anti-gephyrin-Oyster antibody with a mouse monoclonal anti-CGRP antibody (1:2000; Sigma #C7113) instead of α1 GlyR (mAb2b) or β2,3 GABA$_A$R (Fig. 2 and Supplementary Fig. 10). We had shown in a previous report[20] that gephyrin was absent from CGRP-containing primary afferent terminals using the regular primary/secondary protocol. Here, by using the sequential immunostaining protocol with anti-gephyrin-Oyster antibody, we confirmed the absence of gephyrin in CGRP(+) terminals: a non-significant proportion of CGRP(+) pixels (5.8 ± 0.7%) was overlapping with the gephyrin-oyster mask (Supplementary Fig. 10b–d; $n = 6$ naive rats; 24 slices). To conclude, the control of the sequential immunostainings confirms the absence of cross-reaction of the primary mouse anti-gephyrin-Oyster antibodies with the secondary Alexa647 anti-mouse antibodies (Supplementary Fig. 10).

Calcitonin gene-related peptide (CGRP) immunoreactivity was used as a specific marker of nociceptive peptidergic afferent terminals (not present in any other types of axons in the dorsal horn)[116–119]. The polyclonal anti-CGRP antibody (1:2000, Sigma catalog #C8198) was raised in rabbit using synthetic rat CGRP conjugated to KLH as the immunogen. This antibody specifically recognizes the CGRP C-terminal segment (amino-acids 24 to 37) of rat CGRP. This antiserum shows no cross-reactivity with any other peptide except human CGRP and rat and human β-CGRP (data supplied by Sigma); staining was completely abolished when the antiserum was pre-absorbed with rat CGRP. For supplementary Fig. 10, the monoclonal anti-CGRP antibody (1:2000, Sigma catalog #C7113) raised in mouse was used (see explanation for the co-immunolabeling with GephO above). Isolectin-B4 (IB4)-binding was used as a marker of a subset of small diameter primary afferents, including all of the non-peptidergic nociceptive terminals[80,120–123]. IB4 conjugated to Alexa Fluor 488 (1:500, ThermoFisher Scientific / Molecular Probes, catalog #I21411) has been used in this study to label non-peptidergic C-terminals.

A monoclonal purified IgG mouse anti-VGAT (Vesicular GABA Transporter) antibody (1:1000, Sysy, catalog #131011) and a polyclonal detecting the two GAD isoforms anti-GAD65,67 purified rabbit antibody (1:1000, EMD Millipore, catalog #AB1511) were used in this study. Western blot reveals a doublet at approximately 65/67 kDa using this antibody. Immunostaining can be abolished by pre-incubation in 1-10 µg (per ml of diluted antibody) of the corresponding immunogen peptide common to both isoforms (EMD Millipore, catalog #AG252; data from EMD Millipore documentation).

A polyclonal antibody raised in rabbit (Millipore/Upstate, catalog #07-432) was used in this study. This antibody was raised against a His-tag fusion protein corresponding to residues 932–1043 of the rat KCC2 intracellular C-terminal[124,125]. This antibody is highly specific for KCC2 (KCC2a and KCC2b isoforms) and does not share any homologous sequences with other KCCs or co-transporters.

**Confocal microscopy, image segmentation and measurement**. Confocal images were acquired using an Olympus Fluoview FV1000 confocal laser scanning microscope (CLSM). Acquisitions were 12-bit images of size 2048 × 2048 pixels with a pixel dwell time of 12.5 µs. An x20 dry objective was used for low magnification illustrations. An oil-immersion x60 plan-apochromatic objective was used for high magnification CLSM images which were processed for quantification. Laser power was adequately chosen to avoid saturation and limit photobleaching. All the acquisitions were performed with the same laser settings (laser, power, photomultiplier tube (PMT) settings, image size, pixel size and scanning time). To study and compare the distribution of the receptors in the dorsal spinal cord in shams, PNIs or drug-treated spinal cord sections, the first step of the quantification was to manually define specific regions in an image acquired with the CLSM. Homemade MATLAB routines were developed in order to quantify the different sets of images acquired. The fluorescent stainings were studied in specific regions by image binarization of the IB4(+) terminals, the inhibitory presynaptic marker (GAD65,67 and VGAT), inhibitory postsynaptic marker (gephyrin), and for the

different GABA$_A$R and GlyR subunit clusters. The binarization of such structures was defined using an object-based method[126]. The diameter of axon terminal profiles (IB4, VGAT and GAD65,67) or of postsynaptic contacts (gephyrin) and of receptor clusters (GABA$_A$R and GlyR) in CLSM images are of the order of the point spread function (PSF). For this reason, the algorithm was made to recognize structures of a disc shape with dimensions similar to that of the PSF. The disc shape filter was applied to the images to amplify structures with similar shapes and, because of this, the resulting mask was independent of low frequency variations of diffuse signal in different spatial regions and the analysis was not biased by the choice of an arbitrary intensity threshold[20,127]. Binary masks were obtained for each acquired image. Images with labeling for IB4, VGAT, GAD65,67, gephyrin, GABA$_A$Rs subunits and GlyR α1 subunit were used for quantitative analysis. Two types of analysis were developed to quantitatively measure the effect of PNI and the effect of pharmacologically tested controls (incubation with VU024055, BDNF or anti-TrkB antibody) on the spatial distributions of axon terminals, GABA$_A$R and GlyR α1 subunits. In the region of interest (the superficial dorsal horn), under the different conditions just quoted above, we started by quantifying the number of clusters of each marker, which we defined as the number of distinct PSF-scale structures per unit area (µm$^2$)[20,127]. The number of clusters were reported as percentage of values from either sham or control spinal cord sections. In addition to this first measurement, we also measured the fluorescence intensity in intensity unit (i.u.) of 12-bit images of GABA$_A$R and GlyR α1 subunits present at inhibitory postsynaptic sites (within the gephyrin binary mask). For simplicity and for proper comparison between subunit types labeled by immunofluorescence, every analysis was normalized by the appropriate controls. The main goal of this is to present the measurements independently of the variable affinities of the antibodies used for the experiment.

**Mapping of the KCC2 loss in the spinal cord dorsal horn after PNI**. We acquired tile confocal images from a Zeiss LSM 880 microscope with a Plan-Apochromat x20 /0.8 plan air Zeiss objective, zoom 5 (11 × 11 images, pixel size 0.66 µm) of IB4 (labeled with Alexa488, Excitation: 488 nm and Emission Filter: 493–535 nm) and KCC2 (labeled with Cy3, Ex: 561 nm and Emission Filter: 566–625 nm) immuno-cytochemistry labeling. Example images acquired for each labeling are presented in Supplementary Fig. 4. IB4 Images were compared to a selected reference image from the set. Each image was manually translated and rotated to maximize the IB4 structure overlap representing the lamina II. The same transformation was then applied to each corresponding KCC2 image. For each type of images, the images were averaged. This protocol was applied for slices from shams ($N = 3$ rats, 17 slices) and PNIs ($N = 3$ rats, 11 slices). To remove the high frequency fluctuations, a Gaussian filter with 20 µm standard deviation was applied to all KCC2 images. We then designed binary masks of KCC2 areas of interest for all sections, and we only considered the intersection of all these binary masks. In this region, we computed the relative difference, pixels-per-pixel, between the KCC2 average of PNI rats and the one of sham rats $\left(\Delta KCC2 = \frac{KCC2(PNI) - KCC2(sham)}{KCC2(sham)}\right)$. Specific regions of laminae I and II were chosen including the subregions that suffered from the largest loss of IB4 and others where the IB4 labeling was conserved. A deeper region in the dorsal horn corresponding to the laminae III-IV (>180 µm from the white matter)[128] was also added to the analysis for comparison. Histograms of the KCC2 relative differences were calculated and are presented in Supplementary Fig. 4.

**Enhanced resolution microscopy using multi-array Airyscan**. While super-resolution approaches[129] have been very instrumental to decipher biological structures at the nanoscale level in some special circumstances (i.e., isolated cells, brain slices)[130–132] current super-resolution techniques still do not effectively provide significant improvements in resolution in sections of the thickness normally used for immunocytochemistry particularly if extensive field of view need to be studied[129]. This is because improvements in spatial resolution always come in pair with sacrificing image speed acquisition – hence ability to scan large fields of view—and fluorophore flexibility (i.e., very long scan times causing photobleaching limitations). In addition, current super-resolution approaches provide limited multicolor imaging capability necessary for the type of analysis we performed. And these limitations are compounded for tissue-based quantification. Thus, currently, leading super-resolution optical techniques do not provide the required throughput and flexibility to achieve realistically the type of large-scale analysis that we conducted in this study. Beyond the fact that super-resolution would not provide the throughput needed to conduct the quantification we achieved, it is our contention that super-resolution was not needed to address the question we asked. Indeed, to estimate the intensity of well separated clusters as in the system we studied (i.e., the average gephyrin cluster density was 10.5 ± 0.7 and 14.9 ± 0.7 (mean ± S.E.M. for $N = 14$ rats) per 100 µm$^2$ in laminae I and II, respectively), conventional fluorescence microscopy is sufficient and allows an unequaled flexibility to investigate a large number of synapses in single images. Fluorescence intensity is linearly proportional to emitter concentration over a large dynamic scale. This is what allows quantitation of receptor distributions and oligomerization states within well identified clusters over wide fields of view, even with conventional microscopy[37,129,133–138].

To validate that enhanced resolution would not provide significant additional information, we performed additional measurements using a hybrid solution that

allows similar flexibility and field of views as confocal microscopy but that was shown to improve the resolution. A Zeiss LSM 880 microscope equipped with an Airy scan detection unit was used to generate images with higher resolution than with conventional confocal microscopy. This LSM modality uses a multi-array detector, deconvolution and the pixel-reassignment principle to enhance spatial resolution and signal-to-noise-ratio[139]. Using a high numerical aperture oil-immersion objective (Plan-Apochromat 63x/1.4 Oil DIC), we quantified the resolution improvement of our multi-array detector compared to confocal microscopy using 100 nm diameter fluorescent nanospheres (Abberior TS 100 nm). The full width half maximum (FWHM) of the bead intensity profiles showed an ~1.7 improvement compared to conventional confocal microscopy (Supplementary Fig. 11). To estimate the resolution ($\omega_{FWHM}$) of each the imaging modality, the FWHM diameters, obtained from the bead intensity profiles, were deconvolved using the bead diameter value provided by the manufacturer $\omega_{FWHM} = \sqrt{\omega_{fit}^2 - D_{beads}^2}$. We then conducted quantification of GABA$_A$R α2 (Sysy (#224 103) revealed with Cy3 conjugated secondary antibodies; Excitation 561 nm) in gephyrin (Sysy (#147011) revealed with Alexa 647 conjugated secondary antibodies; Excitation 633 nm) masks in the same fields of view, within the same samples, using both modalities: conventional confocal (Meta mode: 530-630 for GABA$_A$R α2 and 570-758 for Gephyrin) and Airy scan (filters: BP 420-480 + BP 495-620, 561 nm laser power of 3.0% and PMT master gain of 750, digital gain of 1 for GABA$_A$R α2 and BP 570-620 + LP 645 633 nm laser power of 20.0% and PMT master gain of 750, digital gain of 1 for Gephyrin), "ChA" 2D SR (super-resolution) capture Airy scan mode, with a pixel size of 0.073 μm, 16-bit image, 8.19 μs of dwell time per pixel. Zoom was of 1.8x and pinhole of 2.5 A.U. as requested by Zeiss. The results show that the intensities integrated between the two modalities correlate (Supplementary Fig. 11h), indicating that the results obtained using the confocal modality are valid to draw the conclusions we drew.

**Real time qPCR, mRNA extraction and cDNA preparation.** TRIzol Reagent (Invitrogen) was used to isolate total RNA from tissues. Briefly, 1 ml TRIzol Reagent was added per 50–100 mg of tissue sample. Then tissue was homogenized with power homogenizer according to the following procedure: Incubate the homogenized sample for 5 min at room temperature, and then add 0.2 ml of chloroform per 1 ml of TRIzol Reagent. Tubes were shaken vigorously by hand for 15 s, and incubated for 2–3 min at room temperature. Samples were centrifuged at 12,000 g for 15 min at 4°C. The aqueous phase (upper layer) of the sample was transferred into a new tube and 0.5 ml of 100% isopropanol were added per 1 ml of TRIzol Reagent used for homogenization. Samples were incubated at room temperature for 10 min and then centrifuged at 12,000 g for 10 min at 4°C. Supernatants were removed from the tubes, and pellets were washed with 1 ml of 75% ethanol per 1 ml of TRIzol Reagent. Samples were centrifuged at 7500 g for 5 min at 4°C. Ethanol was discarded and the RNA pellet was air dried. In order to measure the concentration of RNA, the RNA pellet was dissolved with DEPC H$_2$O. 1 μg total RNA was reverse-transcribed into cDNA by SuperScript TM III Reverse Transcriptase (Invitrogen). Oligo (dT)18 was used as primers. *Real-time qPCR.* PCR was performed with equal amount of cDNA in the LightCycler 480 system (Roche) using LightCycler® 480 SYBR Green I Master (Roche). Reactions (total 20 μl) were incubated at 95 °C for 5 min, followed by 45 cycles of 30 s at 95 °C, 30 s at 58 °C, and 1 min at 72 °C. Water controls and no reverse transcriptase negative control were included to ensure specificity. We quantified the relative expression of *Gabra1,2,3* and *GPHN* mRNAs in sham *vs* PNI rats. The amount of these targeted mRNAs was normalized to the expression of two reference house-keeping genes TFRC (Rat Transferrin Receptor) and Gusb (Rat Glucuronidase-β). Primer efficiencies were integrated to the calculation of the final target/reference gene mRNA ratio by the use of LC480 converter and LinRegPCR softwares. All PCR primers are listed in Supplementary table 1.

**Gabra2 mRNA detection in inhibitory and excitatory neurons.** As in Lorenzo et al. 2008[128], neurons were detected by IHC using NeuroTrace 435/455 Blue Fluorescent Nissl Staining (ThermoFisher Scientific catalog #N-21479) and IB4 lesion in the spinal dorsal horn of PNI animals by using IB4-488 (ThermoFisher Scientific, catalog #I21411 as above). Numbers of transcripts were detected by using the RNAscope fluorescent multiplex assay for fixed frozen tissue of L4–L5 rat lumbar spinal cord segments of 18 μm thick sections (Advanced Cell Diagnostics, ACD)[140]. Probes were designed and provided by Advanced Cell Diagnostics, Inc. The number of *Gabra2* mRNAs (detected by probe catalog #410761-C1) was determined in inhibitory and excitatory neurons (respectively detected by probe #424541-C2 against *Slc32a1* for the detection of VGAT positive neurons; and by probe #317011-C2 against *Slc17a6* for the detection of VGluT2 positive neurons). The probe detections were performed using the RNAscope Fluorescent Multiplex Reagent Kit (catalog #320850) and visualize with a Zeiss Lsm 880 (Plan-Apochromat 63x/1.4 Oil DIC M27 objective, zoom 0.7 and pixel size of 0.094 μm). Z-stacks of 6 images totalizing 5 μm thick section were analyzed. Detection of Neuronal Nissl (+) cell bodies was done using a custom adaptive threshold algorithm in MATLAB. We first applied a 50X50 pixel kernel square convolution, whose pixel sum is equal 1, in order to obtain an image with local average for each pixel. Then, the original image was subtracted from the local average image to obtain an off-set image. Brighter neuronal cell bodies were identified as pixels having intensities higher than the mean plus 0.5-1 standard deviation of the offset image. The adaptive algorithm was coupled to a morphological analysis that scores

cell bodies having a size of more than 500 pixels and maximum eccentricity of 0.98. The function 'imfill' from MATLAB was applied to the sub-threshold cell bodies to close possible holes inside the Nissl staining. The neuronal somata were then extracted from the original image based on their bounding-box, the same bounding-box was used to crop the corresponding RNAscope images. RNAscope images were randomly chosen by MATLAB in order to run blind analysis. Background definition and RNA fluorescent dots were manually drawn by the user using 'imellipse' function in MATLAB. The inhibitory Nissl neurons were considered as inhibitory if they expressed at least one positive fluorescent dot *Slc32a1* (+) mRNA, and as excitatory if the express at least one positive fluorescent dot *Slc17a6* (+) mRNA. The number of *Gabra2*(+) RNA fluorescent dots were counted in each *Slc32a1*(+) and *Slc17a6*(+) Nissl neurons and plotted.

**Spatial intensity distribution analysis.** SpIDA is an analysis technique that was developed to quantify complex mixtures of oligomerization states using fluorescent microscopy images[36,133,141]. SpIDA can measure the concentration and oligomeric states of proteins revealed with fluorescent probes (e.g., fluorescent proteins, fluorescent antibodies). It was previously used to reveal the dimerization of many types of receptors from various cell systems[37]. We sought to investigate the oligomerization states of α1 and α2 GABA$_A$Rs in inhibitory synapses and quantify the spatial distribution shift that occurs after peripheral nerve injury (PNI). We first quantify the monomeric quantal brightness of our fluorescent labeling complex obtained from slices labeled solely with secondary antibodies[133]. As the dominant composition of GABA$_A$Rs appears to be two α's, two β's and one γ subunit[142] a monomer-dimer SpIDA model, using the monomeric quantal brightness, can be applied to study the distribution of any of its subunit type. Because postsynaptic sites are not larger than few beam areas, a single site does not contain a sufficient number of independent pixels to obtain accurate results and averaging over many postsynaptic sites was necessary. To do this, we built a histogram with all the pixels for each GABA$_A$R subunit that was contained within synapses (gephyrin clusters in Figs. 1e, 2a, b) as previously demonstrated[36]. To measure the direct effect at synapses, the values (monomers + dimers) from shams and PNIs are representative of the average number of receptors at the synapse (Fig. 2g–i) assuming the mean cluster size obtained for gephyrin masks (Radius ~ 0.35 μm).

**KCC2 intensity profile analysis.** We sought to quantify the KCC2 expression level at the cellular membrane of single cells. For each immunocytochemistry confocal image, where KCC2 was fluorescently labeled, the average non-specific staining level, quantified in isolated white matter regions, was first subtracted from the whole image. We then manually selected neurons, in the Lamina II of the SDH, which could undoubtedly be defined. We made sure that the neurons selected for this analysis had membrane sections that could be clearly identified on both sides and intracellular signal that was minimally contaminated by out of focus membrane signal (Fig. 5a, c). The intensity profile KCC2 immunostaining of each neuron was then quantified along the selected line. We estimated the position of the membrane, on each side, along the profile by fitting the two maxima. This process was repeated for all selected neurons and for each rat, the average membrane profile was calculated using all the profiles from individual cells. Finally, the final intensity profile for each condition was generated by averaging the profiles from all single rats (Fig. 5b, d). We compared the profiles from sham (77 profiles from $n = 8$ rats) vs. PNI (190 profiles from $n = 8$ rats) rats (Fig. 5b) and PNI + VEH (268 profiles from $n = 12$ rats) vs. PNI + CLP257 (310 profiles from $n = 17$ rats) rats (Fig. 5d).

**Membrane KCC2 index analysis.** As previously presented[51,71], spinal dorsal horn (SDH) corresponds to a complex dense network of cells in which the majority of single cells cannot be clearly separated or delineated. We developed an algorithm to discriminate the KCC2 membrane population from the KCC2 intracellular in the SDH. For this analysis, we defined an index that reflects the global membrane KCC2 intensity and, hence, is not based on manually selection of large easy to delineate cell bodies (Fig. 5e). For each image of an immunostained slice, the average non-specific staining level, quantified in isolated white matter regions, was first subtracted from the whole image. The intensity of the intracellular KCC2 immunostaining was then defined in regions that could undoubtedly be identified as neurons of the SDH. Finally, a large region of the SDH was delineated from which the averaged KCC2 pixel intensity was calculated. Even if one could perfectly delineate the cell membrane, the measurement would still be heavily tainted by the presence of intracellular KCC2 due to the optical resolution as determined by the point spread function. For this reason, to obtain the KCC2 intensity corresponding to the membrane staining, the average intracellular KCC2 intensity value was subtracted to this average KCC2 intensity in the chosen region. This membrane KCC2 intensity index was measured for every rat and the values were averaged. This index is robust and global because it includes many neuronal cell bodies and dendrites and does not depend on arbitrarily visually selected neurons. This index was measured for every rat and the values were averaged using four to six spinal cord sections per rat in every condition ($n = 8$ rats for sham, $n = 8$ rats for PNI, $n = 12$ rats for PNI + vehicle, $n = 17$ rats for PNI + CLP257). Single profile plots of the KCC2 immunostaining intensities in identified dorsal horn neurons are also shown in Fig. 5a, c.

**Transmission electron microscopy**. Four rats were anesthetized with Equithesin and perfused transcardially with a mixture of 4% formaldehyde obtained from paraformaldehyde (PFA), 15% picric acid and 0.1% glutaraldehyde in 0.1 M phosphate buffer (pH 7.4) for 30 min, followed by the same mixture without glutaraldehyde for 30 min and then for 30 min with a solution of 10% sucrose in 0.1 M phosphate buffer (pH 7.4). Spinal cord segments L4–L5 were collected and placed in 30% sucrose in 0.1 M phosphate buffer overnight at 4ºC. Subsequently, spinal cord segments were snap-frozen by immersion in liquid nitrogen, thawed in 0.1 M phosphate buffer at room temperature and cut into 50 µm thick transverse sections with a Vibratome 1000 Plus (TPI). Sections were then incubated for 30 min in 1% sodium borohydride. After extensive washing (5x12 min) in PBS, sections were incubated in 0.5% BSA in PBS for 30 min and then in 5% normal goat serum (Gibco) for 30 min. Then, sections were incubated at 4 °C for 48 h the rabbit polyclonal anti-KCC2 antibody (1:100; Millipore/Upstate, catalog #07-432) also used for confocal microscopy, diluted in PBS with 0.1% BSA. Then sections were washed in PBS (3x10 min), then in 0.1% fish skin gelatin and 0.8% BSA in PBS (2x5 min) and incubated in goat anti-rabbit IgG conjugated to 1 nm gold particles (1:200; Biocell, UK) for 12 h at 4 °C, to label the KCC2 C-terminal tail. Sections were then washed and incubated in 2% glutaraldehyde in PBS for 10 min, washed in PBS (3x5 min), rinsed in distilled water (2 min) and then washed in citrate buffer (3x5 min). Silver intensification of the gold particles was performed for 10 min using a silver enhancement kit (Amersham, UK). Sections were rinsed twice in deionized water then for 5 min in phosphate buffer and incubated in 1% osmium tetroxide ($OsO_4$) in phosphate buffer. Finally, the sections were dehydrated through ascending ethanol concentrations and propylene oxide. Sections were then flat-embedded in Epon (for details see[143]). The regions of interest were selected using light microscopy and re-embedded in Epon blocks. Ultrathin sections were cut with a Reichert ultramicrotome and collected on Formvar-coated one-slot copper grids. The sections were counterstained with uranyl acetate and lead citrate before observation on a Philips 410 LS electron microscope equipped with a Megaview II digital camera.

**Rat spinal cord slices preparation and drug incubation**. Adult male Sprague Dawley rats (around 60 days old) were anesthetized with a mixture of ketamine/ xylazine (respectively 0.875 and 0.125 mg kg$^{-1}$) and perfused intracardially for 15–20 s with ice-cold oxygenated (95% $O_2$ and 5% $CO_2$) sucrose-substituted ACSF (S-ACSF) containing (in mM): 252 sucrose, 2.5 KCl, 1.5 CaCl$_2$, 6 MgCl$_2$, 10 glucose, 26 NaHCO$_3$, 1.25 NaH$_2$PO$_4$, and 5 kynurenic acid, pH 7.35; 360–370 mOsm. The rats were then rapidly decapitated, and the spinal cord was immediately removed by hydraulic extrusion and immersed in ice-cold S-ACSF for 1–2 min.

For immunohistochemistry on slices, lumbar segments (10 mm long) were isolated and glued vertically (rostral side down), their lateral side leaning on a V-grooved agar block itself glued on a metal platform with cyanoacrylate cement. In a cutting chamber filled with oxygenated ice-cold S-ACSF, 600 µm thick transverse sections were cut with a vibrating blade microtome (VT1200S; Leica, Germany). The slices were then incubated in S-ACSF at room temperature (21–25 °C) for 5–15 min and subsequently transferred to a storage chamber filled with oxygenated normal ACSF containing (in mM): 126 NaCl, 2.5 KCl, 2 CaCl$_2$, 2 MgCl$_2$, 10 glucose, 26 NaHCO$_3$, 1.25 NaH$_2$PO$_4$, pH 7.3; 300–310 mOsm at room temperature. The storage chamber was then placed in a warming bath to let the slices recover at 34 °C for about 30–60 min. Half the slices were then transferred to another storage chamber and exposed to human Brain Derived Neurotrophic Factor (BDNF, 50 ng ml$^{-1}$; PeproTech, Rocky Hill, USA) and or KCC2 blocker VU0240551 (VU, 15 µM; Tocris, Cookson, UK) and/or a blocker of TrkB receptor, anti-TrkB antibody, which also acts as a function blocking antibody[49,57] (ATB, 5 µg ml$^{-1}$; R&D Systems). After 3 h incubation, slices were frozen on dry ice and stored at -80 °C for further immunohistochemistry processing.

For electrophysiology, following hydraulic extrusion of the spinal cord, the lumbar segment was isolated and cut in 400 µm thick parasagittal slices, gluing the lateral side down. The same recovering procedure was applied as for immunohistochemistry. All drugs (including BDNF) were bath-applied and prepared in deionized water as 1000× concentrated frozen stock solution aliquots. After 30–60 min incubation in the storage chamber at 34 °C for mIPSC recording or at room temperature for $E_{GABA}$ and repetitive stimulation procedures, slices were transferred to a recording chamber under an upright microscope (BX61WI; Olympus, Tokyo, Japan) equipped with infrared gradient contrast and water immersion-objectives for visualization of neurons in thick live tissue.

**Whole-cell recording from superficial dorsal horn neurons**. For whole-cell voltage-clamp recordings, patch pipettes were pulled from borosilicate glass capillaries (with an inner filament, WPI) using a two-stage vertical puller (PP-83; Narishige, Tokyo, Japan). Recordings were obtained at room temperature, by lowering the patch pipette (<5 MΩ) onto the surface of visually identified neurons in superficial dorsal horn, up to 100 µm from the interface between white and gray matter. Neurons with a healthy appearance presented a smooth surface, and the cell body and parts of the dendrites could be clearly seen. While monitoring current responses to 1 mV pulses, a brief suction was applied to form >1 GΩ seals. A Multiclamp 700B amplifier connected to a Digidata 1440 A data acquisition system and controlled through Clampex software (all from Molecular Devices, Sunnyvale,

CA) were used for the recordings. The access resistance was monitored throughout each experiment. Only recordings with access resistance between 7 MΩ and 30 MΩ, and with stable access (no more than 50% variation) throughout the entire administration of drugs were considered acceptable for analysis.

**Mini-IPSCs recordings**. The slices were perfused at 2 ml min$^{-1}$ in a closed circuit with 50 ml of oxygenated ACSF containing the drugs. The closed-circuit perfusate was changed every 1.5 h. All GABA mIPSC recordings were performed at 32 °C. The pipettes were filled with an intracellular solution composed of (in mM): 100 CsCl, 10 HEPES, 2 MgCl$_2$, 2 ATP, 0.4 GTP, 0.4 CaCl$_2$ and 2 EGTA (or 1 CaCl$_2$, 11 BAPTA), 0.2% Lucifer yellow and 0.2% Neurobiotin (Vector Labs). The pH was adjusted to 7.2 with CsOH, and the osmolarity ranged from 260 mOsm to 280 mOsm. GABA$_A$ mIPSCs were isolated in voltage clamp in presence of the glutamate receptor antagonists 6-cyano-7-nitroquinoxaline-2,3-dione (CNQX, 10 µM; Tocris, Cookson, UK), NMDA receptor antagonist D-2-amino-5-phosphonovaleric acid (DAP-5; 40 µM; Tocris, Cookson, UK), sodium channel blocker tetrodotoxin (TTX, 1 µM; Alomone Labs, Jerusalem, Israel), and glycine receptor antagonist strychnine (0.5 µM, RBI, Natick, USA) at a holding potential of −60 mV. Traces were low-pass filtered at 3 kHz and stored directly on computer. Off-line, the recordings were low-pass filtered at 1 kHz on a computer using the Strathclyde Electrophysiology software (WinEDR; by J. Dempster, Department of Physiology and Pharmacology, University of Strathclyde, Glasgow, UK). The traces were then analyzed with respect to decay time constants with locally designed software.

**$E_{GABA}$ measurement**. $E_{GABA}$ was measured under Cl$^-$ load at room temperature, as previously described[47,68]. The following pipette solution was used (in mM): 115 K-methylsulfate, 25 mM KCl, 2 MgCl$_2$, 10 HEPES, 4 ATPNa, 0.4 GTPNa, pH 7.2. Under these conditions, and assuming a [HCO$_3^-$]$_i$ of 16 mM[78], the theoretical value of $E_{GABA}$ as calculated by the Goldman-Hodgkin-Katz equation is −37 mV. Muscimol (0.5 mM) was applied locally for 30 ms by pressure ejection through a micropipette in presence of in the presence of bath-applied CNQX (10 µM) and APV (40 µM) and TTX (1 µM). The puff pipette was aimed toward the center of the neuronal somata, approximately 5 µm away from the recording pipette. Experimental $E_{GABA}$ was extrapolated from the GABA$_A$ I-V relationships. Traces were low-pass filtered at 4 kHz and analyzed with Clampfit software (Molecular Devices).

**Repetitive stimulation of inhibitory transmission**. For electrical stimulation of the inhibitory synaptic transmission, the following pipette solution was used (in mM): 135 K-methylsulfate, 5 KCl, 2 MgCl$_2$, 10 HEPES, 4 ATPNa, 0.4 GTPNa, pH 7.2. Evoked IPSCs (eIPSCs) were elicited by electrical stimulation (100 µA, 200 µs) delivered focally via a patch micropipette placed in the vicinity of the recorded cell in the presence of CNQX (10 µM) and APV (40 µM), as previously described[144]. Trains of stimuli (24 pulses – 20 Hz) were delivered every 20 s at 0 mV[47,68,77]. The average of ten consecutive stable stimulations for each holding potential was used for subsequent analysis. After every stimulation, peak amplitude was normalized to the first eIPSC. Fits follow an exponential decay relationship between eIPSC amplitude and number of eIPSC. Sections were pre-incubated in ACSF or in CLP257 (5 µM or 100 µM) and L838,417 (2 µM) was bath applied[28] (room temperature) from stock solutions in DMSO (1000 times more concentrated).

**CNS penetration and clearance of CLP257 after i.p. injection**. Concentrations of CLP257 were measured by LC/MS (Liquid Chromatography/Mass Spectrometry) in brain homogenates obtained 30, 60, 120, 240 and 420 min after 100 mg kg$^{-1}$ i.p. injections. CLP257 concentration was reported in pmol·g$^{-1}$ (or nM) or brain weight (assuming a brain tissue density of 1 g ml$^{-1}$; Supplementary Fig. 7). In short, CLP257 was administered i.p. in 20% HPCD to adult male Sprague-Dawley rats ($n = 3$). At the mentioned time points, animals were decapitated using guillotine, and brains carefully removed. They were washed in saline, dissected longitudinally, and snap-frozen in liquid nitrogen. Brain samples were homogenized in 4x volume of sterile water followed by methanol extraction, and dosage of CLP257 was done by LC/MS.

**Pharmacology**. In this section, we characterize the pharmacological properties of L838,417 and CLP257. All drugs were administered intraperitoneally (i.p.) using single injections (4 ml) to adult male Sprague-Dawley rats at the opposite side of the cuff surgery. Drug injections and behavioral testing that follows were done blind to surgery, drug type and drug dose. Drugs were injected one hour prior to the behavioral testing that lasts four hours. In the behavioral comparison between sham and PNI animals L838,417 was diluted in 0.5% methylcellulose and 0.25% Tween 20 diluted in distilled water. In the dose-response curve, L838,417 was diluted in 20% 2-hydroxypropyl-β-cyclodextrin (20% HPCD) in concentrations that varied from 0.05 mg kg$^{-1}$ (0.125 µmol kg$^{-1}$) to 20 mg kg$^{-1}$ (50 µmol kg$^{-1}$). The chloride extrusion enhancer CLP257 (ref. [28]) was also diluted in 20% HPCD and administered i.p. from 5 mg kg$^{-1}$ (16.2 µmol kg$^{-1}$) to 150 mg kg$^{-1}$ (488 mol kg$^{-1}$). For the study of KCC2 immunostaining intensities and profiles in the spinal cord with CLP257, CLP257 was diluted in HPCD and administered by two successive intrathecal (i.t.) injections separated by 30 min (total dose of 40 mg kg$^{-1}$

i.e., 130 μmol kg⁻¹ i.t.). Twelve rats received vehicle (HPCD) i.t. injections and seventeen rats received CLP257.

For dose-response curves, the WD50 thresholds were reported as maximum possible analgesia (MPA) over four hours calculated as follow:

$$\%\text{MPA}(t) = \frac{[\text{WD}_{50}(t) - \text{WD}_{50}(\text{predrug})]}{[\text{WD}_{50}(\text{prePNI}) - \text{WD}_{50}(\text{predrug})]} \cdot 100\% \quad (1)$$

- $\text{WD}_{50}(t) = 50\%$ paw withdrawal threshold at time point $t$,
- $\text{WD}_{50}(\text{predrug}) = 50\%$ paw withdrawal threshold after PNI surgery before drug injection ($t = 1$ h),
- $\text{WD}_{50}(\text{prePNI}) = 50\%$ paw withdrawal threshold baseline before PNI surgery.

Note that MPA cannot be used in sham rat but just in PNI injured animals. The data were fitted using the 4 parameter Hill dose-response curve equation:

$$Y([X]) = Y_0 + (Y_{max} - Y_0) \cdot \frac{[X]^H}{\left([X]^H + \text{EC}_{50}{}^H\right)} \quad (2)$$

$Y([X]) = $ MPA; with $[X] = $ drug concentration; $Y_0 = $ MPA observed with vehicle; $Y_{max} = $ Maximal MPA, $H = $ Hill's slope and $\text{EC}_{50} = $ drug concentration for half of the maximal MPA ($Y(\text{EC}_{50}) = 1/2 \cdot (Y_{max} + Y_0)$).

To test the possible increase in potency of the two drugs when given simultaneously, we injected a total volume of 4 ml of combined doses of L838,417 and CLP257 in a fixed dose ratio of [L838, 417]/[CLP257] = 1:55, given the relative EC50s of the two drugs calculated from the respective dose response curves with a priori assumption on the collapse at high doses of L838,417 (Fig. 4e, g). We then built the isobologram of the corresponding dose-response curves with a fixed dose ratio of R and compared the expected mixture of the two compounds that produced an effect equal to half of the maximal effect produced by the individual drugs and to the experimental value (see later in methods). To also test the maximal possible increase in efficacy of the two drugs when given together, we administered single injections of 4 ml with increasing doses of L838,417 and a constant dose of CLP257 (100 mg kg⁻¹).

If two compounds $a$ and $b$ produce a specified effect that can each be independently parameterized by a four parameter Hill equation (Eq. 2), the additive effect of having $[a]$ & $[b]$ can be modeled *via* the concept of dose equivalency. Starting with a dose-response curve of individual drugs:

$$Y([a]) = Y_0 + (Y_{max}^a - Y_0) \cdot \frac{[a]^{H_a}}{([a]^{H_a} + (\text{EC}_{50}^a)^{H_a})} \text{ and}$$

$$Y([b]) = Y_0 + (Y_{max}^b - Y_0) \cdot \frac{[b]^{H_b}}{([b]^{H_b} + (\text{EC}_{50}^b)^{H_b})}.$$

Then for every dose of drug $b$, a dose of drug $a$ that provides an equivalent effect can be calculated using $Y([b]) = Y(a \equiv a_{eq}(b))$, knowing that $Y_{max}^a \geq Y_{max}^b$. The principle of dose equivalence and the calculations are presented in further details in[72]. Here, it was assumed that both drugs have effects that increase with dose:

$$a_{eq}(b) = \left( \frac{\left((\text{EC}_{50}^a)^{H_a} \cdot (Y_{max}^b - Y_0) \cdot [b]^{H_b}\right)}{(Y_{max}^a - Y_0) \cdot (\text{EC}_{50}^b)^{H_b} - (Y_{max}^b - Y_0) \cdot [b]^{H_b} + (Y_{max}^a - Y_0) \cdot [b]^{H_b}} \right)^{1/H_a}. \quad (3)$$

To generate the additive theoretical response to the presence of two compounds $[a]$ & $[b]$, the dose of drug $b$ has to be transposed to a dose of drug $a$ having an equivalent effect and, finally, the additive effect can be calculated back using the one drug dose-response curve:

$$Y_{2\,\text{drugs}}(a, b) = Y\left(a + a_{eq}(b)\right).$$

Comparing the experimental response of the two compounds, $H_{2\,\text{drugs}}(a, b)$, to additive theoretical response, $Y_{2\,\text{drugs}}(a, b)$ can reveal the evidence of antagonism ($H_{2\,\text{drugs}}(a, b) < Y_{2\,\text{drugs}}(a, b)$) or synergism ($H_{2\,\text{drugs}}(a, b) > Y_{2\,\text{drugs}}(a, b)$) between two compounds.

The drug interaction study between L838,417 and CLP257 was based on the following publications[72–74,145]. We can also quantify the theoretical additive effect of varying the concentrations of $a$ and $b$ using the same ratio ($\text{Ratio}_{b/a}$) between the two drugs, (i.e. $b = \text{Ratio}_{b/a} \cdot a$). Comparing $Y(a + a_{eq}(\text{Ratio}_{b/a} \cdot a))$ to the experimental effect of the same combination of drugs can further reveal evidence pointing toward purely additive, synergic or antagonistic processes between the two compounds. We finally define the isobole curve for which a combination of two drugs induces the same theoretical effect as $\text{EC}_{50}^a$. The isobole can be derived using Eq. 3 by solving $a + a_{eq}(b) = \text{EC}_{50}^a$ for all values of $a \in [0, \text{EC}_{50}^a]$. We define the theoretical A50 as the intersection of the isobole (Fig. 6c, dashed red line) and the line corresponding to the fixed dose ratio $b = \text{Ratio}_{b/a} \cdot a$ (Fig. 6c, dashed black line). The observed A50 corresponds to the drug mixture, on the fixed dose ratio line ($b = \text{Ratio}_{b/a} \cdot a$), that produces the same effect as $\text{EC}_{50}^a$.

In this section, we present how we modeled the effect of the presence of a collapse on a dose-response pharmacological process. Assuming that the two processes are independent, the final fitting function, including the effect of the

collapse, can be calculated simply by the multiplication of a normal dose-response curve without collapse (Eq. 2) and an inverse normalized dose-response curve representing the collapse $C([X]) = 1 - \frac{C_{50}^{H_{col}}}{[X]^{H_{col}} + C_{50}^{H_{col}}}$:

$$Y_{\text{Collapse}}([X]) = Y([X]) \cdot (1 - C([X])). \quad (4)$$

$C_{50}$ is the concentration for which the collapse diminished by a factor 2 the effect of the drug and $H_{col}$ is the corresponding Hill slope of the collapse. Using a non-linear square curve, we obtained the best fit to the data. In Fig. 8b, the three separated curves are presented for the data obtained for L838,417: the normal dose-response curve (dashed red), the collapse (dashed blue) and the resulting fitting function (Eq. 4 in pink, Fig. 8b). For this fit, we set the collapse maximum of the normal dose-response curve to the value obtained from Fig. 6g.

**Statistical analysis**. All data are given as the mean ± S.E.M. Most of the distributions passed normality tests, so $t$-test or paired $t$-test were applied (with Welch correction because standard deviations were often different between sham and cuff animals). For dose-response curve analysis, one-way ANOVA test was applied to compare responses to different dose concentrations of L838,417 or CLP257. When the distributions were not normal, non-parametric tests were used and differences between groups were tested by Mann-Whitney test or by Kruskal-Wallis test with post-hoc Dunn test. Differences were considered to be significant at *$P < 0.05$, **$P < 0.01$ and ***$P < 0.001$. Number in column of bar graphs indicates the numbers of rats analyzed or cells analyzed for electrophysiological recordings.

**Mathematical modeling: mIPSC kinetics**. The first purpose of the computer simulations was to explain the following observation: despite an increase in the number of GABA$_A$R per synapses (in PNI vs. sham rats) indicated by the fluorescence immunolabeling of the GABA$_A$Rs in gephyrin masks analysis (Fig. 2a, c, g–i), the mean amplitude of the events recorded by electrophysiology did not significantly change in both conditions (Fig. 2j). It is known that GABA receptors are unsaturated during synaptic events[146,147] and that the affinity to GABA of α2 and α3 receptors is smaller than the affinity of α1 receptors[38,39]. Hence, we hypothesized that a difference in affinity could explain the fact that event amplitudes do not increase in PNI condition despite an increase in the number of synaptic GABA$_A$Rs (thus in the total conductance). To test whether the affinity hypothesis is consistent with experimental observations, we modeled the channel's dynamics according to the following scheme:

$$\begin{array}{ccccc} & k_1[\text{GABA}] & & k_3 & \\ C_0 & \rightleftarrows & C_1 & \rightleftarrows & O, \\ & k_2 & & k_4 & \end{array}$$

where $C_0$ and $C_1$ correspond respectively to an unbound close state and a bound close state while $O$ stands for an open state. The transition rates $k_1$, $k_2$, $k_3$ *and* $k_4$ stand respectively for the binding rate, unbinding rate, opening rate and closing rate while [GABA] stand for the concentration of GABA neurotransmitters in the synaptic cleft modeled by $[\text{GABA}] = [\text{GABA}]_{max} \cdot e^{-\frac{t}{\tau_{\text{GABA}}}}$ with $[\text{GABA}]_{max} = 0.63$ mM and $\tau_{\text{GABA}} = 1$ ms[148–150]. This model does not aim to describe all the states of the channels which include several desensitized states[39], we rather aimed for the simplest model able to explain the experimental data while more complex models would lead to over-fitting. Assuming that the unbinding rate is the same in PNI and sham conditions $\left(k_2^{\text{sham}} = k_2^{\text{PNI}}\right)$, we fitted the seven parameters $k_1^{\text{sham}}$, $k_2^{\text{sham}} = k_2^{\text{PNI}}$, $k_3^{\text{sham}}$, $k_4^{\text{sham}}$, $k_1^{\text{PNI}}$, $k_3^{\text{PNI}}$ and $k_4^{\text{PNI}}$ to satisfy the following seven constraints: (i-iv). We replicated the rise times and decay times of both sham and PNI conditions as measured in the present study (Fig. 2j). (v) We imposed that the proportion of open channels at maximal amplitude in normal (sham) conditions is consistent with previous reports (>50%)[147,151]. The β2, 3 GABA$_A$R subunits are representative of most of the GABA$_A$R subtype composition in the dorsal horn. (vi) In PNI, at gephyrin synaptic sites, the β2, 3 GABA$_A$R subunits increase of 37.2% in immunostaining intensity (Fig. 2c), so theoretically the total number of GABA$_A$ channels is 37.2% higher in PNI compared to sham, so for similar maximal conductance of GABA$_A$Rs, the proportion of open channels has to be 37.2% smaller in PNI (Fig. 2j). (vii) Finally, we computed the proportion of open channels at steady state for different constant [GABA] values and chose the parameters so that the EC$_{50}$ values of [GABA] (not to be mistaken with the EC$_{50}$ of drugs discussed in the pharmacology section) in sham is consistent with the experimentally observed values[38] (Fig. 2k). Explicitly, we obtained: $k_1^{\text{sham}} = 40$ ms⁻¹, $k_2^{\text{sham}} = 0.8$ ms⁻¹, $k_3^{\text{sham}} = 1$ ms⁻¹, $k_4^{\text{sham}} = 0.15$ ms⁻¹, $k_1^{\text{PNI}} = 11.2$ ms⁻¹, $k_2^{\text{PNI}} = 0.8$ ms⁻¹, $k_3^{\text{PNI}} = 0.43$ ms⁻¹ and $k_4^{\text{PNI}} = 0.06$ ms⁻¹.

Once the values of $k_1^{\text{sham}}$, $k_2^{\text{sham}}$, $k_3^{\text{sham}}$, $k_4^{\text{sham}}$, $k_1^{\text{PNI}}$, $k_2^{\text{PNI}}$, $k_3^{\text{PNI}}$ and $k_4^{\text{PNI}}$ were obtained, we simulated the time course of conductance during an event in both PNI and sham conditions (Fig. 2l) accounting for the observation the maximal conductance 37.2% larger in PNI condition (e.g. the β2,3 GABA$_A$R subunits increase; Fig. 2c). For illustrative purpose, we added a curve obtained with the PNI conductance and sham kinetics (Fig. 2l). Our model also allowed simulating dose response curves for the steady-state proportion of open channels as a function of [GABA] in the synaptic cleft (Fig. 2k) in sham and PNI conditions. This was compared with the experimental dose response curves for various GABA$_A$ channel subtype compositions (Fig. 2k)[39,41]. The kinetics of GABA$_A$R channel opening depend on its composition. These channels display different affinity for [GABA]

depending on the type of $\beta$ subunit, assuming identical $\alpha$ subunit composition, and their kinetics of channel opening follows this trend: $\beta3 > \beta2 > \beta1$[39,41]. The three arrows in Fig. 2k show that the kinetics of $\alpha2\beta2\gamma2$ is slower than the $\alpha2\beta3\gamma2$ stoichiometry.

**Mathematical modeling: inhibitory current**. We also performed computer simulations to explain the experimental analgesic effect under the assumption that the analgesic effect of CLP257 and/or L838,417 is due to the effect of these drugs on the net anionic current through GABA$_A$Rs. We investigated whether the hypothesis that L838,417 and CLP257 act on different effectors (the GABAergic conductance and the KCC2 Cl$^-$ extrusion capacity) is compatible with the experimental synergistic effect (Fig. 6b, c, f, g).

**Anionic current**. GABA$_A$Rs are permeable to both Cl$^-$ and HCO$_3^-$ anions though the permeability to Cl$^-$ ions is about 4 times larger than the permeability to HCO$_3^-$[79,152]. We thus modeled the anionic current ($I_{anion}$) as $I_{anion} = I_{Cl} + I_{HCO_3}$ where $I_{Cl} = g_{Cl} \cdot (V - E_{Cl})$ and $I_{HCO_3} = g_{HCO_3} \cdot (V - E_{HCO_3})$. The apparent conductances with respect to Cl$^-$ and HCO$_3^-$ anions are fractions of the total inhibitory conductance ($g_{inh}$). Explicitly, we have $g_{Cl} = x \cdot g_{inh}$ and $g_{HCO_3} = (1 - x) \cdot g_{inh}$. Though this is a simplification with respect to the more complete Goldman-Hodgkin-Katz (GHK) flux formalism, this is sufficient for the qualitative purpose of the present investigation.

**The equilibrium value of E$_{Cl}$.** As described in Dijkstra et al.[153], the flux of Cl$^-$ through the KCC2 cotransporter is given by $J_{Cl} = U_{KCC2} \cdot (E_{Cl} - E_K)$. The strength of the KCC2 cotransporter is denoted by $U_{KCC2}$ and is in units of mol·s$^{-1}$·V$^{-1}$ while the Cl$^-$ flux has units of mol · s$^{-1}$. The Cl$^-$ current through KCC2 is in turn given by $I_{Cl,KCC2} = U_{KCC2} \cdot F \cdot (E_{Cl} - E_K)$ in units of C · s$^{-1}$, where $F$ is the Faraday constant. To simplify notation and allow easier comparison between synaptic Cl$^-$ influx and efflux through KCC2, we set $g_{KCC2} = F \cdot U_{KCC2}$ and obtain $I_{Cl,KCC2} = g_{KCC2} \cdot (E_{Cl} - E_K)$. We assume the value of $E_{Cl}$ to be such that Cl$^-$ concentration is at equilibrium, that is the Cl$^-$ synaptic influx and extrusion through KCC2 cancel each other out, this yields

$$E_{Cl} = \frac{g_{inh} \cdot V_{mean} + g_{KCC2} \cdot E_K}{g_{inh} + g_{KCC2}}$$

where $V_{mean}$ is the time averaged value the membrane potential taken to be $V_{mean} = -60$ mV. We can thus write the value of $E_{Cl}$ as a function of $g_{inh}$ and $g_{KCC2}$, namely $E_{Cl} = E_{Cl}(g_{inh}, g_{KCC2})$ and in turn the net effective anionic current as a function of $g_{KCC2}$ and $g_{Cl}$

$$I_{anion} = I_{anion}(g_{inh}, g_{KCC2}) = x \cdot g_{inh} \cdot (V_{eff} - E_{Cl}(g_{inh}, g_{KCC2})) \\ + (1 - x) \cdot g_{inh} \cdot (V_{eff} - E_{HCO_3})$$

where $V_{eff}$ is the value of $E_{anion}$ for which there is no effective inhibition (no impact on the firing rate of the neuron). Due to shunting inhibition, this value is more depolarized (-55 mV) than the mean value or resting value of membrane potential[154].

**The values of g$_{inh}$ and g$_{KCC2}$ in PNI with no drug administration**. In order to investigate the effect of increasing $g_{inh}$ and $g_{KCC2}$ through drug administration, we first had to infer their baseline values (i.e., the basal values in PNI condition) that we denote $g_{KCC2}^{PNI}$ and $g_{inh}^{PNI}$. As we are only interested in relative increase in anionic current, without loss of generality we set $g_{inh}^{PNI} = 1$. The PNI value of $g_{KCC2}$ was set so that $E_{anion}$ = -65 mV according to internal data. This yielded $g_{KCC2}^{PNI} \approx 0.81$ which is normalized to $g_{inh}^{PNI}$. From this and using the relation $I_{anion} = I_{anion}(g_{inh}, g_{KCC2})$ define above, we related the increase in net anionic current to the increase in $g_{inh}$ and $g_{KCC2}$:

$$\Delta I_{anion} = \Delta I_{anion}(\Delta g_{inh}, \Delta g_{KCC2}). \tag{5}$$

The $\Delta$ annotation means the relative change of a quantity normalized to its PNI value. Since the goal of our mathematical model is to explain the relationship between the drug dose and the observed analgesic effect, we also need to infer the relation Effect($\Delta I_{anion}$), the analgesic effect as a function of the increase in anionic current, as well as the relations $\Delta g_{inh}(L838, 417)$ and $\Delta g_{KCC2}(CLP257)$ relating the drug concentrations to their respective effectors activity.

**The maximal effect of L838,417**. We define $\Delta g_{inh}$ as the value of inhibitory conductance for which the net anionic current is maximal given that $g_{KCC2}$ is fixed at PNI value. We assume that this corresponds to the maximal analgesic effect of L838,417 experimentally observed ($Y_{max} \approx 31\%$, Fig. 4e). However, the experimental data exhibits and offset of ≈6% corresponding to an analgesic effect in the absence of drug administration that cannot be explained by the paradigm of an increase in anionic current. We thus subtracted this offset and worked with the corrected value ≈ 25%. The value of $\Delta g_{inh}$ was found to be 1.9 and the associated corresponding relative increase in net anion current was maximal at ≈26%. This yields Effect(26%) $\approx$ 25% as well as $\Delta g_{inh}(10$ mg · kg$^{-1}) = 1.9$.

**The maximal analgesic effect of administrating two drugs**. The maximal analgesic effect of CLP257 experimentally observed was ≈ 25% at 100 mg·kg$^{-1}$ Fig. 4g). As above, we took the offset into account and worked with the corrected value ≈ 19%. Assuming that the analgesia current relation is linear in this range, the value of $\Delta I_{anion}$ corresponding to 19% percent analgesia is $\Delta I_{anion} \approx 19.8\%$. Inverting the $\Delta I_{anion}(\Delta g_{KCC2}, \Delta g_{inh})$ relation (assuming that $g_{inh}$ is fixed at PNI value so that $\Delta g_{inh} = 0$) gave that the increase of $g_{KCC2}$ necessary to obtain an increase of 19.8% in net anionic current is $\Delta g_{KCC2} = 34.5\%$. Interestingly, this is consistent with recovery of KCC2 expression measured by immunostaining (Fig. 5g). Given that the maximal analgesic effect of CLP257 occurs for a dose of 100 mg kg$^{-1}$, we obtained $\Delta g_{KCC2}(100$ mg kg$^{-1}) = 34.5\%$. On the one hand, the joint administration of 100 mg kg$^{-1}$ CLP257 and 10 mg·kg$^{-1}$ L838,417 yielded an experimental analgesic effect of ≈ 48% (Fig. 6f, g) which we corrected accounting for the offset to ≈ 42%. On the other hand, using the values $\Delta g_{inh}(10$ mg kg$^{-1}) = 190\%$ and $\Delta g_{KCC2}(100$ mg kg$^{-1}) = 34.5\%$ derived above, the simulated increase in anionic current due to the joint administration of these doses was $I_{anion}(1.9, 0.345) \approx 67\%$. From this, we could infer that an increase of 67% in anionic current corresponds to an analgesic effect of 42%, that is Effect(67%) = 42%.

**The relationship between analgesia and $\Delta I_{anion}$**. In order to infer the analgesia current relationship for an arbitrary value of $\Delta I_{anion}$, we made the natural assumption that the relation Effect($\Delta I_{anion}$) is linear for low values $\Delta I_{anion}$ but saturating for high values. The simplest relation satisfying this assumption is;

$$\text{Effect}(\Delta I_{anion}) = \frac{\text{Max} \cdot \Delta I_{anion}}{\text{Curhalf} + \Delta I_{anion}}, \tag{6}$$

where Max stands for maximal possible MPA and Curhalf is the value of for which the analgesia is equal to 50%·Max. These constants were determined using the values Effect(67%) = 42% and Effect(26%) = 25% obtained above.

**The relation between [CLP257] and g$_{KCC2}$ and the relationship between [L838,417] and g$_{inh}$**. The last step in building our mathematical model was to infer the relations between doses of individual drug administration and the effect on their respective effector. We inferred the relation between [CLP257] and $\Delta g_{KCC2}$ from the experimental dose response curve of CLP257 administered alone (Fig. 4g) by inverting successively the effect-current relationship and the $g_{KCC2}$ current relationship. Explicitly, we have:

$$\Delta g_{KCC2}([CLP257]) = \Delta I_{anion,g_{KCC2}}^{-1}\left(\text{Effect}^{-1}(\text{Exp}([CLP257]))\right) \tag{7}$$

where Exp([CLP257]) denotes the experimentally observed analgesia of a given dose of CLP257, Effect$^{-1}$ is the inversion of Eq. 6 and $\Delta I_{anion,gKCC2}^{-1}$ is the inverse of Eq. 5 with respect to $\Delta g_{KCC2}$ assuming that $\Delta g_{inh} = 0$. Similarly, the relation between [L838,417] and $\Delta g_{inh}$ was obtained by:

$$\Delta g_{inh}([L838, 417]) = \Delta I_{anion,g_{inh}}^{-1}\left(\text{Effect}^{-1}(\text{Exp}([L838, 417]))\right) \tag{8}$$

where Exp([L838,417]) denotes the experimentally observed analgesia of a given dose of L838,417 (Fig. 4e) and $\Delta I_{anion,g_{inh}}^{-1}$ is the inverse of Eq. 5 with respect to $\Delta g_{inh}$ assuming that $\Delta g_{KCC2} = 0$.

**Mathematical simulation of the analgesic effect of drug administration**. Using the equations defined above (Eqs. 5–8), we generated the surface graphs shown in (Fig. 6d, e) under two distinct assumptions: the common effector assumption and the distinct mechanisms assumption (see schematic representations in (Fig. 6d, e).

1) The common effector assumption. Under this assumption, the simulated the effect of joint L838,417 and CLP257 application is computed by first transforming the dose of CLP257 into a dose of L838,417 yielding an equivalent effect. The dose of L838,417 equivalent to a dose of CLP257 was obtained by first computing the current increase due to the administration of CLP257 and then inverting the L838,417-current relation. Explicitly, we computed:

$$\text{Dose}_{L838,417}^{eq}([CLP257]) = \Delta g_{inh}^{-1}\left\{ \Delta I_{anion,g_{inh}}^{-1}[\Delta I_{anion}(0, \Delta g_{KCC2}([CLP257]))] \right\}. \tag{9}$$

The simulated effect of joint administration of CLP257 and L838,417 under the common effector assumption was then computed as the effect of administrating the following dose of L838,417 alone: $[L838, 417] + \text{Dose}_{L838,417}^{eq}([CLP257])$. Explicitly, this gives:

$$\text{Effect}^{common}([L838, 417], [CLP257]) \\ = \text{Effect}\left\{ \Delta I_{anion}\left[ \Delta g_{inh}\left( [L838, 417] + \text{Dose}_{L838,417}^{eq}([CLP257]), 0 \right) \right] \right\} \tag{10}$$

where the functions used in the right-hand side are defined by the Eqs. 6 and 9. Equation 10 was used to generate the common effector modeled surface in Fig. 6d.

2) The distinct effector assumption. We simulated the effect of joint CLP257 and L838,417 administrations under the assumption each drugs works on a distinct effector namely KCC2 activity for CLP257 and $g_{inh}$ for L838,417. The simulated

effect was computed of a dose of CLP257 and a dose of L838,417 is thus given by:

$$\text{Effect}^{\text{distinct}}([L838, 417], [CLP257]) = \text{Effect}(\Delta I_{\text{anion}}(\Delta g_{\text{inh}}([L838, 417]), \Delta g_{\text{KCC2}}([CLP257]))). \tag{11}$$

Equation 11 was used to generate the distinct effector modeled surface in Fig. 6e. The domains of the surface graphs in Fig. 6d, e were constrained so that the inversions in Eqs. 5–8 are always well defined and that the equivalent dose of L838,417 in Eq. 9 always lies within the range of experimental doses.

The quantitative values in the graphics of Fig. 6d, e obviously depend on the specific function used to model the anionic current. However, we tried several values of $V_{\text{mean}}$ and $V_{\text{eff}}$ as well as replacing the ohmic formalism by the GHK formalism and every version of the model gave a qualitative synergy.

**Reporting summary**. Further information on research design is available in the Nature Research Reporting Summary linked to this article.

## Data Availability

Source data underlying Figs. 1, 2, 3, 4, 5, 6, 7, 8 and Supplementary Figs. 1, 4, 5, 6, 7, 8, 9, 10, 11 are available as a Source Data file. Other data that are necessary to interpret replicate and build upon the methods or findings reported in the article are available upon request.

## Code Availability

Codes and software are available upon request.

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

## Acknowledgements

We really thank Geneviève Brindle who has performed liquid nitrogen assay and Dr Cyril Bories for advice on behavioral testing. We acknowledge Dr Annie Castonguay for her advice and assistance regarding use of CLP257, Sylvain Côté for assistance in graphical design and advices, Manon St-Louis and Johanne Ouellette for technical assistance with electron microscopy. We would like to thank Dr. Jean-Marc Fritschy for the generous gift of his anti-α GABA$_A$R subunit antibodies and Dr Werner Sieghart for his equally generous gift of anti-GABA$_A$ receptor β3 subunit antibody. This work was supported by CIHR grants MOP-12942 to Y.D.K. and MOP-79411 and MOP-136903 to A.R., NSERC grants 109166 to N.D. and 06507 to A.G.G., a Canada Research Chair in Chronic Pain and Related Brain Disorders to Y.D.K., a Chercheur-Boursier Award from the Fonds de recherche en santé du Québec to A.G.G., a People Program Marie Curie Actions grant of the European Union's Seventh Framework Programme (FP7/2007-2013) to F.F. under REA grant agreement n° 318 997 – NEUREN.

## Author contributions

L.-E.L., A.G.G., F.F., K.B., A.R., M.G. and Y.D.K. conceived the experiments. L.-E.L., A.G.G., F.F., K.B., I.P.F., D.B. and I.K. conducted the experiments. L.-E.L., A.G.G., S.L., A.A.G., N.D. and Y.D.K. developed analytical tools or simulation models. L.-E.L., A.G.G., F.F., I.P.F., S.L., A.A.G., D.B., I.K. and Y.D.K. analyzed data. L.-E.L., A.G.G., F.F., N.D., A.R. and Y.D.K. wrote the manuscript. All authors read and edited the manuscript.

## Competing interests

The authors declare no competing interests.
