## [Peer Review File · Nature Communications]

Reviewers' comments:

Reviewer #1 (Remarks to the Author):

In this study, Lorenzo and coll. claim that the combination of the GABAAR $\alpha 2,3$ subunit positive modulator L838,417 with the KCC2 stabilizer CLP257 has a synergistic analgesic action on neuropathic pain in a rat model of peripheral nerve injury (PNI).

The authors first report a decrease in the density of inhibitory synapses in the spinal cord immunoreactive for the presynaptic VGAT and GAD65-67 makers as well as for the postsynaptic scaffolding molecule gephyrin, and the GABAAR $\alpha 2$ - $\alpha 3$ and $\beta 3$ subunits. They further show that the GABAAR $\alpha 2,3$ and $\beta 2$ subunits are upregulated at the remaining inhibitory synapses leading to a shift in the expression of GABAAR subunits towards $\alpha 2$ subunit. This change in GABAAR expression is associated with a downregulation of the main Cl⁻ extruder KCC2. They then studied the beneficial effect of a separate or combined administration of the partial GABAAR $\alpha 2$, $\alpha 3$ and $\alpha 5$ subunit agonist L838,417 and of the intracellular Cl⁻ regulator CLP257 showing the synergistic effect of the co-administration and the beneficial impact on PNI. Facilitating the extrusion of Cl⁻ while activating GABAAR is clever since this may prevent the depolarizing shift in EGABA induced by a prolonged activation of the receptor and the subsequent elevation in intracellular chloride level. The conclusion of this work is reminiscent of a study from Kevin Staley's group suggesting the beneficial effect of combining a blocker of the chloride importer NKCC1 bumetanide with a barbiturate for the treatment of neonatal seizures (Dzala et al., *Ann. Neurol* 2008). The present work further shows that targeting simultaneously GABAAR and KCC2 has a strong potential for the treatment of neuropathic pain and possible other pathologies presenting a dysregulation of chloride homeostasis such as epilepsy and neuropsychiatric disorders.

The present study is of high quality. The experimental design and the statistical analysis are appropriate, and the results are convincing. The manuscript is well written and the main claims may permit the development of promising therapeutic strategy for several pathologies with excitation/inhibition imbalance. However, further evidences are needed to address the specificity of the observed effects in order to strengthen the author's conclusions and reach the standards of Nature Communications publications.

The main issue here concerns the debate around the effect of CLP257 as a membrane stabilizer of KCC2 (Gagnon et al. *Nature Medicine* 2013, 2017) vs an activator of GABAAR (Cardarelli et al. *Nature Medicine* 2017). Cardarelli et al. (2017) saw a potentiation of muscimol-evoked currents by CLP257 at concentrations of 10–30 μ M. In their reply, Gagnon et al. (2017) mention CLP257 is used at high doses (25 μ M) in slices. This raises the question of the concentration of CLP257 in the dorsal spinal cord of the injected animals. This is not mentioned in the manuscript. The animals were injected with 40 mg/kg of CLP257. What is the concentration in the spinal cord? The authors need to determine the concentration of CLP257 in the spinal cord in vivo (by HPLC for instance). They then need to prove that this concentration is compatible with an effect on KCC2 stability only (1 μ M) and not on GABAAR activity (≥ 10 μ M). This could be tested in slices of dorsal spinal cord. In the case the detected level of CLP257 could activate GABAAR, a lower and more selective concentration of CLP257 should be tested on PNI.

Another way to counteract the depolarization of EGABA subsequent to increased Cl⁻ influx upon GABAAR benzodiazepine activation will be to block NKCC1-mediated Cl⁻ import with bumetanide. Can the authors test separately or in combination L838,417 and bumetanide (+/- CLP257?) on PNI in vitro and/or in vivo? A synergistic action of the drugs will support their conclusion.

Reviewer #2 (Remarks to the Author):

The manuscript by Etienne Lorenzo reports a very interesting new mechanism to account for failure of benzodiazepine ligands to elicit analgesia and a strategy to overcome these limitations. By providing experimental data on physiology, pharmacology and expression analyses, as well as simulation, the manuscript covers broad ground. The study can have major therapeutic relevance and is beautifully designed and executed, in general. However, there are a few minor points and one major point that deserve to be addressed:

Major point:

The manuscript consistently reports on 'analgesia' and indeed enhancing analgesia is the major

appeal of the study. However, 'only' mechanical thresholds have been measured; mechanical allodynia is not the same as pain and reducing mechanical allodynia does not necessarily amount to analgesia. The data would be convincing if non-reflexive behavioral analyses, particularly addressing pain by targeting affective components as well as sensory behavior, were to be included. This is crucial to make the point that the mechanism described here has some actual relevance to pain therapy.

Minor points:

1. The reviewer missed the red box in Fig. 1a, as stated in figure legends, to indicate the time point at which quantification of synaptic markers was conducted, and could not find this information elsewhere in the text. It is also not clear when the behavioural experiments treated with drugs and electrophysiological recording were performed after injury. Therefore, it is difficult to know whether the quantification is done early or late after injury, or at different time points, and whether the observed reduction of number of inhibitory synapses and $\alpha 1$ to $\alpha 2$ GABA AR subunit switch change along the time. The same group has shown that CCI induced a transient loss of inhibitory terminals (Lorenzo et al., 2014). If the changes after PNI reported in this manuscript are also transient, then drug treatment should be ideally tested at different time points. Can the authors clarify this? Moreover, a discussion on the time window to treat neuropathic pain by combined targeting of the appropriated GABAAR subtypes and restoring Cl-homeostasis is needed.

2. It would be interesting to know in which population of spinal neurons show an increase in staining for the $\alpha 2$ and $\alpha 3$ subunits. Are these types of neurons which have been implicated in tactile allodynia after nerve injury, e.g. PKCY –positive neurons? Is KCC2 downregulation observed in the same neurons which show increase in GABAAR? It may be unlikely, but it is still possible that there is increased inhibition onto inhibitory interneurons due to increased GABAAR while there is loss of inhibition induced by KCC2 downregulation in excitatory neurons.

3. It is nice that in Fig.7 the authors demonstrated the CLP257 could reverse the amplification of the depression of eIPSCs in PNI. However, whether low dose L838,417 affects eIPSCs did not become clear to the reviewer. Is it enhanced and could it thereby contribute to the analgesic effect?

Reference:

Lorenzo, L.E., Magnussen, C., Bailey, A.L., St Louis, M., De Koninck, Y., and Ribeiro-da-Silva, A. (2014). Spatial and temporal pattern of changes in the number of GAD65-immunoreactive inhibitory terminals in the rat superficial dorsal horn following peripheral nerve injury. *Molecular pain* 10, 12.

Reviewer #3 (Remarks to the Author):

Using immunohistochemistry PCR, some electrophysiology and modeling the authors demonstrate multidirectional changes in expression of several GABAA subunits and KCC2 transporter in the superficial dorsal horn following peripheral nerve injury. They also show analgesic efficacy of $\alpha 2,3$ GABAA receptor targeting benzodiazepine L838,417 and a KCC2 modulator (both as monotherapies and in combination). The data are mostly solid and interesting and will be of interest within the field. Yet, in the opinion of this reviewer, the data do not add up into an entirely consistent mechanism. In addition, given the wealth of other studies on the topic and significant discrepancies in published data, this current study does not bring additional clarity to the field.

Major

1. The study relies heavily on the immunostaining and confocal imaging; the authors analyzed

these data with various computer algorithms (which are not always well explained), which are clever but nonetheless do not overcome major issues associated with the method: i) antibody specificity and non-specific binding and ii) insufficient resolution of optical microscopy (for the type of question the authors are asking). It would be better to back up major conclusions with something else, like superresolution microscopy or EM.

2. The authors seem to treat dorsal horn neurons as a single population, i.e. everything that is 'postsynaptic'. From the way the manuscript is written it appears that they consider SDH neurons all to be 2nd order nociceptive projection neurons. Indeed, all 'postsynaptic' changes reported in the manuscript are interpreted in terms of direct effect on nociceptive transmission. Yet, this is a rather striking oversimplification as there is a number of different projection neurons, interneurons etc. (see e.g. chapter 5 in Wall & Melzack's Textbook of Pain). Therefore, changes in GABAA (or any other) receptors may produce completely different outcomes, depending on which cell populations are affected. This is not taken into any consideration in this manuscript, which is a pity.

3. The study does not probe mechanism of observed changes in GABAA subunit and KCC2 abundance at postsynaptic sites. They provide some data that BDNF and Ca²⁺ are required but there is no clarity as to what produces Ca²⁺ signals downstream of (or in parallel with) TrkB activation or how these Ca²⁺ signals are translated in to regulation of channels and transporters' abundance.

4. The authors play down possible contribution of afferent fibers into GABA-mediated analgesia citing some papers in favor of their point of view. Yet they live out other studies that convincingly show that peripheral GABAA receptors do play a significant role (e.g. papers from Todorovic's lab and some others).

Specific concerns

5. The authors used "the territory defined by this IB4 labeling loss to delineate the core of the central projection of injured afferents". IB4 labels non-peptidergic nociceptors synapsing predominantly in the inner part of lamina II. Yet peripheral injury was applied to the entire sciatic nerve, hence, all the other afferent fiber types, which synapse in different laminae were also affected. On a related note – the authors assume that the loss of the IB4 staining in the dorsal horn is synonymous to the loss of IB4-positive terminals. But it is just as well (or even more likely) could be due to reduced staining for whatever reason, not due to the loss of the terminals themselves. The authors also state that "The loss in IB4-labeled non-peptidergic terminals was not associated with a loss in Calcitonin Gene Related Peptide CGRP-immunoreactive peptidergic terminals, making the non-peptidergic afferents a better marker to define a lesion after PNI". But why peptidergic (or other types, e.g. Adeltas etc.) were not affected – these were also injured. Also, as noted before, different fibers terminate in different DH laminae, so using only IB4 will leave out other affected areas. This approach appears very arbitrary. Also the 'changes' in pre-/post-synaptic densities (Fig. 1h, i) are modest. Altogether, the major findings of Fig. 1 appear somewhat questionable and over-interpreted.

6. ECl measurements in the whole-cell patch clamp mode (Fig. 3): these experiments are not well explained and appear to be misguided. I didn't find the extracellular solution composition but assuming it is in the range of 150 mM, in combination with 25 mM Cl⁻ in the pipette, Nernst equation gives ECl around -45 mV, which is close to what was measured (-44.1). But what is the point of such a measurement? Surely ECl will be whatever you put in to your solutions! I assume the physiological ECl in SDH neurons is much more negative than -45 mV. If you want to measure realistic ECl and its changes, you would need to do gramicidin perforated patch recordings.

7. It is somewhat surprising that 3 hrs of BDNF treatment in slice condition was sufficient for the GABAA subunit switch.

8. What are the conclusions from EGTA/BAPTA experiments? Both are Ca²⁺ chelators, are you implying some local Ca²⁺ microdomains to be involved – what are these? These results are just left alone without any follow-up experiments or even a discussion.

9. Behavioral experiments with L838,417: "Anti-hyperalgesia by $\alpha 2$ GABAAR has been shown to preferentially occur via a spinal site of action and not through supraspinal sites." While supraspinal sites may be not involved, but peripheral sites do; see Obradovic et al. 2015 Anesthesiology 123:654-67. Alpha2 is expressed abundantly in peripheral fibers and provides analgesia in neuropathic pain models. Hence, the results presented in Fig. 4 are uninterpretable.

10. KCC2 staining (Fig. 5): there seem to be high background, while in selected examples (white boxes) there membranous staining is discernable, the other neuronal cell bodies appear variably; the background level seems different between different conditions, not only the membrane staining – all this lowers the enthusiasm about the robustness of the analysis.

Minor.

11. Page 3, end of 2nd paragraph: "These results indicate a loss of inhibitory synapse scaffolding protein expression..." – how did you arrive at this conclusion?

12. Page 4, end of 2nd paragraph: "... a function blocking anti-TrkB antibody..." – can you explain 'function-blocking' and provide a reference?

13. Please explain this para: "The antibodies against the α -GABAAR subtypes used in this study were bought from Synaptic Systems (Sysy) and raised in rabbit for the first set of experiments. For the second set of experiments, they were generously provided by Dr. Jean-Marc Fritschy and raised in guinea pig" – why did you use different antibodies for different experiments? Where is the first set of experiments starts and second begins, are there only two sets of experiments in this paper?

Reviewer #1 (Remarks to the Author): *In this study, Lorenzo and coll. claim that the combination of the GABAAR $\alpha_{2,3}$ subunit positive modulator L838,417 with the KCC2 stabilizer CLP257 has a synergistic analgesic action on neuropathic pain in a rat model of peripheral nerve injury (PNI). The authors first report a decrease in the density of inhibitory synapses in the spinal cord immunoreactive for the presynaptic VGAT and GAD65-67 makers as well as for the postsynaptic scaffolding molecule gephyrin, and the GABAAR α_{2-3} and β_3 subunits. They further show that the GABAAR $\alpha_{2,3}$ and β_2 subunits are upregulated at the remaining inhibitory synapses leading to a shift in the expression of GABAAR subunits towards α_2 subunit. This change in GABAAR expression is associated with a downregulation of the main Cl⁻ extruder KCC2. They then studied the beneficial effect of a separate or combined administration of the partial GABAAR α_2 , α_3 and α_5 subunit agonist L838,417 and of the intracellular Cl⁻ regulator CLP257 showing the synergistic effect of the co-administration and the beneficial impact on PNI. Facilitating the extrusion of Cl⁻ while activating GABAAR is clever since this may prevent the depolarizing shift in EGABA induced by a prolonged activation of the receptor and the subsequent elevation in intracellular chloride level. The conclusion of this work is reminiscent of a study from Kevin Staley's group suggesting the beneficial effect of combining a blocker of the chloride importer NKCC1 bumetanide with a barbiturate for the treatment of neonatal seizures (Dzala et al., Ann. Neurol 2008). The present work further shows that targeting simultaneously GABAAR and KCC2 has a strong potential for the treatment of neuropathic pain and possible other pathologies presenting a dysregulation of chloride homeostasis such as epilepsy and neuropsychiatric disorders.*

The present study is of high quality. The experimental design and the statistical analysis are appropriate, and the results are convincing. The manuscript is well written and the main claims may permit the development of promising therapeutic strategy for several pathologies with excitation/inhibition imbalance. However, further evidences are needed to address the specificity of the observed effects in order to strengthen the author's conclusions and reach the standards of Nature Communications publications.

Comment: We thank reviewer#1 for his/her enthusiastic comments regarding this study.

The main issue here concerns the debate around the effect of CLP257 as a membrane stabilizer of KCC2 (Gagnon et al. Nature Medicine 2013, 2017) vs an activator of GABAAR (Cardarelli et al. Nature Medicine 2017). Cardarelli et al. (2017) saw a potentiation of muscimol-evoked currents by CLP257 at concentrations of 10–30 μ M. In their reply, Gagnon et al. (2017) mention CLP257 is used at high doses (25 μ M) in slices. This raises the question of the concentration of CLP257 in the dorsal spinal cord of the injected animals. This is not mentioned in the manuscript. The animals were injected with 40 mg/kg of CLP257. What is the concentration in the spinal cord? The authors need to determine the concentration of CLP257 in the spinal cord in vivo (by HPLC for instance). They then need to prove that this concentration is compatible with an effect on KCC2 stability only (1 μ M) and not on GABAAR activity (≥ 10 μ M). This could be tested in slices of dorsal spinal cord. In the case the detected level of CLP257 could activate GABAAR, a lower and more selective concentration of CLP257 should be tested on PNI.

Response: The Reviewer's point is important and we take the opportunity to clarify this part to avoid misunderstandings and confusion. As we explained in our reply to Cardarelli *et al.* 2017¹, slice or tissue conditions reduce the functional concentration of CLP257 at the action sites (*e.g.*, degradation, trapping, protein-binding). In slices, 25 μ M of CLP257 yields much lower effective concentrations. The observation reported by Cardarelli *et al.* (2017)² is potentially relevant for neuronal cultures experiments; they used 30 μ M under these conditions, which was an order of magnitude higher than the EC₅₀ we reported in cell cultures (Gagnon *et al.*, Nature Medicine 2013)³. The pharmacokinetics data we report in the original paper by Gagnon *et al.* (2013)³ shows that the plasma concentration of CLP257 declines very rapidly, with a terminal half-life (t_{1/2}) of <15 min. The expected dose at the site of action in the spinal cord is thus a small fraction of the systemic administration dose. To confirm this, we first determined, by HPLC, the concentration of CLP257 in the CNS at different time points after 100 mg.kg⁻¹ i.p. injections *in vivo* as suggested by the reviewer. We found that the concentration of CLP257 in brain homogenates was less than 0.2 μ M when measured 30 min after i.p. injection and dropped to less than 0.05 μ M after 1h. Thus, the effective concentration at the target site in the spinal cord was in the sub-micromolar range, within the range of selective activation of KCC2 (Gagnon *et al.*; 2013, 2017)^{1,3}. These results are now shown in Supplementary Fig. 7 and outlined on page 5 of the results section, page 7 of Discussion and page 17 of the Methods section.

Regarding the experiments in slices, the effect of CLP257 on the collapse of IPSC amplitude upon repetitive input cannot be ascribed to an effect on GABA_A channels because, if anything, this would amplify the collapse as we have shown with benzodiazepine site agonist L838, 417 (Fig. 7). We had used

a dose of 100 μ M of CLP257 in the original experiment in Fig.7 because we tested the responses to repetitive stimulation in the same cell before and after the drug. We used a high dose to measure an effect within 15 minutes of drug perfusion. 100 μ M still remains a modest dose given how fast CLP257 is metabolized (Fig. 4 in Gagnon *et al.* 2013³ and new Supplementary Fig. 7 in this paper). Nevertheless, to complement these results and dispel any doubt on dose, we repeated these experiments at a much lower dose of 5 μ M CLP257, which is below any potential effect on GABA_A even in neuronal cultures. Here we compared results obtained in different slices incubated at 5 μ M for 1h to ensure we reached steady state. Bath application of CLP257 at 5 μ M prevented the collapse of Cl⁻ mediated IPSCs similarly to what was observed with a short pulse of CLP257 at 100 μ M. These new results were added to Fig. 7 and outlined in Results p. 6 (*L838,417 and CLP257 act synergistically to produce more analgesia*).

Another way to counteract the depolarization of EGABA subsequent to increased Cl⁻ influx upon GABAAR benzodiazepine activation will be to block NKCC1-mediated Cl⁻ import with bumetanide. Can the authors test separately or in combination L838,417 and bumetanide (+/- CLP257?) on PNI in vitro and/or in vivo? A synergistic action of the drugs will support their conclusion.

Response: Blocking Cl⁻ import via NKCC1 is indeed another means to lower intracellular [Cl⁻]. Dzhalala *et al.*⁴ used this in immature tissue when NKCC1 is dominant over KCC2. In mature tissue, as in our setting, there is no evidence of significant membrane NKCC1 in dorsal horn neurons and there is no compelling/robust evidence in the literature of membrane expression mature neurons (except DRG and some other exceptions). NKCC1 is however highly expressed in the primary afferents, where KCC2 is absent (Coull *et al.*, *Nature* 2003⁵; Lorenzo *et al.*, *J. Neurosci.* 2014⁶). Bumetanide could thus not be used to counteract the Cl⁻ accumulation in dorsal horn neurons. Another advantage of targeting KCC2 is that, contrary to other cation chloride cotransporter, it is CNS specific, making it a more specific target. Targeting dorsal horn neuron Cl⁻ homeostasis is important in this context, as the anti-hyperalgesic effect for α 2-GABA_AR has been shown to involve dorsal horn neurons, not primary afferents nor supraspinal sites (Witschi *et al.*, *J. Neurosci.* 2011⁷ & Paul *et al.*, *Neuropsychopharmacology* 2013⁸). We have added statements to this effect in Discussion on page 7.

Reviewer #2 (Remarks to the Author):

The manuscript by Etienne Lorenzo reports a very interesting new mechanism to account for failure of benzodiazepine ligands to elicit analgesia and a strategy to overcome these limitations. By providing experimental data on physiology, pharmacology and expression analyses, as well as simulation, the manuscript covers broad ground. The study can have major therapeutic relevance and is beautifully designed and executed, in general. However, there are a few minor points and one major point that deserve to be addressed:

Comment: We thank reviewer#2 for his/her enthusiastic appreciation of the manuscript.

Major point:

The manuscript consistently reports on 'analgesia' and indeed enhancing analgesia is the major appeal of the study. However, 'only' mechanical thresholds have been measured; mechanical allodynia is not the same as pain and reducing mechanical allodynia does not necessarily amount to analgesia. The data would be convincing if non-reflexive behavioral analyses, particularly addressing pain by targeting affective components as well as sensory behavior, were to be included. This is crucial to make the point that the mechanism described here has some actual relevance to pain therapy.

Response: It is true that we have exploited the most broadly used Up-Down Von Frey testing method of nociceptive withdrawal reflex as we sought a quantitative index of mechanical allodynia (Chaplan *et al.*, *J. Neurosci. Methods*, 1994)⁹. To complement our results with a more holistic index of pain behavior, as suggested by the reviewer, we used a semi-quantitative assay based on a more complex behavioral response (Colburn, R.W. *et al.*, *Neuron* 2007)¹⁰. Briefly, in response to local application of liquid nitrogen to the hind foot the behavioral response was quantified as a compound index based on prolonged lifting, licking, and shaking of the hindpaw. These additional results confirm synergistic analgesic interaction between low doses of L838,417 and CLP257. The new results are now presented in Supplementary Fig. 8. Beyond this more complex behavioral assessment, we did not pursue with a conditional place preference test because the latter does not allow quantification of the degree of analgesia (one only obtains and all-or-none, binary type of result).

Minor points:

1. The reviewer missed the red box in Fig. 1a, as stated in figure legends, to indicate the time point at which quantification of synaptic markers was conducted, and could not find this information elsewhere in the text. It is also not clear when the behavioural experiments treated with drugs and electrophysiological recording were performed after injury. Therefore, it is difficult to know whether the quantification is done early or late after injury, or at different time points, and whether the observed reduction of number of inhibitory synapses and $\alpha 1$ to $\alpha 2$ GABAAR subunit switch change along the time. The same group has shown that CCI induced a transient loss of inhibitory terminals (Lorenzo et al., 2014). If the changes after PNI reported in this manuscript are also transient, then drug treatment should be ideally tested at different time points. Can the authors clarify this? Moreover, a discussion on the time window to treat neuropathic pain by combined targeting of the appropriated GABAAR subtypes and restoring Cl⁻ homeostasis is needed.

Response: We apologize for the oversight with the box. The leftover reference to the red box has been removed. We also modified the 1st paragraph in the results section (p. 2) to clarify the time window used: the experiments were all run between day 14 and 28 post sham- or PNI-injury. These changes were added to p. 11 and legend of Fig. 1.

Response: The issue of the time course of plastic changes (GABA_A and KCC2) is important, but we think it falls outside of the scope of the present study. This is because the objective of this study was to test the value of rescuing KCC2 to effectively restore the analgesic efficacy of positive modulation of GABA_A receptors. If the deficit in chloride homeostasis subsides, then, indeed the treatment would be of limited value. We have however shown in a recent study that, in a model of sustained hyperalgesia for months, three months after injury changes in the microglia-BDNF-TrkB-KCC2 signaling remain, indicating sustained downregulation of KCC2 well beyond the initial post-injury phase (Echeverry et al., *Pain* 2017)¹¹. Thus, the strategy presented here can be relevant to long-term neuropathic pain syndromes. We have added a Discussion point to this effect at the end of the text as requested by the reviewer.

2. It would be interesting to know in which population of spinal neurons show an increase in staining for the $\alpha 2$ and $\alpha 3$ subunits. Are these types of neurons which have been implicated in tactile allodynia after nerve injury, e.g. PKCY-positive neurons? Is KCC2 downregulation observed in the same neurons which show increase in GABAAR? It may be unlikely, but it is still possible that there is increased inhibition onto inhibitory interneurons due to increased GABAAR while there is loss of inhibition induced by KCC2 downregulation in excitatory neurons.

Response: While the question of which cell types are affected within the dorsal horn network is, by itself of value, it remains always very difficult to predict what the net outcome of individual changes will be on circuit function and output. Thus, to a certain extent, the question falls outside the scope of the present study. We have chosen a more global assessment of changes, and, from this, focus on measuring the net effect of drug of the output of the circuit; in the present case, spinal ascending output or behavioral sensitization. Nevertheless, to address the question the reviewer is raising at the level of excitatory vs. inhibitory components of the dorsal horn circuit; we performed additional experiments, using an RNAscope multiplex assay to measure changes in *Gabra2* mRNA in inhibitory vs. excitatory neurons. We thus performed an analysis of *Gabra2* mRNA in inhibitory (*SLC32A1*; VGAT+) and excitatory (*SLC17A6*; VGLUT2+) neurons (new Fig 2f, Supplementary Fig. 2,3, Results p. 3, Methods p. 15). We found that *Gabra2* mRNA expression was upregulated in both inhibitory and excitatory dorsal horn neurons after nerve injury (new graphic in Fig. 2f). To date, across all our previous studies, we have not found any evidence of an overt differential change in KCC2 among cell types in the dorsal horn following nerve injury. Thus, because changes in $\alpha 2$ and KCC2 appear to occur in both inhibitory and excitatory neurons, the only way to really assess the net functional outcome is to measure the global output response (behavioral; nociceptive withdrawal) as we have conducted here. *In fine*, rather than identifying the specific cell subtypes involved, the impact of our work lies in identifying two major protein targets (*i.e.*, receptor subunit and transporter isoform) for pharmacological intervention and demonstration of the synergistic value of targeting both together, given that they represent the two main actors involved in the feedback loop determining Cl⁻ homeostasis.

3. It is nice that in Fig.7 the authors demonstrated the CLP257 could reverse the amplification of the depression of eIPSCs in PNI. However, whether low dose L838,417 affects eIPSCs did not become clear to the reviewer. Is it enhanced and could it thereby contribute to the analgesic effect?

Response: Indeed, L838,417 enhances the amplitude of the initial eIPSCs (Fig. 7b). In that sense, it may indeed explain the analgesic effect of L838,417. However, this potentiation, in turn precipitates the collapse in Cl⁻ gradient during a barrage of eIPSCs (Fig. 7a,c,d) in slices from nerve injured animals, consistent with the impact of enhancing Cl⁻ loading through GABA_A receptors. Note that CLP257 did not significantly affect the amplitude nor the kinetics of the L838,417-modulated eIPSCs at low frequency (Fig. 7b), indicating that the effect of CLP257 on the collapse of repetitive eIPSCs is independent of any effect on GABA_A receptors. That means that the total charge carried by each eIPSC is the same after CLP257. Thus, the prevention of the collapse in eIPSCs during repetitive input can only be explained by the fact the CLP257 counteracts intracellular Cl⁻ accumulation (by enhancing Cl⁻ extrusion). We have clarified this in the text on page 6.

Reviewer #3 (Remarks to the Author):

Using immunohistochemistry PCR, some electrophysiology and modeling the authors demonstrate multidirectional changes in expression of several GABAA subunits and KCC2 transporter in the superficial dorsal horn following peripheral nerve injury. They also show analgesic efficacy of α 2,3 GABAA receptor targeting benzodiazepine L838,417 and a KCC2 modulator (both as monotherapies and in combination). The data are mostly solid and interesting and will be of interest within the field. Yet, in the opinion of this reviewer, the data do not add up into an entirely consistent mechanism. In addition, given the wealth of other studies on the topic and significant discrepancies in published data, this current study does not bring additional clarity to the field.

Major

1. *The study relies heavily on the immunostaining and confocal imaging; the authors analyzed these data with various computer algorithms (which are not always well explained), which are clever but nonetheless do not overcome major issues associated with the method: i) antibody specificity and non-specific binding and ii) insufficient resolution of optical microscopy (for the type of question the authors are asking). It would be better to back up major conclusions with something else, like super-resolution microscopy or EM.*

Reponses:

i) First, regarding the issue of antibody specificity, we took special care to address this point and it may not have been clear enough in the previous version of the manuscript. To clarify this, we added a table listing all the antibodies used in this study, species they were raised in, sequence they were generated against, dilution, source, references, and the conditions used to demonstrate specificity. It should be noted that the same quantitative results were obtained for the key anti- α GABA_A subunits (α 1, α 2, α 3) with more than one antibody (*e.g.*, Fig 2c vs. Supplementary Fig 1). This shows consistency across different antibodies. Moreover, the different antibodies used for each subunit of the GABA_A receptors were raised against different sequences and in different species, further validates the results and addresses the issue of antibody and binding specificity. Finally, the new data presented in Fig. 2f (and supplementary Fig. 2,3; see response to Rev. 2, point 2 and Rev. 3 point 3) further confirms the increase in α 2 mRNA across both excitatory and inhibitory dorsal neurons.

ii) For the issue of resolution, the reviewer raises an important issue that needs proper attention. While super-resolution approaches have been very instrumental to decipher biological structures at the nanoscale level in various reduced preparations (Huang *et al.* *Cell*, 2010; Dani *et al.*, *Neuron* 2010 & Durand *et al.*, *Nat. comm.*, 2018)¹²⁻¹⁴, current super-resolution techniques still do not effectively provide significant improvements in resolution in thick tissue over large fields of views (Godin *et al.*, *Biophysical journal* 2014)¹⁵. This is because improvements in spatial resolution always come in pair with sacrificing temporal resolution – hence ability to scan large fields of view– and fluorophore flexibility (*i.e.*, very long scan times causing photobleaching limitations). In addition, current super-resolution approaches provide limited multicolor imaging capability necessary for the type of analysis we have performed here. And these limitations are compounded for tissue-based quantification. Thus, currently, leading super-

resolution optical techniques do not provide the required throughput and flexibility to realistically achieve the type of large-scale analysis that we conducted in this study.

Beyond the fact that super-resolution would not provide the throughput needed to conduct the quantification we achieved, it is our contention that super-resolution was not needed to address the question we asked. Indeed, to estimate the intensity of well separated clusters as in the system we studied (*i.e.*, the average gephyrin cluster density was 10.5 ± 0.7 and 14.9 ± 0.7 (mean \pm S.E.M. for $N = 14$ rats) per $100 \mu\text{m}^2$ in laminae I and II, respectively), conventional fluorescence microscopy is sufficient and allows an unequaled flexibility to investigate a large number of synapses in single images. Fluorescence intensity is linearly proportional to emitter concentration over a large dynamic scale. This is what allows quantitation of receptor distributions and oligomerization states within well identified clusters over wide fields of view, even with conventional microscopy (Sugiyama *et al.*, 2005¹⁶, Godin *et al.*, 2011, 2015^{17,18} and Sergeev *et al.*, 2012¹⁹, Patrizio *et al.*, 2016²⁰).

To validate that enhanced resolution would not provide significant additional information, we still performed additional measurements using a hybrid solution that allows similar flexibility and field of views as confocal microscopy but that was shown to improve the resolution. A Zeiss LSM 880 microscope equipped with an Airyscan detection unit was used to generate images with higher resolution than with conventional confocal microscopy. This LSM modality uses a multi-array detector, deconvolution and the pixel-reassignment principle to enhance spatial resolution and signal-to-noise-ratio (Korobchevskaya *et al.*, *Photonics* 2017)²¹. Using a high numerical aperture oil-immersion objective (Plan-Apochromat 63x/1.4 Oil DIC), we quantified the resolution improvement of our multi-array detector compared to confocal microscopy using 100 nm diameter fluorescent nanospheres (Abberior TS 100 nm). The full width half maximum (FWHM) of the bead intensity profiles showed a ~ 1.7 improvement compared to conventional confocal detection (Supplementary Fig. 6). We then conducted quantification of the $\alpha 2$ GABA_AR in gephyrin masks in the same fields of view, within the same samples, using both modalities: conventional confocal and Airyscan detections. The results show that the intensities integrated between the two modalities correlate, indicating that the results obtained using the confocal modality are valid to draw the conclusions we drew.

2. The authors seem to treat dorsal horn neurons as a single population, i.e. everything that is 'postsynaptic'. From the way the manuscript is written it appears that they consider SDH neurons all to be 2nd order nociceptive projection neurons. Indeed, all 'postsynaptic' changes reported in the manuscript are interpreted in terms of direct effect on nociceptive transmission. Yet, this is a rather striking oversimplification as there is a number of different projection neurons, interneurons etc. (see e.g. chapter 5 in Wall & Melzack's Textbook of Pain). Therefore, changes in GABAA (or any other) receptors may produce completely different outcomes, depending on which cell populations are affected. This is not taken into any consideration in this manuscript, which is a pity.

Response: This is the same issue as that raised by reviewer #2 point #2. We redirect the reviewer to our response and description of additional data above.

3. The study does not probe mechanism of observed changes in GABAA subunit and KCC2 abundance at postsynaptic sites. They provide some data that BDNF and Ca2+ are required but there is no clarity as to what produces Ca2+ signals downstream of (or in parallel with) TrkB activation or how these Ca2+ signals are translated in to regulation of channels and transporters' abundance.

Response: The reviewer is right that the present study does not attempt to delve into the mechanisms of regulation of the GABA_AR subunit switch, nor the KCC2 downregulation. This is clearly not the objective of the present study. The data provided on Ca²⁺ requirement for the BDNF effect was only a technical methodological issue to clarify potential discrepancies in recording conditions between laboratories, *i.e.* recordings performed with strong and fast Ca²⁺ buffering conditions (intrapipette BAPTA) would have missed revealing the commonality of effect of nerve injury and BDNF on mIPSC properties. We have now moved the buffering issue to supplementary Fig. 4 not to side-track the reader from the focus of the study.

4. The authors play down possible contribution of afferent fibers into GABA-mediated analgesia citing some papers in favor of their point of view. Yet they live out other studies that convincingly show that peripheral GABA_A receptors do play a significant role (e.g. papers from Todorovic's lab and some others).

Response: It was not our intention to down play the role of GABA_A receptors on primary afferents. It is indeed the basis of presynaptic inhibition on afferent endings. We have ourselves conducted a detailed study of the expression of GABA_A receptor subtypes on small diameter primary afferents (Lorenzo et al., *J.Neurosci.* 2014)⁶. However, the results of the current study indicate that the $\alpha 2$ subunit on dorsal horn neurons is a key target for analgesia in neuropathic pain because of a subunit switch that occurs on dorsal horn neurons after peripheral nerve injury. This is consistent with the original data from H.U. Zeilhofer's group who has shown that selective deletion of the $\alpha 2$ in small diameter primary afferents had no effect on benzodiazepine-induced analgesia in the nerve injury model of neuropathic pain (Witschi et al., *J.Neurosci.* 2011)⁷. In contrast, total deletion of $\alpha 2$ at the spinal level significantly attenuated benzodiazepine-induced analgesia (Paul et al *Neuropsychopharmacology*, 2014)⁸, indicating that $\alpha 2$ on dorsal horn neurons is a prime target for analgesia after nerve injury.

Specific concerns

5. The authors used "the territory defined by this IB4 labeling loss to delineate the core of the central projection of injured afferents". IB4 labels non-peptidergic nociceptors synapsing predominantly in the inner part of lamina II. Yet peripheral injury was applied to the entire sciatic nerve, hence, all the other afferent fiber types, which synapse in different laminae were also affected. On a related note – the authors assume that the loss of the IB4 staining in the dorsal horn is synonymous to the loss of IB4-positive terminals. But it is just as well (or even more likely) could be due to reduced staining for whatever reason, not due to the loss of the terminals themselves. The authors also state that "The loss in IB4-labeled non-peptidergic terminals was not associated with a loss in Calcitonin Gene Related Peptide CGRP-immunoreactive peptidergic terminals, making the non-peptidergic afferents a better marker to define a lesion after PNI". But why peptidergic (or other types, e.g. Adelta etc.) were not affected – these were also injured. Also, as noted before, different fibers terminate in different DH laminae, so using only IB4 will leave out other affected areas. This approach appears very arbitrary. Also the 'changes' in pre-/post-synaptic densities (Fig. 1h, i) are modest. Altogether, the major findings of Fig. 1 appear somewhat questionable and over-interpreted.

Response: The reference to the loss of IB4 we use is based on several of our previous findings and that of others to help define a reproducible site of quantification. For example, in one our previous studies (Lorenzo et al., *Mol. Pain* 2014)²² we showed that the changes in densities of GAD65 boutons mapped to the region of loss of IB4 labeling. Similarly, Beggs et al., *Brain. Behav. Immun.* 2007²³ conducted a detailed analysis of the area of fiber termination and microglia activation in the dorsal horn after peripheral nerve injury and showed that the region of activated microglia mapped to the region of loss of IB4 labeling. Thus, the focus on the core of the IB4 loss is to provide a means to standardize the quantification across lesions in different animals. The reviewer is right that we cannot equate loss of IB4 staining to loss of fibers. This is not the intent, but rather define the site of projection within the dorsal horn where the change is maximum. In the present study, we show that the change in KCC2 also maps to the region of loss of IB4 labeling intensity. This is shown on the transects in Fig. 3c,d. This demonstrates the relevance of the region chosen for quantification. To respond to the reviewer's question regarding deeper laminae, we now show in Supplementary Fig. 4 a plot of the KCC2 staining with respect to IB4 staining to show how the changes map to each other, objectivizing the criteria we used to delineate the region for quantification (Supplementary Fig. 4). We also do not pretend that the plastic changes are strictly restricted to the region of IB4 loss, but rather that centering on that region provides a reproducible registration reference across animals. It is true in fact that the plastic changes extend beyond the region of IB4 loss as the values return to baseline extending beyond (in the form of a transition zone). Consistent with this view, we had originally performed a separate quantification of changes specifically in lamina I just above the region of IB4 loss. The data were not included in the manuscript to simplify the presentation, but we now added them as supplemental material (Supplementary Fig. 4 and Supplementary Fig. 5). The findings are the same as those observed in lamina II.

As for the changes reported in Fig. 1h,i, we found significant changes in more than half of the measurements and some of the changes reported are of 25% with a large signal to noise (standard

errors) ratio, so we have to disagree with the reviewer that those are modest. They are all likely to produce significant loss of synapses (and hence inhibition).

6. E_{Cl} measurements in the whole-cell patch clamp mode (Fig. 3): these experiments are not well explained and appear to be misguided. I didn't find the extracellular solution composition but assuming it is in the range of 150 mM, in combination with 25 mM Cl⁻ in the pipette, Nernst equation gives E_{Cl} around -45 mV, which is close to what was measured (-44.1). But what is the point of such a measurement? Surely E_{Cl} will be whatever you put in to your solutions! I assume the physiological E_{Cl} in SDH neurons is much more negative than -45 mV. If you want to measure realistic E_{Cl} and its changes, you would need to do gramicidin perforated patch recordings.

Response: Proper measurement of Cl⁻ extrusion capacity has to be performed in a Cl⁻ load condition, not with gramicidin perforated patch conditions, where E_{GABA} can easily be maintained with weak KCC2 under low load conditions, precluding detection of changes in KCC2 function. This has been well established by many authors (e.g., DeFazio et al., *J.Neurosci.* 2000²⁴; Rivera et al. *J.Cell.Biol.* 2002²⁵; Cordero-Erausquin et al. *J.Neurosci.* 2005²⁶; Gagnon et al. *Nat.Med.* 2013³; Ferrini et al *Nat.Neurosci.* 2013)²⁷. In these papers, we and others have described and validated the Cl⁻ load approach extensively. For a review of this issue, the reviewer is referred to Doyon *et al.*, *Neuron* 2016²⁸.

The value we measured is not that of E_{Cl} , but of E_{GABA} . Given that the GABA channels are permeable to both Cl⁻ and HCO₃⁻ (4:1), E_{GABA} is given by the Goldman-Hodgkin-Katz (GHK) equation, not Nernst. Under the conditions we used (concentrations given in Methods section "Rat spinal cord slices preparation and drug incubation" and "Electrophysiology"), the theoretical value of E_{GABA} according to the GHK equation is -37 mV, comparable to that obtained in the presence of the KCC2 blocker VU024055 (-36.1 ± 1.0 mV; Fig. 3f), within error, especially taking into account the variable purity of chloride salts between commercial sources (DeFazio et al., *J.Neurosci.* 2000²⁴; Cordero-Erausquin et al. *J.Neurosci.* 2005²⁶). The difference between E_{GABA} in normal condition vs. that in the presence of VU024055 provides an index of the KCC2-dependent Cl⁻ extrusion capacity of the cell. For greater clarity we have reiterated these principles in the results section on page 3, and method section page 17.

7. It is somewhat surprising that 3 hrs of BDNF treatment in slice condition was sufficient for the GABAA subunit switch.

Response: There are plenty of examples in the literature, starting with the entire LTP literature, showing rapid changes, of the order of minutes, in receptor density/composition at synapses. Thus, it does not appear necessarily surprising to us that BDNF could cause a change in subunit composition within 3h of incubation. In fact, we have previously shown that BDNF causes functional and structural downregulation of KCC2 within an hour of incubation (e.g., Coull *et al.*, *Nature* 2005²⁹; Gagnon *et al.*, *Nat. Med.* 2013³).

8. What are the conclusions from EGTA/BAPTA experiments? Both are Ca²⁺ chelators, are you implying some local Ca²⁺ microdomains to be involved – what are these? These results are just left alone without any follow-up experiments or even a discussion.

Response: The reviewer is right that the conventional conclusion of a differential effect of EGTA and BAPTA is due to buffer kinetics, the slower kinetics of EGTA rendering it incapable of clamping certain fast Ca²⁺ transients. These kinds of incongruities between EGTA and BAPTA have been reported before by Wang *et al.*, 1993, *Neurosci. Letters* 1993³⁰. The original sentence we had in the manuscript "Intracellular Ca²⁺ transients thus appear necessary for the GABA_AR subunit switch" was incomplete. We have expanded this comment. But, as stated above, this is a technical point that sidetracks from the focus of the paper. We have thus moved this technical data into Supplementary Fig. 6 as it is simply aimed at documenting the recording conditions to reconcile potential discrepancies across studies.

*9. Behavioral experiments with L838,417: "Anti-hyperalgesia by $\alpha 2$ GABAAR has been shown to preferentially occur via a spinal site of action and not through supraspinal sites." While supraspinal sites may be not involved, but peripheral sites do; see Obradovic *et al.* 2015 *Anesthesiology* 123:654-67. Alpha2 is expressed abundantly in peripheral fibers and provides analgesia in neuropathic pain models. Hence, the results presented in Fig. 4 are uninterpretable.*

Response: As mentioned above, the reviewer's comment is in contradiction with reports from H.U. Zeilhofer's group. Indeed selective deletion of the $\alpha 2$ in primary afferents had no effect on benzodiazepine-induced analgesia in the nerve injury model (Witschi *et al.*, *J.Neurosci.* 2011)⁷. In contrast, total deletion of $\alpha 2$ at the spinal level significantly attenuated benzodiazepine-induced analgesia (Paul *et al.*, *Neuropsychopharmacology*, 2014)⁸, indicating that $\alpha 2$ on dorsal horn neurons is a prime target for analgesia after nerve injury. The reference quote by the reviewer used topical administration of an antisense. This approach cannot differentially target afferent vs. dorsal horn subunits. In contrast, the genetic approach used by Zeilhofer allows selective targeting of afferents. Our conclusions are consistent with the findings from Zeilhofer's group. Furthermore, the result in Fig. 4c shows that L838,417 does not produce significant analgesia in control animals. The effect was only observed in animals with nerve injury consistent with our observation of a subunit switch at inhibitory synapses on dorsal horn neurons (identified by gephyrin clusters since as outlined in the paper, gephyrin clusters are absent from primary afferent terminals⁶).

10. KCC2 staining (Fig. 5): there seem to be **high background**, while in selected examples (white boxes) there membranous staining is discernable, the other neuronal cell bodies appear variably; the background level seems different between different conditions, not only the membrane staining – all this lowers the enthusiasm about the robustness of the analysis.

This staining is not background noise but represents staining of neuronal dendrites. To better illustrate and explain this point, in addition to the confocal acquisition in Fig. 5a,c,e, some KCC2(+) dendrites are clearly visible by electron microscopy in Fig.5h.

Minor.

11. Page 3, end of 2nd paragraph: "These results indicate a loss of inhibitory synapse scaffolding protein expression..." – how did you arrive at this conclusion?

In text: '...Both *Gphn* mRNA transcripts, encoding the inhibitory postsynaptic scaffolding protein, gephyrin, and gene *SLC12A5* encoding KCC2 were significantly **decreased to $52 \pm 11\%$ ($P < 0.05$)** and $47 \pm 6\%$ ($P < 0.01$), respectively;...'

See also Fig. 2d.

12. Page 4, end of 2nd paragraph: "... a function blocking anti-TrkB antibody..." – can you explain 'function-blocking' and provide a reference?

We have added the original reference as well as our previous reference on the topic (Balkowiec *et al.*, *J. Neurosci.* 2000³¹; Coull *et al.*, *Nature* 2005²⁹). "Function blocking" indicates that the antibody does not only recognized the receptor, but obstructs function (*e.g.*, ligand-binding) and thus acts as a receptor antagonist. We have added a statement in Methods to clarify this (Section entitled "Rat spinal cord slices preparation and drug incubation").

13. Please explain this para: "The antibodies against the α -GABAAR subtypes used in this study were bought from Synaptic Systems (Ssys) and raised in rabbit for the first set of experiments. For the second set of experiments, they were generously provided by Dr. Jean-Marc Fritschy and raised in guinea pig" – why did you use different antibodies for different experiments? Where is the first set of experiments starts and second begins, are there only two sets of experiments in this paper?

Response: We agree with referee#3 that this explanation was unclear. We added a table sheet (table #2) with all the antibodies used in this study and a column to show in what figure each antibody was used. Also we clarified this point in the methodological section of our paper:

"Anti-GABA_A receptor subtype-specific antibodies. The antibodies against the α -GABA_AR subtypes used in this study were bought from Synaptic Systems (Ssys) and raised in rabbit. A set of complementary experiments using guinea pig anti- $\alpha 1,2,3$ GABA_AR antibodies is illustrated in Supplementary Fig. 1, the antibodies were generously provided by Dr. Jean-Marc Fritschy. "

References:

1. Gagnon, M. *et al.* Reply to The small molecule CLP257 does not modify activity of the K(+)-Cl(-) co-transporter KCC2 but does potentiate GABAA receptor activity. *Nat. Med.* **23**, 1396-1398 (2017).
2. Cardarelli, R.A. *et al.* The small molecule CLP257 does not modify activity of the K(+)-Cl(-) co-transporter KCC2 but does potentiate GABAA receptor activity. *Nat. Med.* **23**, 1394-1396 (2017).
3. Gagnon, M. *et al.* Chloride extrusion enhancers as novel therapeutics for neurological diseases. *Nat. Med.* (2013).
4. Dzhala, V.I., Brumback, A.C., & Staley, K.J. Bumetanide enhances phenobarbital efficacy in a neonatal seizure model. *Ann. Neurol.* **63**, 222-235 (2008).
5. Coull, J.A. *et al.* Trans-synaptic shift in anion gradient in spinal lamina I neurons as a mechanism of neuropathic pain. *Nature* **424**, 938-942 (2003).
6. Lorenzo, L.E. *et al.* Gephyrin Clusters Are Absent from Small Diameter Primary Afferent Terminals Despite the Presence of GABAA Receptors. *J. Neurosci.* **34**, 8300-8317 (2014).
7. Witschi, R. *et al.* Presynaptic α 2-GABAA Receptors in Primary Afferent Depolarization and Spinal Pain Control. *J. Neurosci.* **31**, 8134-8142 (2011).
8. Paul, J. *et al.* Antihyperalgesia by α 2-GABA Receptors Occurs Via a Genuine Spinal Action and Does Not Involve Supraspinal Sites. *Neuropsychopharmacology* (2013).
9. Chaplan, S.R., Bach, F.W., Pogrel, J.W., Chung, J.M., & Yaksh, T.L. Quantitative assessment of tactile allodynia in the rat paw. *J. Neurosci. Methods* **53**, 55-63 (1994).
10. Colburn, R.W. *et al.* Attenuated cold sensitivity in TRPM8 null mice. *Neuron* **54**, 379-386 (2007).
11. Echeverry, S. *et al.* Spinal microglia are required for long-term maintenance of neuropathic pain. *Pain* (2017).
12. Dani, A., Huang, B., Bergan, J., Dulac, C., & Zhuang, X. Superresolution imaging of chemical synapses in the brain. *Neuron* **68**, 843-856 (2010).
13. Durand, A. *et al.* A machine learning approach for online automated optimization of super-resolution optical microscopy. *Nat. Commun.* **9**, 5247 (2018).
14. Huang, B., Babcock, H., & Zhuang, X. Breaking the diffraction barrier: super-resolution imaging of cells. *Cell* **143**, 1047-1058 (2010).
15. Godin, A.G., Lounis, B., & Cognet, L. Super-resolution microscopy approaches for live cell imaging. *Biophys. J.* **107**, 1777-1784 (2014).
16. Sugiyama, Y., Kawabata, I., Sobue, K., & Okabe, S. Determination of absolute protein numbers in single synapses by a GFP-based calibration technique. *Nat. Methods* **2**, 677-684 (2005).
17. Godin, A.G. *et al.* Revealing protein oligomerization and densities in situ using spatial intensity distribution analysis. *Proc. Natl. Acad. Sci. U. S. A.* **108**, 7010-7015 (2011).
18. Godin, A.G. *et al.* Spatial Intensity Distribution Analysis Reveals Abnormal Oligomerization of Proteins in Single Cells. *Biophys. J.* **109**, 710-721 (2015).
19. Sergeev, M. *et al.* Determination of membrane protein transporter oligomerization in native tissue using spatial fluorescence intensity fluctuation analysis. *PLoS One* **7**, e36215 (2012).
20. Patrizio, A. & Specht, C.G. Counting numbers of synaptic proteins: absolute quantification and single molecule imaging techniques. *Neurophotonics* **3**, 041805 (2016).
21. Korobchevskaya, K., Lagerholm, C.B., Colin-York, H., & Fritzsche, M. Exploring the Potential of Airyscan Microscopy for Live Cell Imaging. *Photonics* **4**, (2017).
22. Lorenzo, L.E. *et al.* Spatial and temporal pattern of changes in the number of GAD65-immunoreactive inhibitory terminals in the rat superficial dorsal horn following peripheral nerve injury. *Mol. Pain* **10**, 57 (2014).
23. Beggs, S. & Salter, M.W. Stereological and somatotopic analysis of the spinal microglial response to peripheral nerve injury. *Brain Behav. Immun.* **21**, 624-633 (2007).
24. DeFazio, R.A., Keros, S., Quick, M.W., & Hablitz, J.J. Potassium-coupled chloride cotransport controls intracellular chloride in rat neocortical pyramidal neurons. *J. Neurosci.* **20**, 8069-8076 (2000).
25. Rivera, C. *et al.* BDNF-induced TrkB activation down-regulates the K+-Cl- cotransporter KCC2 and impairs neuronal Cl- extrusion. *J. Cell Biol.* **159**, 747-752 (2002).
26. Cordero-Erausquin, M., Coull, J.A., Boudreau, D., Rolland, M., & De Koninck, Y. Differential maturation of GABA action and anion reversal potential in spinal lamina I neurons: impact of chloride extrusion capacity. *J. Neurosci.* **25**, 9613-9623 (2005).
27. Ferrini, F. *et al.* Morphine hyperalgesia gated through microglia-mediated disruption of neuronal Cl(-) homeostasis. *Nat. Neurosci.* **16**, 183-192 (2013).
28. Doyon, N., Vinay, L., Prescott, S.A., & De Koninck, Y. Chloride Regulation: A Dynamic Equilibrium Crucial for Synaptic Inhibition. *Neuron* **89**, 1157-1172 (2016).
29. Coull, J.A. *et al.* BDNF from microglia causes the shift in neuronal anion gradient underlying neuropathic pain. *Nature* **438**, 1017-1021 (2005).
30. Wang, Y.T., Pak, Y.S., & Salter, M.W. Rundown of NMDA-receptor mediated currents is resistant to lowering intracellular $[Ca^{2+}]$ and is prevented by ATP in rat spinal dorsal horn neurons. *Neurosci. Lett.* **157**, 183-186 (1993).
31. Balkowiec, A. & Katz, D.M. Activity-dependent release of endogenous brain-derived neurotrophic factor from primary sensory neurons detected by ELISA in situ. *J. Neurosci.* **20**, 7417-7423 (2000).

REVIEWERS' COMMENTS:

Reviewer #1 (Remarks to the Author):

I want to thank the authors for taking into consideration the remarks and requests made by the reviewers. I consider that they have convincingly answered the main questions. This reinforced their argument. I think the article is now ready to be published in its current form.

Reviewer #2 (Remarks to the Author):

The authors have done an excellent job in addressing comments of all reviewers. The reviewer is satisfied with the changes made. This findings of this study are important and will make a major impact on the field.

-

Rohini Kuner